# Multiclass Learnability Beyond the PAC Framework: Universal Rates and Partial Concept Classes

**Alkis Kalavasis***
National Technical University of Athens
kalavasisalkis@mail.ntua.gr

**Grigoris Velegkas***
Yale University
grigoris.velegkas@yale.edu

**Amin Karbasi**
Yale University, Google Research
amin.karbasi@yale.edu

## Abstract

In this paper we study the problem of multiclass classification with a bounded number of different labels $k$, in the realizable setting. We extend the traditional PAC model to **a)** distribution-dependent learning rates, and **b)** learning rates under data-dependent assumptions. First, we consider the *universal learning* setting (Bousquet, Hanneke, Moran, van Handel and Yehudayoff, STOC'21), for which we provide a complete characterization of the achievable learning rates that holds for every fixed distribution. In particular, we show the following trichotomy: for any concept class, the optimal learning rate is either exponential, linear or arbitrarily slow. Additionally, we provide complexity measures of the underlying hypothesis class that characterize when these rates occur. Second, we consider the problem of multiclass classification with *structured* data (such as data lying on a low dimensional manifold or satisfying margin conditions), a setting which is captured by *partial concept classes* (Alon, Hanneke, Holzman and Moran, FOCS'21). Partial concepts are functions that can be *undefined* in certain parts of the input space. We extend the traditional PAC learnability of total concept classes to partial concept classes in the multiclass setting and investigate differences between partial and total concepts.

## 1 Introduction

Classifying data into multiple different classes is a fundamental problem in machine learning that has many real-life applications, such as image recognition, web advertisement and text categorization. Due to its importance, multiclass classification has been an attractive field of research both for theorists [NT88, BCHL95, RBR09, Nat89, DSBDSS11, DSS14, SSBD14] and for practitioners [SSKS04, Col04, Aly05]. Essentially, it boils down to learning a classifier $h$ from a domain $\mathcal{X}$ to a label space $\mathcal{Y}$, where $|\mathcal{Y}| \geq 2$, and the error is measured by the probability that $h(x)$ is incorrect. In this work, we focus on the setting where the number of labels is finite and we identify $\mathcal{Y}$ with $[k] := \{0, 1, ..., k\}$ for some constant $k \in \mathbb{N}$.

**Multiclass PAC Learning.** The PAC model [Val84] constitutes the gold-standard learning framework. A seminal result in learning theory [VC71, BEHW89] characterizes PAC learnability of binary classes ($k = 1$) through the Vapnik-Chervonenkis (VC) dimension and provides a clear algorithmic landscape with the *empirical risk minimization* (ERM) principle yielding (almost) optimal statistical learning algorithms. The picture established for the binary setting extends to the case of multiple labels when

---

*Equal contribution.

36th Conference on Neural Information Processing Systems (NeurIPS 2022).

the number of classes $|\mathcal{Y}|$ is bounded. The works of [NT88, Nat89] and [BCHL95] identified natural extensions of the VC dimension such as the *Natarajan dimension* whose finiteness characterizes multiclass learnability in this setting. Moreover, the ERM principle still holds and achieves the desired learning rate by essentially reducing learnability to optimization. The fundamental result of the multiclass PAC learning (in the realizable setting) can be summarized in the following elegant equation for any $n \in \mathbb{N}$ which we explain right-after:

$$\inf_{\widehat{h}_n} \sup_{P \in \mathrm{RE}(\mathcal{H})} \mathbf{E}[\mathrm{er}(\widehat{h}_n)] = \min\left( \widetilde{\Theta}_k \left( \frac{\mathrm{Ndim}(\mathcal{H})}{n} \right), 1 \right), \tag{1}$$

where $\mathrm{Ndim}(\mathcal{H})$ stands for the Natarajan of $\mathcal{H}$ and $\widetilde{\Theta}_k$ subsumes logarithmic factors and dependencies on $k$. In words, assume that $n$ is the number of training samples and let $\mathcal{H} \subseteq [k]^{\mathcal{X}}$ be a set of multiclass classifiers mapping the elements of the domain $\mathcal{X}$ to $[k]$ that the learner has access to. The learner observes $n$ labeled examples $(x, y) \in \mathcal{X} \times [k]$ generated i.i.d. from some unknown distribution $P$ with the constraint that $P$ is *realizable* with respect to $\mathcal{H}$, i.e., there is some $h \in \mathcal{H}$ that has (almost surely) zero classification error. The learner then outputs a guess hypothesis $\widehat{h}_n : \mathcal{X} \to [k]$. The fundamental theorem of PAC learning, as shown above, controls the expected error $\mathbf{E}[\mathrm{er}(\widehat{h}_n)]$, where $\mathrm{er}(\widehat{h}_n) := \mathbf{Pr}_{(x,y) \sim P}[\widehat{h}_n(x) \neq y]$, in a *minimax* sense, i.e., it controls the performance of the best algorithm $\widehat{h}_n$ (inf) against the worst-case realizable distribution $P \in \mathrm{RE}(\mathcal{H})$ (sup) and states that the following dichotomy occurs: if the Natarajan dimension $\mathrm{Ndim}(\mathcal{H})$ is finite, the error rate decreases as, roughly, $1/n$, so $\mathcal{H}$ is PAC learnable at a linear rate; otherwise, the class $\mathcal{H}$ is not PAC learnable. Additionally, this theory provides a clean algorithmic landscape: the ERM principle, which means outputting some classifier in $\mathcal{H}$ that best fits the training set, achieves the rates of Eq. (1).

**Towards Novel Learning Theories.** While the PAC model provides a solid and attractive theoretical framework, it fails to (fundamentally) capture the real-world behavior of various applied ML problems. In this work, we focus on the following two points of criticism for the standard PAC model. The first natural point concerns the supremum over all realizable distributions in Eq. (1).

**Observation 1.** *The PAC model is distribution-independent and captures the worst-case learning rate. Is it possible to design a learning theory that provides distribution-dependent learning rates?*

Another critique is that one cannot express natural data-dependent assumptions through the PAC framework. For instance, high-dimensional data may lie in a low-dimensional manifold. To be more specific, consider the task of classifying images of vehicles. The representation of such images corresponds to a low-dimensional subset of the space of all possible images, most of which do not correspond to vehicles. A prominent way to capture such assumptions is via *partial concepts*: these are functions which can be *undefined* on a subset of $\mathcal{X}$, a departure from the traditional model.

**Observation 2.** *The PAC model only considers total concept classes, i.e., $\mathcal{H} \subseteq [k]^{\mathcal{X}}$, which cannot express data-dependent constraints. Is it possible to design a learning theory for partial concepts $h : \mathcal{X} \to \{0, 1, ..., k, \star\}$, where $h(x) = \star$ means that $h$ is undefined at $x$?*

The aim of this paper is to develop (i) a distribution-dependent learning theory for multiclass classification and (ii) a learning theory for partial multiclass concept classes in the distribution-independent setting[2]. We comment that we focus on the realizable setting that already poses important challenges and requires novel ideas and we believe that our results can be extended to the agnostic case, which is left for future work. We remark that such theories for binary classification were recently developed [BHM+21, AHHM22]. However, in various practical applications, such explanations may not suffice, since it is rarely the case that there are only two classes. As it is already evident from the PAC setting, moving from binary classification to multiclass classification is not trivial [DSBDSS11]. We now discuss Observation 1 and 2; we underline that our goal is not to replace, but to build upon and complement, the traditional PAC model, which constitutes the bedrock of learning theory.

**Distribution-Dependent Learning Rates.** In many modern machine learning applications the generalization error $\mathbf{E}[\mathrm{er}(\widehat{h}_n)]$ drops *exponentially fast* as a function of the sample size $n$ [CT90, CT92, Sch97, VL21]. However, the traditional PAC learning theory predicts merely $\widetilde{O}(d/n)$ rates in the realizable setting, where $d$ is the complexity measure of the underlying concept class that the algorithm is trying to learn. A possible explanation for this discrepancy between the theoretical

---

[2]While we believe that one could design a unified learning theory addressing the two questions at once, we prefer to provide two separate theories, since they are both interesting in their own.

guarantees and the empirical performance of the learning algorithms is the worst-case nature of the PAC guarantees. Notice that in Eq. (1), for any fixed learning algorithm, one considers its performance against the worst distribution for it. In particular, this means that as the sample size $n$ increases and new classifiers $\widehat{h}_n$ are produced, the distribution that is used as a benchmark can differ. However, in many practical applications, one considers some *fixed* distribution $P$ and measures the performance of the classifier as $n \to \infty$ without changing $P$. Hence, there is an important need to study mathematical models that capture this behavior of learning algorithms and not just the minimax one. One such approach that was recently proposed by [BHM+21] is to study *universal learning rates*, which means that the learning rates guarantees hold for *every* fixed (realizable) distribution $P$, but there is not a *uniform* bound over all of the distributions. To be more precise, $\mathcal{H}$ is learnable at rate $R$ (where $\lim_{n \to \infty} R(n) = 0$) in the universal setting if

$$\exists \widehat{h}_n : \forall P \in \mathrm{RE}(\mathcal{H}), \ \exists C = C(P), c = c(P) > 0 \text{ so that } \mathbf{E}[\mathrm{er}(\widehat{h}_n)] \leq C \cdot R(c \cdot n), \forall n \in \mathbb{N}. \quad (2)$$

Note that the above equation is the same as in the PAC model with the exception of a change between the existential quantifiers: in PAC, the focus is on the case where $\exists C, c > 0 : \forall P \in \mathrm{RE}(\mathcal{H})$ the guarantee holds (which expresses *uniformity*), while in the universal setting distribution-dependent constants are allowed. This subtle change in the definition can make the error-rate landscape vastly different. As an example, consider the case where for $\widehat{h}_n$, we have that $\mathrm{er}(\widehat{h}_n) \leq C(P)e^{-c(P)n}$, for every distribution $P$, where $C(P), c(P)$ are some distribution-dependent constants. When we take the pointwise supremum over all of these infinitely many distributions, it can be the case that the resulting function drops as $C'/n$, where $C'$ is a distribution-independent constant [BHM+21].

**Partial Concept Classes.** The motivation behind Observation 2 is that, in various practical learning tasks, the data satisfy some special properties that make the learning process simpler. For instance, it is a common principle to use classification with margin where the points in the dataset have a safe gap from the decision boundary. Such properties induce *data-dependent* assumptions that the traditional PAC learning theory framework provably fails to express. In fact, existing data-dependent analyses diverge from the standard PAC model [STBWA98, HW01] and provide problem-specific approaches. Thus, there is a need for a formal framework that allows us to express such data-dependent restrictions and study these problems in a *unified and principled way*. Recently, [AHHM22] proposed an elegant extension of the binary PAC model to handle such tasks via the framework of *partial concept classes*. As an intuitive example, a halfspace with margin is a partial function that is undefined inside the forbidden margin and is a well-defined halfspace outside the margin boundaries.

## 1.1 The Traditional Multiclass Learning Problem

Let $[k] = \{0, 1, ..., k\}$ for some fixed positive integer $k \in \mathbb{N}$. We consider a domain $\mathcal{X}$ and a concept class $\mathcal{H} \subseteq [k]^{\mathcal{X}}$. A classifier is a universally measurable[3] function $h : \mathcal{X} \to [k]$. The error rate of a classifier $h$ with respect to a probability distribution $P$ on $\mathcal{X} \times [k]$ is equal to $\mathrm{er}(h) = \mathrm{er}_P(h) = \mathbf{Pr}_{(x,y) \sim P}[h(x) \neq y]$. We focus on the setting where $P$ is realizable, i.e., $\inf_{h \in \mathcal{H}} \mathrm{er}_P(h) = 0$. Formally, a (deterministic[4]) learning algorithm is a sequence of universally measurable functions, which take as input a sequence of $n$ independent pairs $(X_i, Y_i) \sim P$ (training set) and output a classifier $\widehat{h}_n : \mathcal{X} \to \mathcal{Y}$. The goal is to come up with algorithms whose $\mathbf{E}[\mathrm{er}(\widehat{h}_n)]$ admits a fast decay as a function of $n$, where the expectation is over the training set.

## 1.2 Universal Multiclass Learning: Our Results

The aim of our first theory is to fully characterize the admissible universal rates of learning, i.e., $\mathbf{E}[\mathrm{er}(\widehat{h}_n)]$, in the multiclass classification setting with a bounded number of labels. The following definition formalizes this notion of achievable rate in the (realizable) universal learning model [BHM+21].

**Definition 1** ([BHM+21]). *Let $\mathcal{H} \subseteq [k]^{\mathcal{X}}$ and let $R : \mathbb{N} \to [0, 1]$, with $R(n) \to 0$, be a rate function. We say that $\mathcal{H}$ is **learnable at rate** $R$ if there exists a learning algorithm $\widehat{h}_n$ such that for every realizable distribution $P$ on $\mathcal{X} \times [k]$ with respect to $\mathcal{H}$, there exist distribution-dependent $C, c > 0$ for which $\mathbf{E}[\mathrm{er}(\widehat{h}_n)] \leq CR(cn)$, for all $n \in \mathbb{N}$. Also, $\mathcal{H}$ is **not learnable at rate faster than** $R$ if for any learning algorithm $\widehat{h}_n$, there exists a realizable distribution $P$ on $\mathcal{X} \times [k]$ with respect to $\mathcal{H}$ and*

---

[3]We discuss measurability formally in Appendix B.5

[4]We focus for simplicity on deterministic learners. Our results extend to randomized algorithms.

*distribution-dependent $C, c > 0$ for which $\mathbf{E}[\mathrm{er}(\widehat{h}_n)] \geq CR(cn)$ for infinitely many $n \in \mathbb{N}$. $\mathcal{H}$ is **learnable with optimal rate** $R$ if it is learnable at rate $R$ and is not learnable faster than $R$. Finally, $\mathcal{H}$ requires **arbitrarily slow rates** if, for every $R(n) \to 0$, $\mathcal{H}$ is not learnable at rate faster than $R$.*

In the universal multiclass setting, we show that the following fundamental trichotomy occurs (in comparison with the dichotomy witnessed in the uniform PAC model). This result is a theoretical justification of the exponential error rates observed in practice.

**Theorem 1.** *Fix a constant $k \in \mathbb{N}$. Consider a hypothesis class $\mathcal{H} \subseteq [k]^{\mathcal{X}}$ with $|\mathcal{H}| > k + 2$. Then, exactly one of the following holds for the learning rate of $\mathcal{H}$ in the realizable case:*

- *$\mathcal{H}$ is learnable at an optimal rate $e^{-n}$.*

- *$\mathcal{H}$ is learnable at an optimal rate $1/n$.*

- *$\mathcal{H}$ requires arbitrarily slow rates.*

We mention that $|\mathcal{H}| > k + 2$ comes without loss of generality.[5] In contrast to the standard PAC model, any concept class is learnable in the universal rates setting [HKSW20]. The analogue of non-learnability in the uniform setting is the case of arbitrarily slow rates. Our second result is the specification of some combinatorial complexity measures of $\mathcal{H}$ that characterize the optimal learning rate of this class. Let us first provide some informal definitions of these measures. We begin with the notion of *multiclass Littlestone trees*, which extends the binary Littlestone trees from [BHM+21].

**Definition 2** (Informal (see Definition 11)). *A **multiclass Littlestone tree** for $\mathcal{H} \subseteq [k]^{\mathcal{X}}$ is a complete binary tree of depth $d \leq \infty$ whose internal nodes are labeled by $\mathcal{X}$, and whose two edges connecting a node to its children are labeled by two different elements in $[k]$, such that every path of length at most $d$ emanating from the root is consistent with a concept $h \in \mathcal{H}$. We say that $\mathcal{H}$ has an **infinite multiclass Littlestone tree** if there is a multiclass Littlestone tree for $\mathcal{H}$ of depth $d = \infty$.*

For some intuition we refer the reader to Figure 1. The above complexity measure appears in the definition of the multiclass Littlestone dimension [DSBDSS11]. In fact, a class $\mathcal{H} \subseteq [k]^{\mathcal{X}}$ has multiclass Littlestone dimension $d$ if it has a multiclass Littlestone tree of depth $d$ but not of depth $d + 1$. We underline that having an infinite multiclass Littlestone tree is **not** the same as having an unbounded multiclass Littlestone dimension. A class $\mathcal{H}$ has unbounded Littlestone dimension if for every $d \in \mathbb{N}$ there is *some* tree of depth $d$. However, this does not mean that there is a *single* infinite tree. This is a fundamental conceptual gap between the uniform and the universal settings.

The next definition is novel and is motivated by the fundamental notion of the Natarajan dimension from the multiclass PAC setting (see Definition 7). We first need some terminology: a tuple $(x_1, ..., x_t, s_1^{(0)}, ..., s_t^{(0)}, s_1^{(1)}, ..., s_t^{(1)}) \in \mathcal{X}^t \times [k]^t \times [k]^t$ with $s_i^{(0)} \neq s_i^{(1)}$, for any $i \in [t]$, is $N$-**consistent** with the edge $(y_1, ..., y_t) \in \{0, 1\}^t$ and the concept $h \in \mathcal{H}$ if $h(x_i) = s_i^{(y_i)}$ for any $i \in [t]$. Recall that if the tuple is $N$-consistent with any binary pattern $y \in \{0, 1\}^t$, we say that $(x_1, ..., x_t)$ is $N$-**shattered**. More generally, a path is $N$-consistent with a concept $h \in \mathcal{H}$ if each node of the path is $N$-consistent with the edge connecting the node with its child across the path and $h$. Since the next definition might be hard to parse, we refer the reader to Figure 2 for some intuition.

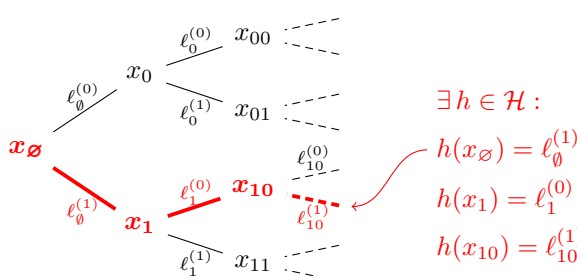

Figure 1: A multiclass Littlestone tree of depth 3. Each node $x_u$ has two children $\ell_u^{(0)} \neq \ell_u^{(1)}$ where $\ell_u^{(i)} \in [k]$ for any $i \in \{0, 1\}$ and $u \in \{0, 1\}^*$, i.e., the set of binary strings of arbitrary length. Every branch is consistent with some concept $h \in \mathcal{H}$. The figure is adapted from [BHM+21].

**Definition 3** (Informal (see Definition 12)). *A **Natarajan-Littlestone (NL) tree** for $\mathcal{H} \subseteq [k]^{\mathcal{X}}$ is a complete tree of depth $d \leq \infty$ so that every level $1 \leq t \leq d$ has branching factor $2^t$ and nodes that*

---

[5]The constraint $|\mathcal{H}| > k + 2$ rules out some degenerate scenarios, we kindly refer to Appendix A.2.

*are labeled by $\mathcal{X}^t \times [k]^t \times [k]^t$ (so that for all $i \in [t]$ the two labels in $[k] \times [k]$ are different) and whose $2^t$ edges connecting a node to its children are labeled by the elements of $\{0,1\}^t$. It must hold that every path of length at most $d$ emanating from the root is $N$-consistent with a concept $h \in \mathcal{H}$. We say that $\mathcal{H}$ has an **infinite Natarajan-Littlestone tree** if there is an NL tree for $\mathcal{H}$ of depth $d = \infty$.*

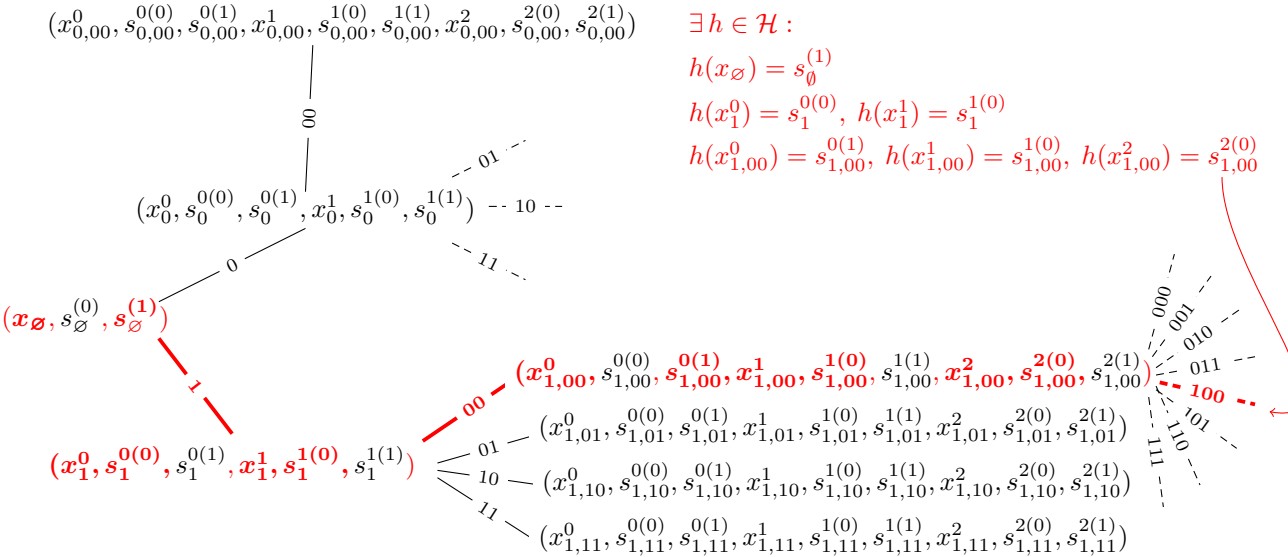

Figure 2: A Natarajan-Littlestone tree of depth 3. Every branch is consistent with a concept $h \in \mathcal{H}$. This is illustrated here for one of the branches. Due to lack of space, not all nodes and external edges are drawn. The figure is adapted from [BHM$^+$21]. The root of the tree is the point $(x_\emptyset)$ with two colors $s_\emptyset^{(0)} \neq s_\emptyset^{(1)}$. In this example, the branch picks the string $'1'$ and hence the node of the second level contains the two points $(x_1^0, x_1^1)$ and the associated colors. We proceed in a similar manner.

An NL tree looks like a multiclass Littlestone tree whose branching factor increases exponentially with the depth of the tree and where each node at depth $t$ in the NL tree contains $t$ points $x_1, ..., x_t$ of $\mathcal{X}$ and two colorings $s^{(0)}, s^{(1)}$ so that $s^{(0)}(x_i) \neq s^{(1)}(x_i)$ for all $i \in [t]$. Crucially, this structure encapsulates the notion of $N$-shattering in the combinatorial structure of a Littlestone tree. Intuitively, along each path in the NL tree, we encounter $N$-shattered sets of size increasing with the depth. Using these two definitions, we can state our second result which is a complete characterization of the optimal rates achievable for any given concept class $\mathcal{H} \subseteq [k]^{\mathcal{X}}$.

**Theorem 2.** *Fix a constant positive integer $k$. Consider a hypothesis class $\mathcal{H} \subseteq [k]^{\mathcal{X}}$ with $|\mathcal{H}| > k+2$. Then, one of the following holds for any $n \in \mathbb{N}$ in the realizable case:*

- *If $\mathcal{H}$ does not have an infinite multiclass Littlestone tree, then it is learnable at an optimal rate $e^{-n}$.*

- *If $\mathcal{H}$ has an infinite multiclass Littlestone tree but does not have an infinite Natarajan-Littlestone tree, then it is learnable at an optimal rate $1/n$.*

- *If $\mathcal{H}$ has an infinite Natarajan-Littlestone tree, then it requires arbitrarily slow rates.*

It is clear that the above result implies Theorem 1. We remark that not only the achievable rates are different compared to the uniform setting, but also the algorithms we use to get these rates differ vastly from ERM. We sketch the main techniques in Section 2. For the formal proof, see Appendix C. We briefly summarize our main technical contributions in this setting: using the pre-described complexity measures of $\mathcal{H}$, we introduce novel Gale-Stewart games (see Section 1.4 for a definition) which lead to new learning algorithms that achieve the optimal learning rates. For the exponential rates setting, the algorithm that we obtain through the Gale-Stewart game cannot be directly used to make predictions, since it takes as input two labels and it outputs the correct one. Hence, our predictor, essentially, plays a tournament between all the potential labels using this function we just mentioned. The case of linear rates is also more technically involved compared to its binary counterpart and

this is related to the fact that the Natarajan-Littlestone tree has a more complicated structure than the VCL tree (for example, we need to check all the possible mappings from some given points to labels). The Gale-Stewart game that handles the case of linear rates goes as follows: the adversary presents to the learner a tuple of points $x$, and two (different) colorings for these points. Similarly as before, we could not use a simpler game to obtain the result. Also, we extend the lower bounds from [BHM$^+$21] that hold for binary classification to the multiclass setting using our combinatorial measures. Ensuring the realizability of the distribution in the case of arbitrarily slow rates is a bit delicate and it raises questions about how the equivalence between various combinatorial dimensions, like the Graph dimension and the Natarajan dimension, which is established in the PAC setting translates to the universal setting. For further details, we refer to Section 2 and Appendix C.5.

## 1.3 Partial Multiclass Learning: Our Results

As we mentioned earlier, traditional PAC learning cannot capture data-dependent assumptions. Inspired by [AHHM22], we slightly modify the basic multiclass learning problem in a quite simple manner: instead of dealing with concept classes $\mathcal{H} \subseteq \{0, 1, ..., k\}^{\mathcal{X}}$ where each concept $h \in \mathcal{H}$ is a **total function** $h : \mathcal{X} \to \{0, 1, ..., k\}$, we study **partial concept classes** $\mathcal{H} \subseteq \{0, 1, ..., k, \star\}^{\mathcal{X}}$, where each concept $h$ is now a **partial function** and $h(x) = \star$ means that the function $h$ is **undefined** at $x$. We define the support of $h$ as the set $\text{supp}(h) = \{x \in \mathcal{X} : h(x) \neq \star\}$. To illustrate the power of partial classes, we comment that the fundamental class of $d$-dimensional halfspaces with margin $\gamma > 0$ and $k$ labels can be cast as a partial class $\mathcal{H}_\gamma = \{h_W : W \in \mathbb{R}^{k \times d}\}$, where $h_W(x) = i \in [k]$ if $(W_i - W_j) \cdot x \geq \gamma$ for all $j \neq i$ and $h_W(x) = \star$ otherwise [AHHM22]. As another example, we can express the constraint that the data have to be in a low-dimensional space by considering the partial concept class $\mathcal{H} = \{h : \mathbb{R}^d \to \{0, 1, ..., k, \star\} : \dim(\text{supp}(h)) \ll d\}$, where $\dim(S)$ captures the dimension of the set of points in $S$. We characterize multiclass PAC learnability of partial concepts in the realizable setting. A distribution $P$ on $\mathcal{X} \times \{0, 1, ..., k\}$ is **realizable** by $\mathcal{H}$ if, almost surely, for any $n$, a training set $(x_i, y_i)_{i \in [n]} \sim P^n$ is realizable by some partial concept $h \in \mathcal{H}$, i.e., $\{x_i\}_{i \in [n]} \subseteq \text{supp}(h)$ and $h(x_i) = y_i$ for all $i \leq n$. For a partial concept $h$ and a distribution $P$ on $\mathcal{X} \times \{0, 1, ..., k\}$, we let $\text{er}_P(h) = \mathbf{Pr}_{(x,y) \sim P}[h(x) \neq y]$, i.e., whenever $h$ outputs $\star$ it is counted as a mistake. We mention that the standard combinatorial measures such as the VC or the Natarajan dimension *naturally extend to the partial setting*; e.g., a partial class $\mathcal{H}$ VC shatters a set of points if any *binary* pattern is realized by $\mathcal{H}$ (we forget about $\star$).

**Definition 4** (Multiclass Partial PAC Learnability [AHHM22])**.** *A partial concept class $\mathcal{H} \subseteq \{0, 1, ..., k, \star\}^{\mathcal{X}}$ is **PAC learnable** if for every $\epsilon, \delta \in (0, 1)$, there exists a finite $\mathcal{M}(\epsilon, \delta) \in \mathbb{N}$ and a learning algorithm $\mathbb{A}$ such that, for every distribution $P$ on $\mathcal{X} \times \{0, 1, ..., k\}$ realizable with respect to $\mathcal{H}$, for $S \sim P^{\mathcal{M}(\epsilon, \delta)}$, it holds that $\mathbf{Pr}_S[\text{er}_P(\mathbb{A}(S)) \leq \epsilon] \geq 1 - \delta$. The sample complexity of $\mathbb{A}$ is the value $\mathcal{M}(\epsilon, \delta)$ and the optimal sample complexity is the minimum possible value of $\mathcal{M}(\epsilon, \delta)$ for any given $\epsilon, \delta$.*

We provide a combinatorial characterization of multiclass PAC learnability in the partial setting with a bounded number of labels. We additionally give bounds for $\mathcal{M}(\epsilon, \delta)$ in our more general Theorem 6.

**Theorem 3.** *Fix a positive constant $k \in \mathbb{N}$. For any partial concept class $\mathcal{H} \subseteq \{0, 1, \ldots, k, \star\}^{\mathcal{X}}$, it holds that $\mathcal{H}$ is PAC learnable if and only if $\text{Ndim}(\mathcal{H}) < \infty$.*

Here, $\text{Ndim}(\mathcal{H})$ is the Natarajan dimension of $\mathcal{H}$ (see Definition 7 and Remark 2). At first sight this result may not seem surprising. However, its proof is different from the standard multiclass PAC learning (for constant $k$), which goes through uniform convergence and ERM. In fact, such tools provably fail [AHHM22]. For a sketch, see below Theorem 6. We complement the above structural result with some additional insight which sheds light towards the, perhaps unanticipated, complexity of partial concept classes. To this end, we discuss the question of *disambiguation* [AKM19, AHHM22]: Can a partial Natarajan class (i.e., with finite $\text{Ndim}$) be represented by total Natarajan classes? To address this task, the notion of disambiguation is required: roughly, a total class $\overline{\mathcal{H}}$ disambiguates the partial class $\mathcal{H}$ if every partial concept $h \in \mathcal{H}$ can be extended to some total concept $\overline{h} \in \overline{\mathcal{H}}$, i.e., $\overline{h}$ agrees with $h$ in the support of $h$ and assigns to the undefined points some labels. For a formal definition of disambiguation, see Definition 15. For the case $k = 1$, [AHHM22] provided an easy-to-learn partial class that cannot be represented by any total class of bounded VC dimension using, surprisingly, some recent results from communication complexity and graph theory (see e.g., [BBDG$^+$22]). We extend this result to the multiclass setting using Sauer's lemma, which provides a bound on the growth function [SSBD14], and tools from the binary impossibility result.

**Theorem 4** (Informal, see Theorem 16). *Fix $k \in \mathbb{N}$. For any $n \in \mathbb{N}$, there exists a class $\mathcal{H} \subseteq \{0, 1, ..., k, \star\}^{\mathbb{N}}$ with $\mathrm{Ndim}(\mathcal{H}) = O_k(1)$ such that any disambiguation $\overline{\mathcal{H}}$ of $\mathcal{H}$ has $\mathrm{Ndim}(\overline{\mathcal{H}}) = \infty$.*

We denote with $O_k(1)$ a constant that depends on $k$. Via Theorem 3, the above partial class $\mathcal{H}$ is PAC learnable; however, *any* disambiguation of $\mathcal{H}$ causes a blow-up to the Natarajan dimension. This result showcases the complexity of partial concepts. We briefly outline the main technical contributions in this regime: we extend the main results of [AHHM22] for binary classification to the multiclass setting using appropriate complexity measures. Recall that the combinatorial Sauer-Shelah-Perles (SSP) lemma [Sau72, SSBD14] bounds the size of a (total) class $\mathcal{H} \subseteq \{0, 1\}^n$ by $\sum_{i=0}^{\mathrm{VC}(\mathcal{H})} \binom{n}{i}$. Notably, [AHHM22] showed that this lemma does *not* hold true for partial concept classes. To obtain our disambiguation result, we prove that the second variant of the SSP lemma, which uses the growth function of the class [SSBD14], *does* hold in the partial regime, which may be of independent interest.

## 1.4 Preliminaries and Related Work

**Preliminaries.** In this section, we discuss some important preliminaries for this paper. Due to space constraints, we refer the reader to Appendix B for a more extensive discussion.

**Gale-Stewart Games.** An important tool we leverage to establish our results in the universal learning setting is the theory of *Gale-Stewart* (GS) *games* [GS53]. Every such game consists of two players, a learner $\mathsf{P}_L$ and an adversary $\mathsf{P}_A$, and is played over an infinite sequence of discrete steps. In each step, the adversary presents some point $x_t \in \mathcal{X}_t$ to the learner and the learner picks a response $y_t \in \mathcal{Y}_t$. If some predefined condition gets violated at some step $t$, the game terminates and the learner wins. On the other hand, if the condition does not get violated during the infinite sequence of these time-steps, the adversary wins. The main property which characterizes the GS game is that the winning strategy of the learner is *finitely-decidable*, i.e., she knows that she has won the game after playing a finite number of rounds. [GS53, Kec12, HW+93] proved that either $\mathsf{P}_L$ or $\mathsf{P}_A$ has a winning strategy, i.e., playing that strategy makes them win regardless of the opponents actions. Similar to [BHM+21], the main reason we use GS games in this work is to obtain functions that are *eventually* correct.

**Related Work.** As we mentioned in the introduction, we focus on the well-studied problem of multiclass classification [Nat89, BCHL95, RBR09, DSBDSS11, DSS14, SSBD14, BCD+22, BDK+22]. For a more detailed overview, see Appendix A.1. Our work provides two theoretical perspectives complementing and extending the standard multiclass PAC learning. For the **universal rates**, the seminal work of [BHM+21] provided a similar trichotomy for the binary setting (we obtain their results by setting $k = 1$). The gap between exponential and linear rates was studied by [Sch97] in some special cases. Also, [AL98] showed that there exist concept classes for which no improvement on the PAC learning rate is possible in the universal setting. A natural approach to obtain results for the multiclass setting is via reductions to the binary setting. In the exponential rates setting, a first idea would be to consider for the class $\mathcal{H} \subseteq [k]^{\mathcal{X}}$ the binary restrictions $\mathcal{H}|_i = \{h_i : h \in \mathcal{H}\}$, where $h_i$ denotes the $i$-th bit of the output of $h$. In order to obtain the desired result for exponential rates, one has to prove that if $\mathcal{H}$ does not have an infinite multiclass Littlestone tree, then every $\mathcal{H}|_i$ does not have an infinite binary Littlestone tree. However, it is not clear how to obtain such a result. This is why we design statistical learning algorithms for the multiclass setting from scratch, following the conceptual roadmap introduced in [BHM+21]. We note that the existence of multiple labels requires various novel ideas compared to the binary setting. We introduce novel Gale-Stewart games (see Algorithm 1, Algorithm 2) in order to provide winning strategies for the learning player both in the exponential and the linear rates settings. Finally, in terms of reductions for the linear rates setting, we provide a sufficient condition for learnability at a linear rate using a reduction to the binary setting; however, it is again not clear how to use these complexity measures in order to obtain lower bounds (see Open Question 1). In general, the idea of "universal learning" has been studied in various concepts such as universal consistency [Sto77, DGL13, Han20, HKSW20, BC21, Han22, Bla22, BJ22, BCH22] and active learning [Han09, BHV10, Han11, HY15]. For an extended discussion, we refer to [BHM+21]. For the **partial concepts** setting, our work builds on the seminal work [AHHM22] and uses tools from [BCD+22]. The work of [AHHM22] shows that the algorithmic landscape behind partial concept classes is quite elusive. We extend some of their results to learning scenarios where one deals with more than two labels. Our contributions draw ideas from various directions, namely the one-inclusion hypergraph algorithm [HLW94, RBR09, DSS14, BCD+22] which we define formally in Appendix B.2, the Sauer-Shelah-Perles lemma [Sau72] and recent breakthroughs in the intersection of graph theory and complexity theory concerning the Alon-Saks-Seymour problem [HS12, Ama14, Göö15, SA15, BDHT17, GPW18, AHHM22, BBDG+22].

**Remark 1** (Connection between Universal Rates and Partial Concept Classes)**.** *There is an interesting connection between universal rates and partial concepts: in the universal learning setting, the first step of the approach is to use the data to simulate some Gale-Stewart games and show that, with high probability, most of them will have "converged", i.e., the function that corresponds to the learning strategy of the learner will be correct. In turn, this defines some data-dependent constraints. For instance, assume that $g$ is a successful NL pattern avoidance function, i.e., a function which takes as input any $\ell$ points $x_1, \ldots, x_\ell$ and any two (everywhere different) mappings $s^{(0)}, s^{(1)}$ from points to labels and returns an invalid pattern, i.e., a binary pattern $y$ of length $\ell$ that is not compatible with the definition of the Natarajan dimension (i.e., there is no function $h \in \mathcal{H}$ such that if $y_i = 1$ then $h(x_i) = s^{(1)}(x_i)$ and if $y_i = 0$ then $h(x_i) = s^{(0)}(x_i)$, for all $i \in [\ell]$). Then, we can define a partial concept class $\mathcal{H}'$, the set of all functions from $X$ to $\{1, \ldots, k, \star\}$ that satisfy the constraint of this pattern avoidance function, and it has two important properties: its Natarajan dimension is bounded by $\ell$ and a learning algorithm for $\mathcal{H}'$ also learns $\mathcal{H}$. Hence, understanding the learnability of partial concept classes is an essential step in coming up with more natural learning strategies in the universal learning setting. Importantly, in both of these settings, ERM is not an optimal learning algorithm and the one-inclusion graph predictor is an essential part in deriving results in both theories.*

## 2 Technical Overview & Proof Sketches

In this section we briefly discuss the technical details and provide proof sketches of our main results.

**Technical Overview of Universal Multiclass Learning.** In the universal multiclass setting, we provide three lower bounds and two algorithms in order to get the desired trichotomy of Theorem 1. The first lower bound states that no class $\mathcal{H}$ is learnable at rate faster than exponential (see Proposition 1). Our first essential contribution is that any $\mathcal{H} \subseteq [k]^{\mathcal{X}}$ is learnable at (optimal) rate $e^{-n}$ if and only it has no infinite multiclass Littlestone tree. For this task, we provide Algorithm 1 that achieves this rate. Our approach uses tools from infinite game theory (Gale-Stewart games) and set theory (ordinals) in order to show that there exists an online learning algorithm that makes only a finite number of mistakes. We denote this key subroutine with $g_t$ in Algorithm 1. This subroutine $g_t$ is obtained by using the winning strategy of the game, which given the correct label and an incorrect one outputs the correct one, to play a tournament between all the labels. Our final algorithm runs the above subroutine $g_t$ on multiple batches using data-splitting techniques and then takes a majority vote; the intuition behind this step is that the majority vote of various executions of our algorithm will be much better concentrated than a single execution and will achieve the desired exponential rate. The main technical challenge is to construct $g_t$.

---

**Algorithm 1** Exponential Rates Algorithm for Universal Multiclass Learning

---

Exponential Rates
Let $g_t : \mathcal{X} \to Y$ be an *eventually correct* labeling function (Theorem 5).
Let $(X_1, Y_1, \ldots, X_n, Y_n)$ be the training set.
Estimate $\hat{t}_n$ such that $\mathbf{Pr}[\mathrm{er}(g_{\hat{t}_n})] \leq 3/8$.
Break the training set into $N = n/\hat{t}_n$ batches.
Create $N$ copies of $g$: $g^1, \ldots, g^N$ where the $i$-th copy is trained on the $i$-th batch.
To predict the label of some $x \in \mathcal{X}$, take the majority vote over all $g_{\hat{t}_n}^i$.

Exponential GS Game
For any $t \in \mathbb{N}$ :
  $\mathrm{P}_A$ picks $\kappa_t = (\xi_t, y_t^{(0)}, y_t^{(1)}) \in \mathcal{X} \times [k] \times [k]$.
  $\mathrm{P}_A$ reveals $\kappa_t$ to the learner $\mathrm{P}_L$.
  $\mathrm{P}_L$ chooses $\eta_t \in \{0, 1\}$.
  $\mathrm{P}_L$ wins the game if for some $t \in \mathbb{N}$

$$\{h \in \mathcal{H} : h(\xi_\ell) = y_\ell^{(\eta_\ell)} \ \forall \ell \in [1..t]\} = \emptyset \,.$$

---

Our approach to construct the eventually correct function $g_t$ passes through the adversarial online learning setting. As a first step, we introduce the standard multiclass online learning game [DSBDSS11] between an adversary and a learner. In this game, the adversary picks a point $x_t \in \mathcal{X}$ and the learner guesses its true label $y_t \in [k]$. In the standard mistake bound model [Lit88, DSBDSS11], the learner's goal is to achieve a *uniformly* bounded number of mistakes (and this is associated with the multiclass Littlestone dimension and the Standard Optimal Algorithm). We extend this model to the case where we can guarantee a **finite** number of mistakes for each realizable sequence, but without an a priori bound on the number of mistakes, i.e., this number is not *uniformly*

*bounded*. This is the motivation behind Definition 2. We prove that when $\mathcal{H}$ does not have an infinite multiclass Littlestone tree, there exists an online learning algorithm for this setting which makes finitely many mistakes. This is exactly the eventually correct function $g_t$ of Algorithm 1. To be precise, the function $g_t$ corresponds to the tournament that we run using the *winning strategy* after round $t$ of the learning player in the above game.

**Theorem 5** (Informal, see Theorem 9). *For any $\mathcal{H} \subseteq [k]^{\mathcal{X}}$, if $\mathcal{H}$ does not have an infinite multiclass Littlestone tree, there is a strategy $g_t, t \in \mathbb{N}$, for the learner that makes only finitely many mistakes. Otherwise, the adversary has a winning strategy.*

To prove this result, we invoke the theory of Gale-Stewart games. We introduce a novel two-player game, the `Exponential GS Game` outlined in Algorithm 1. The structure of this game looks like the standard multiclass online learning game but has some evident differences; the adversary not only reveals a point $\xi_t$ but also two colors for it. Then, the learner should choose between these two. The structure of this game (while unaccustomed) is crucial and generalizes the game of [BHM+21]. The learner wins if the class of consistent hypotheses $\mathcal{H}_{\kappa_1, \eta_1, \ldots, \kappa_t, \eta_t} = \{h \in \mathcal{H} : h(\xi_\ell) = y_\ell^{(\eta_\ell)} \ \forall \ell \in [1..t]\}$ becomes empty after a finite number of rounds. If the game continues indefinitely, the adversary wins. The intuition behind the definition of the class in Algorithm 1 is that the adversary wins as long as there is always a hypothesis in $\mathcal{H}$ that is consistent with the examples (this is in parallel with the definition of an infinite path in the multiclass Littlestone tree). Using tools from Gale-Stewart games (Appendix B.6), we manage to show that the learning player $P_L$ has a winning strategy if and only if $\mathcal{H}$ does not have an infinite multiclass Littlestone tree. This winning strategy is in fact the *ordinal Standard Optimal Algorithm*. Recall that it is possible to have unbounded multiclass Littlestone dimension while not having an infinite multiclass Littlestone tree.

In order to quantify this intermediate state (between uniformly bounded and truly infinite), we invoke the theory of ordinal numbers and introduce the *ordinal multiclass Littlestone dimension*, which quantifies "how infinite" the multiclass Littlestone dimension is. Hence, the learner's strategy is to play according to the SOA where the standard Littlstone dimension is replaced by the ordinal one (see Appendix C.1.1). We note that the analysis of the above game constitutes an important technical contribution in the exponential rates setting. More to that, we believe that the link between the multiclass SOA and ordinals' theory is an interesting conceptual step. For further details concerning the exponential rates, we refer to Appendix C.1. Our next result (Theorem 11) is a lower bound indicating a sharp transition in the learning rate: A class $\mathcal{H} \subseteq [k]^{\mathcal{X}}$ that has an infinite multiclass Littlestone tree is learnable no faster than $1/n$. Its proof uses the probabilistic method and shows that for any learning algorithm $\widehat{h}_n$, there exists a realizable distribution $P$ over $\mathcal{X} \times [k]$ such that $\mathbf{E}[\mathrm{er}(\widehat{h}_n)] \geq \Omega(1/n)$ for infinitely many $n$, when $\mathcal{H}$ has an infinite multiclass Littlestone tree.

We can now move to the linear rates setting where the situation is significantly more involved technically. In this setting, we show that any $\mathcal{H} \subseteq [k]^{\mathcal{X}}$ is learnable at rate $1/n$ if and only if it has no infinite NL tree. The structure of an NL tree indicates that the notion of the Natarajan dimension (which characterizes learnability in the uniform setting) is invoked in order to control the complexity/expressivity of our concept class. Compared to the exponential rates setting, we shift our goal from hoping for a finite number of mistakes to looking for a control over the *model complexity*. This model complexity is quantified by the notion of an *NL pattern in the data* (see Definition 13). Conceptually, the design of the algorithm for the linear rates follows a similar path as in the exponential case; we first develop an infinite game which makes use of the structure of the NL trees. This `Linear GS Game` is also original and can be found in Algorithm 2. The precise structure of the game is quite important for our results and various modifications of it seem to fail.

In the `Linear GS Game`, the adversary picks $t$ points and two colorings for these points which are everywhere different. Then the learner responds with an *NL pattern* $\eta_t \in \{0, 1\}^t$ with the goal that there is *no* $h \in \mathcal{H}$ that is $N$-consistent with the adversary's input. Hence, the learner aims to find *forbidden NL patterns in the data*. The finiteness of the NL tree implies the existence of a winning strategy for the learner in the game and, hence, an algorithm (which one can construct) for learning to rule out NL patterns. Then, we show how to simulate this game using any $\mathcal{H}$-realizable sequence and utilize the learner's strategy to, eventually, find forbidden NL patterns in the data. The simulation of the game is another novel part for the linear rates (see Figure 7). Intuitively, there exists some finite number $m$, which depends on the data sequence, such that for any collection of $m + 1$ points, there exists some invalid NL pattern. The definition of the NL patterns then indicates that we cannot $N$-shatter any collection of $m + 1$ points and hence we can work, in some sense, with

---

**Algorithm 2** Linear Rates Algorithm for Universal Multiclass Learning

---

`Linear Rates`

Let $g_t : \mathcal{X}^t \times [k]^t \times [k]^t \to \{0,1\}^t$ be an *eventually correct* NL-pattern avoidance function.
Let $(X_1, Y_1, \ldots, X_n, Y_n)$ be the training set.
Estimate $\hat{t}_n$ such that $\mathbf{Pr}[\mathrm{er}(g_{\hat{t}_n})] \leq 3/8$.

Break the training set into $N = n/\hat{t}_n$ batches.
Create $N$ copies of $g$: $g^1, \ldots, g^N$, where the $i$-th copy is trained on the $i$-th batch.
Create $N$ copies of the one-inclusion graph predictor, each copy is equipped with $g^i_{\hat{t}_n}$.
To predict the label of $x$, take the majority vote over all the one-inclusion graph predictors.

`Linear GS Game`

For any $t \in \mathbb{N}$ :

$\quad$ $\mathrm{P}_A$ picks a point $\xi_t = \left( \xi_t^{(0)}, \ldots, \xi_t^{(t-1)}, s_t^{(0)}, s_t^{(1)} \right)$

$\quad$ where (i) $\xi_t \in \mathcal{X}^t \times [k]^t \times [k]^t$ and

$\quad$ (ii) $s_t^{(0)}, s_t^{(1)}$ s.t. $s_t^{(0)}(\xi_t^{(i)}) \neq s_t^{(1)}(\xi_t^{(i)}) \ \forall i$.

$\quad$ $\mathrm{P}_A$ reveals $\xi_t$ to the learner $\mathrm{P}_L$.

$\quad$ $\mathrm{P}_L$ chooses a pattern $\eta_t \in \{0,1\}^t$.

$\mathrm{P}_L$ wins the game if for some $t \in \mathbb{N}$

$$\left\{ h \in \mathcal{H} \text{ s.t. } \begin{array}{l} h(\xi_z^{(i)}) = s_z^{(\eta_z^{(i)})}(\xi_z^{(i)}) \\ \text{for } 0 \leq i < z, z \in [1..t] \end{array} \right\} = \emptyset \, .$$

---

a class whose Natarajan dimension is $m$. We can then use the one-inclusion hypergraph algorithm [HLW94, RBR09, DSS14, BCD$^+$22] (see Appendix B.2) to get a good predictor for the data. Again a single execution of the above strategy is not sufficient, we have to use data-splitting and aggregate our collection of predictors using the majority vote. Algorithm 2 achieves an optimal rate of $1/n$. Finally, we prove that a class with an infinite NL tree requires arbitrarily slow rates.

**Technical Overview of Partial Multiclass Learning.** In the partial multiclass setting with a bounded number of labels, we first characterize learnability in terms of the Natarajan dimension. For the proof of Theorem 3, it suffices to show the following more fine-grained Theorem 6.

**Theorem 6.** *For any partial class $\mathcal{H} \subseteq \{0,1,...,k,\star\}^\mathcal{X}$ with $\mathrm{Ndim}(\mathcal{H}) \leq \infty$, the sample complexity of PAC learning $\mathcal{H}$ satisfies $C_1 \cdot \frac{\mathrm{Ndim}(\mathcal{H}) + \log(1/\delta)}{\epsilon} \leq \mathcal{M}(\epsilon, \delta) \leq C_2 \cdot \frac{\mathrm{Ndim}(\mathcal{H}) \log(k) \log(1/\delta)}{\epsilon}$, for some constants $C_1, C_2$. In particular, if $\mathrm{Ndim}(\mathcal{H}) = \infty$, then $\mathcal{H}$ is not PAC learnable.*

For the upper bound, we have to employ the one-inclusion hypergraph algorithm (see Appendix B.2). Following the methodology of [AHHM22], we extend its guarantees (which hold for total concept classes) to the partial setting. This algorithm guarantees an expected error which can be boosted to a high probability result using standard concentration and boosting techniques. To show that the partial concept class $\mathcal{H} \subseteq \{0,1,...,k,\star\}$ is not learnable if it has infinite Natarajan dimension, we reduce the problem to classification with total concepts and invoke the existing (standard) lower bound. The main take-away from Theorem 6 is that the algorithmic landscape of partial concept classes is provably elusive, as already indicated by the seminal work of [AHHM22]. To this end, we provide a second result that shows when one can apply the well-understood ERM principle (which is valid when the number of labels is bounded) with partial concepts. For details, we refer to Proposition 2. To conclude, we address the task of disambiguation [AKM19, AHHM22] of partial concepts (see Definition 15). Our proof of Theorem 4 relies on an interesting observation: the seminal work of [AHHM22] showed that the combinatorial variant of the SSP lemma [Sau72] does not hold in this setting. This lemma has a second variant that uses the growth function [SSBD14] instead of the size of the class. We show that a natural extension of this variant for partial classes is still correct (see Lemma 13). Using this tool and techniques from [AHHM22], we obtain our impossibility result.

## Acknowledgments and Disclosure of Funding

Alkis Kalavasis is supported by the Hellenic Foundation for Research and Innovation (H.F.R.I.), Project BALSAM, HFRIFM17-1424. Grigoris Velegkas is supported by NSF (IIS-1845032), an Onassis Foundation PhD Fellowship and a Bodossaki Foundation PhD Fellowship. Amin Karbasi acknowledges funding in direct support of this work from NSF (IIS-1845032), ONR (N00014-19-1-2406), and the AI Institute for Learning-Enabled Optimization at Scale (TILOS). The authors would like to thank the anonymous reviewers for helpful comments and suggestions.

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
