# A Omitted Details from the Main Body

## A.1 Previous Work

Classification with multiple labels is extensively studied and, for the setting with bounded $k$, PAC learnability is well-understood [Nat89, BCHL95, SSBD14, DSBDSS11]. The works of [VC71, BEHW89] provide a fundamental dichotomy/equivalence between the finiteness of the VC dimension and binary classification (PAC learnability). [NT88] and [Nat89] extend the PAC framework to the multiclass setting by providing the notions of the Graph and the Natarajan dimension. When the number of labels $k$ is a finite constant, then these two dimensions both characterize PAC learnability since $\mathrm{Ndim}(\mathcal{H}) \leq \mathrm{Gdim}(\mathcal{H}) \leq \mathrm{Ndim}(\mathcal{H}) \cdot O(\log(k))$ [DSBDSS11]. Afterwards, [BCHL95] and [HL95] provide a combinatorial abstraction which captures as special cases e.g., the Graph and Natarajan dimensions and Pollard's pseudo-dimension. In this general setting, [BCHL95] identify the notion of $\Psi$-*distinguishers* that characterize PAC learnability when $k$ is bounded. More to that, uniform convergence still applies when the number of classes is bounded and, so, PAC learning provides the ERM principle for algorithm design. We remark the situation gets much more complicated when the number of labels is not bounded [DSS14, BCD$^+$22, RBR09, DSBDSS11]. For instance, [DSS14] show that the ERM principle does not apply in this case. In a recent breakthrough, [BCD$^+$22] show that the DS dimension captures learnability and the Natarajan dimension provably fails to achieve this.

## A.2 Assumption on Cardinality of $\mathcal{H}$

We briefly discuss why the assumption that $|\mathcal{H}| > k + 2$ comes without loss of generality. If $\mathcal{H}$ contains either a single hypothesis or contains some hypotheses which have no conflict, i.e., $h(x) \neq h'(x)$ everywhere, then $\mathbf{Pr}_{(x,y)\sim P}[\widehat{h}_n(x) \neq y] = 0$ is trivially achievable for all $n$. Now, if $|\mathcal{H}| \leq k + 2$ but does not fall in the above two cases, then $\mathcal{H}$ is learnable at an optimal exponential rate. To see this, let $\epsilon$ be the minimal error among all hypotheses $h \in \mathcal{H}$ with non-zero error. The probability that there exists a hypothesis with error $\epsilon$ that makes no mistakes in the $n$ training data is at most $|\mathcal{H}|(1 - \epsilon)^n$ using the union bound. Thus a learning algorithm that outputs any classifier $\widehat{h}_n \in \mathcal{H}$ that correctly classifies the training set satisfies $\mathbf{E}[\mathbf{Pr}_{(x,y)\sim P}[\widehat{h}_n(x) \neq y]] \leq C \exp(-cn)$ with $C = C(\mathcal{H}, P)$ and $c = c(\mathcal{H}, P)$. Moreover, this is optimal due to Proposition 1.

## A.3 Future Directions

We deal with the settings of universal learning and partial concept classes, two fundamental questions [BHM$^+$21, AHHM22] which are witnessed in real-life applications, nevertheless the classical theory fails to explain. Our results raise various interesting questions for future work (apart from Open Question 1). First, it would be interesting to extend our results to the agnostic setting. Second, for the universal setting, we believe it is an important next step to shed light towards multiclass learning with unbounded labels (whose uniform learnability was recently characterized by [BCD$^+$22]). Moreover, for the partial concepts setting, the work of [AHHM22] leaves numerous fascinating open questions for the binary setting that can be asked in the multiclass setting too. In general, our work along with its seminal binary counterparts [BHM$^+$21, AHHM22] shows that the algorithmic landscape occuring in practice is quite diverse and the ERM principle is provably insufficient. It is important to come up with principled algorithmic strategies that bring theory closer to practice.

# B Preliminaries

In this section we discuss more extensively the Preliminaries from Section 1.4.

## B.1 Complexity Measures

We first state the classical definition of the Littlestone-dimension [Lit88] that characterizes learnability in the online setting.

**Definition 5** (Littlestone dimension). *Consider a complete binary tree $T$ of depth $d + 1$ whose internal nodes are labeled by points in $\mathcal{X}$ and edges by $\{0, 1\}$, when they connect the parent to the*

*right, left child, respectively. We say that $\mathcal{H} \subseteq \{0,1\}^{\mathcal{X}}$ Littlestone-shatters $T$ if for every root-to-leaf path $x_1, y_1, x_2, y_2, \ldots, x_d, y_d, x_{d+1}$ there exists some $h \in \mathcal{H}$ such that $h(x_i) = y_i, 1 \le i \le d$. The Littlestone dimension is denoted by $\mathrm{Ldim}(\mathcal{H})$ is defined to be the largest $d$ such that $\mathcal{H}$ Littlestone-shatters such a binary tree of depth $d+1$. If this happens for every $d \in \mathbb{N}$ we say that $\mathrm{Ldim}(\mathcal{H}) = \infty$.*

The above definition naturally extends to multiple labels [DSBDSS11]; we denote the multiclass Littlestone dimension by $\mathrm{Ldim}_k(\cdot)$. We next recall the well-known notion of VC dimension that characterizes PAC-learnability of binary concept classes [VC71].

**Definition 6** (VC dimension). *We say that $\mathcal{H} \subseteq \{0,1\}^{\mathcal{X}}$ VC-shatters a sequence $\{x_1, \ldots, x_n\} \in \mathcal{X}^n$ if $\{h(x_1), \ldots, h(x_n) : h \in \mathcal{H}\} = \{0,1\}^n$. The VC-dimension of $S$ is denoted by $\mathrm{VCdim}(\mathcal{H})$ and is defined to be the largest $d$ such that $\mathcal{H}$ VC-shatters some sequence of length $d$. If this happens for every $d \in \mathbb{N}$ we say that $\mathrm{VCdim}(\mathcal{H}) = \infty$.*

When moving to $\mathcal{H} \subseteq \{0, \ldots, k\}^{\mathcal{X}}, k > 1$, there are many different extensions of the VC-dimension that have been considered in the literature. We recall the definition of two of them that are important in our work, the Natarajan dimension [Nat89] and the Graph dimension [NT88, Nat89].

**Definition 7** (Natarajan dimension). *We say that $\mathcal{H} \subseteq \{0, \ldots, k\}^{\mathcal{X}}$ N-shatters a sequence $\{x_1, \ldots, x_n\} \in \mathcal{X}^n$ if there exist two colorings $s^{(0)}, s^{(1)}$ such that $s^{(0)}(x_i) \neq s^{(1)}(x_i), \forall i \in [n]$, and $\forall I \subseteq [n]$ there exists some $h \in \mathcal{H}$ with $h(x_i) = s^{(0)}(x_i), i \in I$ and $h(x_i) = s^{(1)}(x_i), i \in [n] \setminus I$. The Natarajan dimension is denoted by $\mathrm{Ndim}(\mathcal{H})$ and is defined to be the largest $d$ for which $\mathcal{H}$ N-shatters a sequence of length $d$. If this happens for every $d \in \mathbb{N}$ we say that $\mathrm{Ndim}(\mathcal{H}) = \infty$.*

**Definition 8** (Graph dimension). *We say that $\mathcal{H} \subseteq \{0, \ldots, k\}^{\mathcal{X}}$ G-shatters a sequence $\{x_1, \ldots, x_n\} \in \mathcal{X}^n$ if there exists a coloring $s^{(0)}$ such that $\forall I \subseteq [n]$ there exists some $h \in \mathcal{H}$ with $h(x_i) = s^{(0)}(x_i), i \in I$ and $h(x_i) \neq s^{(0)}(x_i), i \in [n] \setminus I$. The Graph dimension is denoted by $\mathrm{Gdim}(\mathcal{H})$ and is defined to be the largest $d$ for which $\mathcal{H}$ G-shatters a sequence of length $d$. If this happens for every $d \in \mathbb{N}$ we say that $\mathrm{Gdim}(\mathcal{H}) = \infty$.*

Notice that if $k = 2$ the definitions of the Natarajan dimension and the Graph dimension are equivalent to the definition of the VC-dimension.

**Remark 2.** *We underline that the above definitions can be naturally extended to the partial concepts setting. For instance, we say that $\mathcal{H} \subseteq \{0, 1, \star\}^{\mathcal{X}}$ VC-shatters a set $S$ if every **binary** pattern is realized by some $h \in \mathcal{H}$. For further details, see [AHHM22].*

### B.2 One-Inclusion Hypergraph Algorithm

We next review a fundamental result which is a crucial ingredient in the design of our algorithms, namely the one-inclusion hypergraph algorithm $\mathbb{A}_{\mathcal{H}}$ for the class $\mathcal{H} \subseteq [k]^{\mathcal{X}}$ [HLW94, RBR09, DSS14, BCD$^+$22]. This algorithm gets as input a training set $(x_1, y_1), ..., (x_n, y_n)$ realizable by $\mathcal{H}$ and an additional example $x$. The goal is to predict the label of $x$. In this sense, the one-inclusion graph constitutes a transductive model in machine learning. The idea is to construct the one-inclusion (hyper)graph of $\mathcal{H}|_{x_1, ..., x_n, x} \subseteq [k]^{n+1}$. The nodes of this graph are the equivalence classes of $\mathcal{H}$ induced by the examples $x_1, ..., x_n, x$. For the binary classification case, two equivalence classes are connected with an edge if the nodes differ by exactly one element $x$ of the $n + 1$ points and $x$ appears only once in $(x_1, ..., x_n, x)$. For the case $k > 1$, the hyperedge set is is generalized accordingly. Having created the one-inclusion graph, the goal is to orient the edges; the crucial property is that good orientations of this graph yield low error learning algorithms. Here, an orientation is good if the maximum out-degree of the graph is small. Intuitively, if the maximum out-degree of any node is $M$, then this can yield a prediction for the label $x$ with $n + 1$ training samples with permutation mistake bound at most $M/(n + 1)$.

**Lemma 1** (One-Inclusion Hypergraph Algorithm (See Lemma 17 of [BCD$^+$22])). *Let $\mathcal{X} = [n]$ with $n \in \mathbb{N}$, $k$ be positive constant and $\mathcal{H} \subseteq [k]^n$ be a class with Natarajan dimension $\mathrm{Ndim}(\mathcal{H}) < \infty$. There exists an algorithm $\mathbb{A} : 2^{[k]^X} \times (X \times [k])^{n-1} \times X$ such that*

$$\frac{1}{n!} \sum_{\sigma \sim \mathcal{U}(\mathbb{S}_n)} [\mathbb{A}(\mathcal{H}, \sigma(1), h(\sigma(1)), ..., \sigma(n-1), h(\sigma(n-1)), \sigma(n)) \neq h(\sigma(n))] \le \frac{\mathrm{Ndim}(\mathcal{H}) \log(k)}{n},$$

*for any $h \in \mathcal{H}$.*

### B.3 Ordinals

The discussion of ordinals is borrowed from [BHM$^+$21]. For a thorough treatment of the subject, the interested reader is referred to [HJ99, Sie58].

We consider some set $S$. A well ordering of $S$ is defined to be any linear ordering $<$ so that every non-empty subset of $S$ contains a least element. For example, the set of natural numbers $\mathbb{N}$ along with the usual ordering is well-ordered, whereas $[0,1]$ is not (take, e.g., $S = (0,1)$, then it is clear that there is not a least element in $S$.)

We say that two-well ordered sets are *isomorphic* if there exists a bijection between the two which preserves the ordering. There is a canonical way to construct "equivalence classes" of well-ordered sets, called *ordinals*, so that every well-ordered set is isomorphic to exactly one such ordinal. In that sense, ordinals uniquely encode well-ordered sets in the same way as cardinals uniquely encode sets. We denote the class of all ordinals by ORD.

An important property of ordinals is that every pair of well-ordered sets is either isomorphic, or one of them is isomorphic to an *initial segment* of the other. This fact induces an *ordering* over the ordinals. To be more precise, we say that for two ordinals $\alpha, \beta \in \text{ORD}$ we have that $\alpha < \beta$ if $\alpha$ is isomorphic to an initial segment of $\beta$. The defining property of ordinals is that every ordinal $\beta$ is isomorphic to the set of ordinals that precede it, i.e., $\{\alpha : \alpha < \beta\}$. Moreover, the ordering $<$ is itself a well-ordering over the ordinals. This is because every non-empty set of ordinals contains a least element and it has a least upper bound.

Given the above discussion, we can see that ordinals provide a set-theoretic extension of natural numbers. This is because every ordinal $\beta$ has a successor $\beta + 1$, which is the smallest ordinal that is larger that $\beta$. Thus, we can create a list of all ordinals: the first elements in the least are $0, 1, 2, \ldots$, which are all the finite ordinals where we identify each number $k$ with the well-ordered set $\{0, \ldots, k-1\}$. We let the "smallest" infinite ordinal be $\omega$. This can be identified with the family of all natural numbers along with their natural ordering. The way to count past infinity is the following: we write $0, 1, 2 \ldots, \omega, \omega + 1, \omega + 2, \ldots, \omega + \omega, \ldots$, and we denote the smallest uncountable ordinal with $\omega_1$.

A concept that is defined by ordinals and is useful for proving the guarantees of our algorithms in the Gale-Stewart games is that of *transfinite recursion.* Roughly speaking, this principle states that if we have a "recipe", which is given sets of "objects" $O_\alpha$, indexed by all ordinals $\alpha < \beta$, defines a new set of "objects" $O_\beta$ and has access to a "base set" $\{O_\alpha : \alpha < \alpha_0\}$, then $O_\beta$ is uniquely defined for all ordinals $\beta$. To give an example, this concept helps us define addition $\gamma + \beta$ between two ordinals. We set $\gamma + 0 = \gamma$ and $\gamma + 1$ to be the successor of $\gamma$. We continue inductively and define for any $\beta$ the addition $\gamma + \beta = \sup\{(\gamma + \alpha) + 1 : \alpha < \beta\}$. Following this principle we can define an arithmetic in the ordinals.

### B.4 Well-founded Relations and Ranks

We continue the above discussion by extending the notion of a well-ordering to a more general type of orders, and we introduce the notion of *rank,* which is important for the derivation of the winning strategies in the Gale-Stewart games. We follow the presentation from [BHM$^+$21]. The classical reference for this topic is [Kec12].

We define a *relation* $\prec$ on a set $S$ by an arbitrary subset $R_\prec \subseteq S \times S$ and we let $x \prec y$ if and only if $(x, y) \in R_\prec$. An element $x$ of $(S, \prec)$ is called *minimal* if there is no $x' \in S$ with $x' \prec x$. The relation is called *well-founded* if every non-empty subset of $S$ has a minimal element.

We associate every well-founded relation on $S$ with a rank function $\rho_\prec : S \to \text{ORD}$, which is defined by the following transfinite recursion: we let $\rho_\prec(x) = 0$ if $x$ is the minimal element of $S$ and we let $\rho_\prec(x) = \sup\{\rho_\prec(y) + 1 : y \prec x\}$. Intuitively, the rank of some element $x$ quantifies how far away it is from the minimal element. The following property of rank justifies the name "transfinite recursion".

**Remark 3.** *Any element $x \in S$ has a well defined rank. This is because the transfinite recursion defines $\rho_\prec(x)$ as soon as all $\rho_\prec(y)$ is defined, for all $y \prec x$. Indeed, if $\rho(x)$ is undefined it means that $\rho_\prec(x')$ is also undefined, for some $x' \prec x$. Repeating this for $x'$ constructs an infinitely decreasing*

*chain of elements in S, which contradicts the fact that $\prec$ is well-founded, since an infinitely decreasing chain does not contain a minimal element.*

We give some examples that help develop some intuition about the notion of the rank of a well-founded relation. Even though a well-founded relation does not admit an infinitely decreasing chain, it can contain finite chains of arbitrary length. Essentially, the rank of some element $\rho_{\prec}(x)$ measures the length of a decreasing chain starting from $x$.

**Example 1.** *As a warmup, consider the case where $\rho_{\prec}(x) = k$, where $k$ is a finite ordinal. Then, there exists some $x_1 \prec x$ such that $\rho_{\prec}(x_1) = k - 1$. Continuing in the same manner, we can see that we can create a decreasing chain of length $k + 1$.*

**Example 2.** *Moving on, assume that $\rho_{\prec}(x) = \omega$. Recall that $\omega$ is the smallest infinite ordinal. This means that any $y \prec x$ has rank $\rho_{\prec}(y) = k$, for some finite number $k$. Hence, starting from $x$ we can create a decreasing chain of arbitrary length, but this length is determined when we fix the first element of the chain.*

**Example 3.** *Now assume that $\rho_{\prec}(x) = \omega + k$, for some finite number $k$. Then, we can create $x \succ x_1 \succ \ldots \succ x_k$, with $\rho_{\prec}(x_k) = \omega$. Thus, the length of the chain is determined after we pick $x_{k+1}$.*

**Example 4.** *A slightly more involved example is the case where $\rho_{\prec}(x) = \omega + \omega$. As a first element in the chain, we pick some $x_1 \prec x$ with $\rho_{\prec}(x_1) = \omega + k$, for some finite $k$. Then, we continue as in the previous example. So in this case, the length of the chain is determined by two choices.*

## B.5 Measurability

This short exposition is mainly from [BHM+21]. A **Polish space** is a separable topological space that can be metrized by a complete metric. For example, the $n$-dimensional Euclidean space, any compact metric space, any separable Banach space are Polish spaces.

**Definition 9** (See [BHM+21]). *A concept class $\mathcal{H} \subseteq [k]^{\mathcal{X}}$ on a Polish space $\mathcal{X}$ is said to be **measurable** if there is a Polish space $\Theta$ and Borel-measurable map $h : \Theta \times \mathcal{X} \to [k]$ so that $\mathcal{H} = \{h(\theta, \cdot) : \theta \in \Theta\}$.*

Roughly, a subset $B$ of a Polish space $\mathcal{X}$ is universally measurable if it is measurable with respect to every complete probability measure on $\mathcal{X}$.

**Definition 10** (Universally Measurable). *Let $\mathcal{F}$ be the Borel $\sigma$-field on a Polish space $\mathcal{X}$. For any probability measure $\mu$, denote by $\mathcal{F}_\mu$ the completion of $\mathcal{F}$ with respect to $\mu$, that is, the collection of all subsets of $\mathcal{X}$ that differ from a Borel set at most on a set of zero probability. A set $B \subseteq \mathcal{X}$ is called **universally measurable** if $B \in \mathcal{F}_\mu$ for every probability measure $\mu$. Similarly, a function $f : \mathcal{X} \to \mathcal{Y}$ is called **universally measurable** if $f^{-1}(B)$ is universally measurable for any universally measurable set $B$.*

Let $\mathcal{X}, \mathcal{Y}$ be Polish spaces, and let $f : \mathcal{X} \to \mathcal{Y}$ be a continuous function. It holds that $f$ is Borel measurable, that is, $f^{-1}(B)$ is a Borel subset of $\mathcal{X}$ for any Borel subset $B$ of $\mathcal{Y}$. A subset $B \subseteq \mathcal{X}$ of a Polish space is called **analytic** if it is the image of some Polish space under a continuous map. The complement of an analytic set is called **coanalytic**. A set is Borel if and only if it is both analytic and coanalytic. The following is a consequence of Choquet's Capacitability Theorem.

**Fact 1.** *Every analytic (or coanalytic) set is universally measurable.*

An important property of analytic sets is that they help us establish measurability of uncountable unions of measurable sets.

We now state a very important result regarding well-founded relations on Polish spaces. We let $\mathcal{X}$ be a Polish space and $\prec$ a well-found relation on $\mathcal{X}$. We say that $\prec$ is analytic if $R_{\prec} \subseteq \mathcal{X} \times \mathcal{X}$ is an analytic set. The following theorem, known as Kunen-Martin [Kec12, Del77], bounds the rank function of such a relation.

**Theorem 7.** *Let $\prec$ be an analytic well-founded relation on a Polish space $\mathcal{X}$. Then, its rank function satisfies $\sup_{x \in \mathcal{X}} \rho_{\prec}(x) \leq \omega_1$.*

### B.6 Gale-Stewart Games

We add a short discussion about GS games from [BHM$^+$21]. We refer to their work for further details and pointers. Fix sequences of sets $\mathcal{X}_t, \mathcal{Y}_t$ for $t \geq 1$. We consider infinite games between two players: in each round $t \geq 1$, the first player $P_A$ selects an element $x_t \in \mathcal{X}_t$, and then player $P_L$ selects an element $y_t \in \mathcal{Y}_t$. The rules of the game are determined by specifying a set $\mathcal{W} \subseteq \prod_{t \geq 1}(\mathcal{X}_t \times \mathcal{Y}_t)$ of winning sequences for $P_L$. That is, after an infinite sequence of consecutive plays $x_1, y_1, x_2, y_2, \ldots$, we say that $P_L$ wins if $(x_1, y_1, x_2, y_2, \ldots) \in \mathcal{W}$; otherwise, $P_A$ is declared the winner of the game.

A **strategy** is a rule used by a given player to determine the next move given the current position of the game. A strategy for $P_A$ is a sequence of functions $f_t : \prod_{s<t}(\mathcal{X}_s \times \mathcal{Y}_s) \to \mathcal{X}_t$ for $t \geq 1$, so that $P_A$ plays $x_t = f_t(x_1, y_1, \ldots, x_{t-1}, y_{t-1})$ in round $t$. Similarly, a strategy for $P_L$ is a sequence of $g_t : \prod_{s<t}(\mathcal{X}_s \times \mathcal{Y}_s) \times \mathcal{X}_t \to \mathcal{Y}_t$ for $t \geq 1$, so that $P_L$ plays $y_t = g_t(x_1, y_1, \ldots, x_{t-1}, y_{t-1}, x_t)$ in round $t$. A strategy for $P_A$ is called **winning** if playing that strategy always makes $P_A$ win the game regardless of what $P_L$ plays; a winning strategy for $P_L$ is defined analogously. The crucial question follows:

*When do winning strategies exist in infinite two-player games?*

The simplest assumption was introduced by [GS53]: we call $\mathcal{W}$ **finitely decidable** if for every sequence of plays $(x_1, y_1, x_2, y_2, \ldots) \in \mathcal{W}$, there exists $n < \infty$ so that

$$(x_1, y_1, \ldots, x_n, y_n, x'_{n+1}, y'_{n+1}, x'_{n+2}, y'_{n+2}, \ldots) \in \mathcal{W}$$

for all choices of $x'_{n+1}, y'_{n+1}, x'_{n+2}, y'_{n+2}, \ldots$ In words, that "$\mathcal{W}$ is finitely decidable" means that if $P_L$ wins, then she knows that she won after playing a finite number of rounds. Conversely, in this case $P_A$ wins the game precisely when $P_L$ does not win after any finite number of rounds.

An infinite game whose set $\mathcal{W}$ is finitely decidable is called a **Gale-Stewart game**. The main result behind GS games follows.

**Proposition.** *In any Gale-Stewart game, either $P_A$ or $P_L$ has a winning strategy.*

The above existential result provides no information, however, about the complexity of the winning strategies. In particular, it is completely unclear whether winning strategies can be chosen to be measurable. The next lemma addresses this concern.

**Lemma 2** (Theorem B.1 of [BHM$^+$21])**.** *Let $\{X_t\}_{t \geq 1}$ be Polish spaces and $\{Y_t\}_{t \geq 1}$ be countable sets. Consider a Gale-Stewart game whose set $\mathcal{W} \subseteq \prod_{t \geq 1}(X_t \times Y_t)$ of winning strategies for $P_L$ is finitely decidable and coanalytic. Then there is a universally measurable winning strategy.*

For an extensive exposition on *game values* which connect ordinals with the positions of the game, i.e., the sequences of choices between the two players, we refer to [BHM$^+$21].

## C   Universal Multiclass Learning: The Proof of Theorem 2

In this section, we prove Theorem 2 which then directly gives us Theorem 1 as well. Our text is organized as follows:

- In Appendix C.1, we analyze the exponential rates case for multiclass learning. We introduce the notion of the multiclass Littlestone tree and prove Theorem 8.
    - In order to achieve this, we first analyze the problem from an adversarial online perspective in Appendix C.1.1. The main result in this section is Theorem 9.
    - The above result is in the adversarial setting and hence we have to transform this online algorithm into a statistical one, i.e., we have to move from the adversarial setting to the probabilistic setting. This is done is Appendix C.1.2. We provide the analysis of our final algorithm in Theorem 10.
- In Appendix C.2, we show that a class with infinite multiclass Littlestone tree cannot be learned at a rate faster than linear.
- In Appendix C.3, we introduce the notion of a Natarajan-Littlestone tree and we prove Theorem 12.

- In Appendix C.3.1, we provide the important notion of a Natarajan-Littlestone game which guarantees the existence of an eventually correct algorithm when the class does not have an infinite NL tree.
- In Appendix C.3.2, we introduce the notion of an NL pattern in the data and using the above algorithm as a pattern avoidance function.
- The behavior of the pattern avoidance algorithm in the probabilistic setting is given in Appendix C.3.3.
- The final linear rate algorithm can be found at Appendix C.3.4.

- In Appendix C.4, the final lower bound which is related to arbitrarily slow rates (see Theorem 14).

- In Appendix C.5, we provide the notion of a Graph-Littlestone tree and give a sufficient condition for learning with a linear rate. More to that, we propose Open Question 1.

## C.1 Exponential Rates for Multiclass Learning

For a sequence $\boldsymbol{y} = (y_1, y_2, ...)$, we denote $\boldsymbol{y}_{\leq k} = (y_1, ..., y_k)$. We may also usually identify elements of $\{0, 1\}^d$ with strings or a prefix of a sequence of length $d$. We begin with a formal definition of a crucial combinatorial measure, namely the multiclass Littlestone tree of a class $\mathcal{H}$.

**Definition 11** (Multiclass Littlestone Tree). *A multiclass Littlestone tree for $\mathcal{H} \subseteq \{0, 1, ..., k\}^{\mathcal{X}}$ is a complete binary tree of depth $d \leq \infty$ whose internal nodes are labeled by $\mathcal{X}$, and whose two edges connecting a node to its children are labeled by two different elements in $[k]$, such that every path of length at most $d$ emanating from the root is consistent with a concept $h \in \mathcal{H}$. Typically, a multiclass Littlestone tree is a collection*

$$\bigcup_{0 \leq \ell < d} \left\{ x_u : u \in \{0, 1\}^{\ell} \right\} = \{x_\emptyset\} \cup \{x_0, x_1\} \cup \{x_{00}, x_{01}, x_{10}, x_{11}\} \cup ...$$

*such that for every path $\boldsymbol{y} \in \{0, 1\}^d$ and finite $n < d$, there exists $h \in \mathcal{H}$ so that $h(x_{\boldsymbol{y}_{\leq \ell}}) = s_{\boldsymbol{y}_{\leq \ell+1}}$ for $0 \leq \ell \leq n$, where $s_{\boldsymbol{y}_{\leq \ell+1}}$ is the label of the edge connecting the nodes $x_{\boldsymbol{y}_{\leq \ell}}$ and $x_{\boldsymbol{y}_{\leq \ell+1}}$. We say that $\mathcal{H}$ has an infinite multiclass Littlestone tree if there is a multiclass Littlestone tree for $\mathcal{H}$ of depth $d = \infty$.*

To give some intuition about this construction we state some of its properties. First, it is crucial to note that a class $\mathcal{H}$ with a finite multiclass Littlestone tree can have infinite multiclass Littlestone dimension. In fact, a class has finite multiclass Littlestone dimension if the depth of the tree admits a *uniform* upper bound. However, it may be the case that the class admits an unbounded tree, in the sense that for any finite depth, there exists a tree of that depth; nevertheless the class does not have an infinite tree. Second, a class with a finite multiclass Littlestone tree may contain trees with infinite paths (e.g., a class that contains the constant mapping $h = 1$ can shatter the rightmost path at an infinite depth). Finally, a class with finite multiclass Littlestone tree cannot have a tree with an infinite complete binary subtree, since one could use only this subtree and obtain an infinite tree. Essentially, this tree captures "how infinite" the multiclass Littlestone dimension of $\mathcal{H}$ is: even if there is no uniform bound on the multiclass Littlstone dimension of $\mathcal{H}$, whenever $\mathcal{H}$ does not have an infinite tree we know that for *any* $S = \bigcup_{0 \leq \ell < d} \left\{ x_u : u \in \{0, 1\}^{\ell} \right\}$ there is some $n^*(S) < \infty$, which depends on the sequence $S$, so that the tree can be shattered up to level $n^*(S)$.

The goal of this section is to prove the next result.

**Theorem 8** (Exponential rates). *If $\mathcal{H} \subseteq [k]^{\mathcal{X}}$ does not have an infinite multiclass Littlestone tree, then for any distribution $P$, $\mathcal{H}$ is learnable with an exponential optimal rate.*

In order to prove this result, two steps are required: first, it suffices to show that no class of hypotheses can be learned in a rate faster than exponential (lower bound) and second, we have to show that not having an infinite multiclass Littlestone tree is a sufficient for learning at an exponential rate condition (upper bound). Hence, these two directions tightly characterize universal learnability at exponential rates for multiclass classification.

For the lower bound, the argument is essentially the same as in [BHM+21]. Let us first give some intuition. If the distribution is supported on a finite number of points, e.g., two, there is an exponentially small probability that all the $n$ samples will contain the same point $(x, y)$, so it will

not help the learner distinguish between the functions $h \in \mathcal{H}$ that label this point correctly. For completeness, we present the argument formally.

**Proposition 1** (Exponential Rates (Lower Bound)). *Fix $\mathcal{H} \subseteq \{0, 1, ..., k\}^{\mathcal{X}}$. For any learning algorithm $\widehat{h}_n$, there exists a realizable distribution $P$ over $\mathcal{X} \times \{0, 1, ..., k\}$ such that $\mathbf{E}[\mathrm{er}(\widehat{h}_n)] \geq \Omega(2^{-n})$ for infinitely many $n$. This means that $\mathcal{H}$ is not learnable at rate faster than exponential.*

*Proof.* Recall that $|\mathcal{H}| > k + 2$. Hence, there are $h_0, h_0 \in \mathcal{H}$ and $x, x' \in \mathcal{X}$ such that $h_0(x) = h_1(x) = y$ and $h_0(x') = y_0 \neq h_1(x) = y_1$. We fix some learning algorithm $\widehat{h}_n$ and two distributions $P_0, P_1$ where $P_i \{(x, y)\} = \frac{1}{2}, P_i \{(x, y_i)\} = \frac{1}{2}, i \in \{0, 1\}$. We let $I \sim \mathrm{Bernoulli}(1/2)$ and given I, we let $(X_1, Y_1), (X_2, Y_2), \ldots$ be i.i.d. samples from $P_I$ and $(X_1, Y_1), \ldots, (X_n, Y_n)$ are the training samples for $\widehat{h}_n$ and $(X_{n+1}, Y_{n+1})$ is the test point. Then, we have that

$$\mathbf{E}[\mathbf{Pr}(\widehat{h}_n(X_{n+1}) \neq Y_{n+1} | \{X_t, Y_y\}_{t=1}^n, I)] \geq \frac{1}{2} \mathbf{Pr}(X_1 = \ldots = X_n, X_{n+1} = x') = 2^{-n-2}.$$

We also have that

$$\mathbf{E}[\mathbf{Pr}(\widehat{h}_n(X_{n+1}) \neq Y_{n+1} | \{X_t, Y_y\}_{t=1}^n, I)]$$
$$= \frac{1}{2} \sum_{i \in \{0,1\}} \mathbf{E}[\mathbf{Pr}(\widehat{h}_n(X_{n+1}) \neq Y_{n+1} | \{X_t, Y_y\}_{t=1}^n, I = i) | I = i].$$

Thus, for every $n$, there exists some $i_n\{0, 1\}$ such that for $(X_1, Y_1), \ldots, (X_n, Y_n)$ i.i.d. from $P_{i_n}$ it holds that

$$\mathbf{E}[\mathrm{er}_{P_{i_n}}(\widehat{h}_n)] \geq 2^{-n-2}.$$

Hence, there exists some fixed $i \in \{0, 1\}$ such that $\mathbf{E}[\mathrm{er}_{P_i}(\widehat{h}_n)] \geq 2^{-n-2}$ for infinitely many $n$. $\square$

We continue with the upper bound, i.e., the design of an algorithm that learns $\mathcal{H} \subseteq [k]^{\mathcal{X}}$ at an exponentially fast rate when its multiclass Littlestone tree is not infinite. Our approach consists of two main steps. First, we consider the classical online adversarial setting where a learner has to guess the label of a point that is presented to her by an adversary. We prove that if $\mathcal{H}$ does not have an infinite multiclass Littlestone tree, there is a strategy the learner can employ that makes a *finite* number of mistakes. Then, given such a strategy, we prove that there is an algorithm in the statistical setting which achieves exponential learning rates. The details are outlined in the subsequent sections.

### C.1.1 Viewing Exponential Rates in an Online Setting

In order to design our algorithms, we have to consider the following setting. We introduce the multiclass online learning game (Figure 3) that has been studied extensively (see e.g., [DSBDSS11]). In this game, there are two players, the adversary that chooses features and reveals them to the second player, the learner whose goal is to guess a label for the given example. The learner makes a mistake

---

1. The adversary picks a point $x_t \in \mathcal{X}$.
2. The learner guesses a value $\widehat{y}_t \in [k]$
3. The adversary chooses the value $y_t$ as true label so that $y_t = h(x_t)$ for some $h \in \mathcal{H}$ that is consistent with the previous examples $(x_p, y_p)$ for any $p \leq t$.

---

Figure 3: Realizable Online Setting

in round $t$ whenever the guess $\widehat{y}_t$ differs from the true label $y_t$. The goal of the learner is to minimize her loss and the adversary's intention is to provoke many errors to the learner.

We say that the concept class $\mathcal{H}$ is online learnable if there exists a strategy $\widehat{y}_t = \widehat{y}_t(x_1, y_1, ..., x_{t-1}, y_{t-1}, x_t)$ that makes a mistake only *finitely* many times, regardless of what realizable sequence is presented by the adversary. Notice that compared to the classical online learning setting that asks for a bounded number of mistakes $d$, we settle for a more modest goal. The main result in this setting is the following.

**Theorem 9** (Strategies in the Adversarial Setting). *For any concept class $\mathcal{H} \subseteq [k]^{\mathcal{X}}$, the following dichotomy occurs.*

1. *If $\mathcal{H}$ does not have an infinite multiclass Littlestone tree, then there is a strategy for the learner that makes only finitely many mistakes against any adversary.*

2. *If $\mathcal{H}$ has an infinite multiclass Littlestone tree, then there is a strategy for the adversary that forces any learner to make a mistake in every round.*

Before the formal proof, we provide a proof sketch. The adversary's strategy is clear. Whenever the learner predicts a label, she must choose a different label that will cause the learner to make a mistake in that round; an infinite Littlestone tree is exactly the combinatorial structure the adversary is looking for. On the other hand, since there is no infinite multiclass Littlestone tree, the learner's predictions should lead her to a leaf of the Littlestone tree that is defined by her interaction with the learner. This is established by a variant of the (multiclass) Standard Optimal Algorithm (SOA) [Lit88, DSBDSS11], which works whenever $\mathcal{H}$ has a finite multiclass Littlestone dimension $d$. To gain some intuition, it is instructive to consider the first step of her algorithm. The learner is presented with a point $x_1$ and she picks the label $y_1 = \mathrm{argmax}_{y \in [k]} \mathrm{Ldim}_k(\mathcal{H}_{x_1, y})$. It is easy to see that there is at most one $y$ such that $\mathrm{Ldim}_k(\mathcal{H}_{x_1, y}) = d$; if there were two of them then $\mathrm{Ldim}_k(\mathcal{H}) = d + 1$. Thus, by picking the one that induces the subset of the hypothesis class with the largest Littlestone dimension, the learner either does not make a mistake or gets closer to a leaf of the tree.

In our universal setting, this approach does not work immediately since it may be the case that the multiclass Littlestone tree is not infinite but the associated class has infinite multiclass Littlestone dimension. To this end, we have to introduce the *ordinal multiclass Littlestone dimension*, which quantifies "how infinite" the multiclass Littlestone dimension is. Hence, the learner's strategy will be to play according to the *ordinal Standard Optimal Algorithm*. The intuition is similar as in the classical setting, but the proof becomes more involved. To establish the result, we follow the approach of [BHM+21] and define a more general infinite game $\mathcal{G}$ between the learner and the adversary and prove that it belongs to the family of the so-called Gale-Stewart games. Then, we leverage results from the theory of Gale-Stewart games and [BHM+21] in order to show that the ordinal SOA makes a finite number of mistakes whenever $\mathcal{H}$ does not have an infinite multiclass Littlestone tree. Let us continue with the proof.

*Proof of Theorem 9.* We first introduce a two-player game $\mathcal{G}$ that is played in discrete timesteps $t = 1, 2, \ldots$ between the adversary and the learner.

---

1. The adversary picks a point $\kappa_t = \left( \xi_t, y_t^{(0)}, y_t^{(1)} \right) \in \mathcal{X} \times [k] \times [k]$ and reveals it to the learner.
2. The learner chooses a point $\eta_t \in \{0, 1\}$.

---

Figure 4: Adversarial Setting - 2-Player Game

The learning player wins in some finite round $t$ if $\mathcal{H}_{\xi_1, y_1^{(\eta_1)}, \ldots, \xi_t, y_t^{(\eta_t)}} = \emptyset$. The adversary wins if the game continues indefinitely (i.e., the class of consistent hypotheses from $\mathcal{H}$ never gets empty) . Clearly, the set of winning strategies for the learning player is

$$\mathcal{W} = \left\{ (\boldsymbol{\kappa}, \boldsymbol{\eta}) \in (\mathcal{X} \times [k] \times [k] \times \{0,1\})^{\infty} : \exists\, 1 \leq t^{\star} < \infty \text{ such that } \mathcal{H}_{\xi_1, y_1^{(\eta_1)}, \ldots, \xi_{t^\star}, y_{t^\star}^{(\eta_{t^\star})}} = \emptyset \right\}.$$

We now recall an important theorem (see Appendix B.6) about Gale-Stewart games: In any Gale-Stewart game, exactly one of the adversary player and the learning player has a winning strategy.

Equipped with Appendix B.6, we can show that the adversary has a winning strategy if and only if $\mathcal{H}$ has an infinite multiclass Littlestone tree (provided that $\mathcal{G}$ is a Gale-Stewart game). This is summarized in the next claim.

**Claim 1.** *The game $\mathcal{G}$ is a Gale-Stewart game and the adversary has a winning strategy in $\mathcal{G}$ if and only if the hypothesis class $\mathcal{H}$ has an infinite multiclass Littlestone tree.*

*Proof.* It is clear from the definition of $\mathcal{W}$ that every winning strategy of the learner is finitely decidable, hence $\mathcal{G}$ is a Gale-Stewart game. For the other part of the claim, notice that if $\mathcal{H}$ has an infinite multiclass Littlestone tree, then the adversary's strategy is to present the learner at step $t$ the point of the tree at depth $t$ that is consistent with the execution of the game so far along with the labels of the edges that connect it with its children. By the definition of the tree, this strategy ensures that the game will keep going on forever. For the other direction, assume that the adversary has a winning strategy $\kappa_\tau(\eta_1, \ldots, \eta_{\tau-1}) \in \mathcal{X} \times [k] \times [k]$. Then, define the multiclass Littlestone tree $\mathcal{T} = \{x_{\boldsymbol{u}} : 0 \le k < \infty, \boldsymbol{u} \in \{0,1\}^k\}$ where $x_{\eta_1, \ldots, \eta_{\tau-1}} = \xi_\tau(\eta_1, \ldots, \eta_{\tau-1})$ where the labels that connect $x_{\eta_1, \ldots, \eta_{\tau-1}}$ with its left, right children are $y_\tau^{(0)}(\eta_1, \ldots, \eta_{\tau-1}), y_\tau^{(1)}(\eta_1, \ldots, \eta_{\tau-1})$, respectively. We can see that $\mathcal{T}$ is infinite since this is a winning strategy for the adversary. $\square$

Having shown the above statement, we are ready to establish the desired dichotomy in the online game. Assume first that $\mathcal{H}$ has an infinite multiclass Littlestone tree $\{x_u\}$. The adversary's strategy is defined inductively based on the path followed so far in the game: In round $t$, set $\boldsymbol{b}_t = (b_1, ..., b_{t-1}) \in \{0,1\}^{t-1}$ denote the path parsed so far in the tree by the two players. Then, the adversary picks $x_t = x_{\boldsymbol{b}_t}$. After the learner reveals her choice $\widehat{y}_t$, the worst case adversary chooses as a response the branch of the Littlestone tree which does not correspond to the learner's choice (the adversary may even have two choices). By the definition of the tree, this chosen label is valid since there exists some $h \in \mathcal{H}$ that realizes the path $(x_{b_1}, ..., x_{b_{t-1}}, x_t)$. Moreover, this choice provokes a mistake to the learning player and this is true for any round. Hence, there is a strategy for the adversary that forces any learner to make a mistake in every round.

For the other direction, assume that the class $\mathcal{H}$ does not have an infinite multiclass Littlestone tree. Before we describe the winning strategy of the learner, we need to introduce the notion of ordinal multiclass Littlestone dimension. We will assign an ordinal to every finite multiclass Littlestone tree. For some preliminaries on ordinals and transfinite recursion, we refer to [BHM+21]. The rank is defined by a partial order $\prec$. We set $t' \prec t$ if $t'$ is a multiclass Littlestone tree that extends $t$ by one level, i.e., $t$ is obtained from $t'$ by removing its leaves. A multiclass Littlestone tree $t$ is minimal if it cannot be extended to a multiclass Littlestone tree of larger depth. For such a tree, we set $\mathrm{rank}(t) = 0$. If the tree $t$ is non-minimal, then it can be extended and this is quantified using transfinite recursion by

$$\mathrm{rank}(t) = \sup\{\mathrm{rank}(t') + 1 : t' \prec t\}.$$

The rank is well-defined as long as $\mathcal{H}$ has no infinite multiclass Littlestone tree (since $\prec$ is well-founded). In particular, we define

$$\overline{\mathrm{Ldim}}_k(\mathcal{H}) = \begin{cases} -1 & \text{if } \mathcal{H} \text{ is empty}, \\ \Omega & \text{if } \mathcal{H} \text{ has an infinite multiclass Littlestone tree}, \\ \mathrm{rank}(\emptyset) & \text{otherwise}. \end{cases}$$

The strategy is chosen so that $\overline{\mathrm{Ldim}}_k(\mathcal{H}_{x_1, y_1, \ldots, x_t, y_t})$ decreases in every round and the learner that follows this strategy will win the game, since the ordinals do not admit an infinite decreasing chain. We note that this statement at first is purely existential via the theory of Gale-Stewart games. We next shortly provide a "constructive" way to compute the winning strategy of the learning player in the set of games we consider. Let us describe the winning strategy: The learner invokes the ordinal (multiclass) Standard Optimal Algorithm and chooses the label $y_t$ (given $x_t$) that maximizes the ordinal multiclass Littlestone dimension, i.e., $y_t = \mathrm{argmax}_{y \in [k]} \overline{\mathrm{Ldim}}_k(V_t^y)$, where $= V_t^y = \{h \in \mathcal{H}_{x_1, y_1, \ldots, x_{t-1}, y_{t-1}} : h(x_t) = y\}$. The ordinal SOA at round $t = 1, 2, \ldots$ with initial set $V_0 = \mathcal{H}$ works as follows:

1. Receive $x_t$.

2. For any $y \in [k]$, let $V_t^y = \{h \in V_{t-1} : h(x_t) = y\}$.

3. Predict $\widehat{y}_t \in \mathrm{argmax}_{y \in [k]} \overline{\mathrm{Ldim}}_k(V_t^y)$, where $\overline{\mathrm{Ldim}}_k$ is the ordinal multiclass Littlestone dimension.

4. Receive true answer $y_t$ and set $V_t = V_t^{y_t}$.

This algorithm drives the game in a win-win phenomenon for the learner in every round: If the adversary forces the learner to a mistake, then she will "prune" the tree and set the learner closer

to winning the game. Otherwise, the learner will be correct and will not incur any loss. In order to show that the ordinal SOA makes a finite number of mistakes, we couple the online game with a Gale-Stewart game. The idea is that every time the learner makes a mistake in the online game on point $x_t$, we advance the Gale-Stewart game by one round where we pretend that $\xi_\tau = x_t, y_\tau^{(0)} = \widehat{y}_t, y_\tau^{(1)} = y_t, \eta_\tau = y_t$. Notice that if the learner makes an infinite number of mistakes in the online game using the ordinal SOA, then the Gale-Stewart game can proceed infinitely. Hence, to conclude the proof, we need to show that in this coupled game, there is some finite point $\tau^*$ such that $\mathcal{H}_{\xi_1, y_1^{(\eta_1)}, \ldots, \xi_{\tau^*}, y_{\tau^*}^{(\eta_{\tau^*})}} = \emptyset$. The following result helps us establish that. In fact, the next lemma follows from [BHM$^+$21](Proposition B.8) by choosing the value of the game being the ordinal multiclass Littlestone dimension.

**Lemma 3** (See Proposition B.8 of [BHM$^+$21])**.** *Assume that $\mathcal{H}$ does not contain an infinite multiclass Littlestone tree. Then, for any choices of the adversary $\kappa_1, \ldots, \kappa_{t-1}$ up to round $t$ and for any choice $\kappa_t = (\xi_t, y_t^{(0)}, y_t^{(1)})$ in round $t$ there is a choice $\eta_t$ of the learner such that*

$$\overline{\mathrm{Ldim}}_k \left( \mathcal{H}_{\xi_1, y_1^{(\eta_1)}, \ldots, \xi_{t-1}, y_{t-1}^{(\eta_{t-1})}, \xi_t, y_t^{(\eta_t)}} \right) < \overline{\mathrm{Ldim}}_k \left( \mathcal{H}_{\xi_1, y_1^{(\eta_1)}, \ldots, \xi_{t-1}, y_{t-1}^{(\eta_{t-1})}} \right) .$$

The previous result shows that for every $\xi_t$ there is at most one label $\ell_t \in [k]$ such that

$$\overline{\mathrm{Ldim}}_k \left( \mathcal{H}_{\xi_1, y_1^{(\eta_1)}, \ldots, \xi_{t-1}, y_{t-1}^{(\eta_{t-1})}, \xi_t, \ell_t} \right) = \overline{\mathrm{Ldim}}_k \left( \mathcal{H}_{\xi_1, y_1^{(\eta_1)}, \ldots, \xi_{t-1}, y_{t-1}^{(\eta_{t-1})}} \right) .$$

Indeed, assume that there are two such labels $\ell_t, \ell_t'$ for some $\xi_t$. Then, if the adversary proposes the point $(\xi_t, \ell_t, \ell_t')$, there is no choice $\eta_t$ of the learner that decreases that ordinal Littlestone dimension in this round, which leads to a contradiction. Hence, the learner can pick any label as long as it is not the one that maximizes the ordinal Littlestone dimension. This is exactly how the coupled Gale-Stewart game proceeds, so we know that every time the learner makes a mistake in the online game the ordinal Littlestone dimension of the coupled game decreases. Since ordinals that are less than $\Omega$ do not admit infinitely decreasing chains, we get the desired result.

$\square$

### C.1.2 Moving from the Adversarial Setting to the Probabilistic Setting

The measurability of the winning strategies and of the learning algorithm developed in the previous section constitutes an important detail, extensively discussed in [BHM$^+$21], in order to move from the adversarial setting to the probabilistic one. We provide the next useful result.

**Lemma 4.** *Let $\mathcal{X}$ be Polish, $k \in \mathbb{N}$ be a finite constant and $\mathcal{H} \subseteq [k]^{\mathcal{X}}$ be measurable. Then, the Gale-Stewart game $\mathcal{G}$ of Figure 4 has a universally measurable winning strategy.*

*Proof.* It suffices to prove that the set $\mathcal{W}$ of the winning strategies for the learning player in the Gale-Stewart game is coanalytic (see Lemma 2). Equivalently, we will prove that the set of winning strategies of the adversary $\mathcal{W}^c$ is analytic, where

$$\mathcal{W}^c = \left\{ (\boldsymbol{\kappa}, \boldsymbol{\eta}) \in (\mathcal{X} \times [k] \times [k] \times \{0, 1\})^\infty : \mathcal{H}_{\xi_1, y_1^{(\eta_1)}, \ldots, \xi_t, y_t^{(\eta_t)}} \neq \emptyset \text{ for all } t < \infty \right\} .$$

This set is equal to

$$\mathcal{W}^c = \bigcap_{1 \leq \tau < \infty} \bigcup_{\theta \in \Theta} \bigcap_{1 \leq t \leq \tau} \left\{ (\boldsymbol{\kappa}, \boldsymbol{\eta}) \in (\mathcal{X} \times [k] \times [k] \times \{0, 1\})^\infty : h(\theta, \xi_t) = y_t^{(\eta_t)} \right\} .$$

The set $\left\{ (\theta, \boldsymbol{\kappa}, \boldsymbol{\eta}) : h(\theta, \xi_i) = y_i^{(\eta_j)} \right\} = \bigcup_{(y_i^{(0)}, y_i^{(1)}) \in [k] \times [k], y_i^{(0)} \neq y_i^{(1)}} \left\{ (\theta, \boldsymbol{\xi}, \boldsymbol{\eta}) : h(\theta, \xi_i) = y_i^{(\eta_j)} \right\}$ is a Borel set using the standard measurability assumption of Definition 9. The set $\mathcal{W}^c$ is analytic since the two intersections are over countable sets and the union is a projection of a Borel set. $\square$

Crucially the above result states that the winning strategy $\eta_t$ of the learning player is measurable. However, the previous proof made use of the ordinal multiclass SOA algorithm, whose measurability is not directly implied. To this end, we modify the adversarial algorithm to handle the measurability issue. The modification follows:

1. Initialize $\tau \leftarrow 1, G = \texttt{Clique}(V = [k]), f(\cdot, \cdot, \cdot) \leftarrow \eta_1(\cdot, \cdot, \cdot)$     $\triangleright \tau$ is the mistake counter
2. For every round $t \geq 1$ :
    (a) Observe $x_t$
    (b) For any $y \neq y'$ with $y, y' \in [k]$, orient the edge $(y, y')$ of $G$ according to $f(x_t, y, y')$
    (c) Let $G'$ the directed clique
    (d) Predict $\widehat{y}_t \leftarrow \text{argmax}_{y \in [k]} \text{outdeg}(y; G')$
    (e) If $\widehat{y}_t \neq y_t$, let $\xi_\tau \leftarrow x_t, f(\cdot, \cdot, \cdot) \leftarrow \eta_{\tau+1}(x_1, y_1, \ldots, x_\tau, y_\tau, \cdot, \cdot, \cdot), \tau \leftarrow \tau + 1$

Figure 5: Measurable Modification of Online Learning Algorithm for Exponential Rates

The above algorithm makes use of a tournament procedure. The algorithm is a measurable function since (i) the winning strategy of the learner is measurable and (ii) the countable maximum of measurable functions is measurable. This algorithm can be used in order to show that if $\mathcal{H}$ does not have an infinite multiclass Littlestone tree, then the above algorithm makes only a finite number of mistakes against any adversary. Essentially, this is due to the fact that when the winning strategy has converged to a zero-mistake prediction rule (which occurs after a finite number of mistakes), the tournament procedure will always output the correct label for the observed example. Hence, the algorithm will eventually make a finite number of mistakes in the adversarial setting.

The algorithm of Figure 5 works in the adversarial setting. We first show that it also applies to the probabilistic setting (and this is why we require the above measurability discussion). The proof is quite similar to Lemma 4.3 of [BHM$^+$21].

**Lemma 5** (From Adversarial to Probabilistic). *For any distribution $P$ over $\mathcal{X} \times [k]$ and for the learning algorithm $\widehat{y}_t : \mathcal{X} \rightarrow [0, 1]$ of Theorem 9, we have*

$$\Pr_{S_t} \left[ \Pr_{(x,y) \sim P} [\widehat{y}_t(x) \neq y] > 0 \right] \rightarrow 0 \text{ as } t \rightarrow \infty \,,$$

*where $S_t$ is the training set $(x_1, y_1, ..., x_{t-1}, y_{t-1})$ of the algorithm.*

*Proof.* Since the distribution $P$ is realizable, there exists a sequence of functions $h_k \in \mathcal{H}$ so that

$$\Pr_{(x,y) \sim P} [h_k(x) \neq y] < \frac{1}{2^k} \,.$$

Let us fix $t \geq 1$. We have that

$$\sum_{k=1}^{\infty} \mathbf{Pr}[\exists s \leq t : h_k(X_s) \neq Y_s] \leq t \sum_{k=1}^{\infty} \Pr_{(X,Y) \sim P} [h_k(X) \neq Y] < \infty \,,$$

where the first inequality is due to union bound. By Borel-Cantelli, with probability one, there exists for every $t \geq 1$ a hypothesis $h \in \mathcal{H}$ so that $h(X_s) = Y_s$ for all $s \leq t$. Hence, the sequence $X_1, Y_1, X_2, Y_2, ...$ is a valid input for the online learning game with probability one. In particular, we make use of the following statement: If $\mathcal{H}$ does not have an infinite multiclass Littlestone tree, then there is a strategy for the learner that makes only finitely many mistakes against any adversary. This is proved in Theorem 9. The existence of a winning strategy $\widehat{y}_t$ for the learning player implies that the time $T$ where the player makes a mistake is

$$T = \sup\{s \in \mathbb{N} : \widehat{y}_{s-1}(X_s) \neq Y_s\}$$

is a random variable that is finite with probability one. Moreover, the online learner is selected so that it is changed only when a loss is observed. This means that $\widehat{y}_s = \widehat{y}_t$ for all rounds $s \geq t \geq T$.

We now employ the law of large numbers in order to understand the asymptotic behavior of the online learner:

$$\mathbf{Pr} \left[ \Pr_{(x,y) \sim P} [\widehat{y}_t(x) \neq y] = 0 \right] = \mathbf{Pr} \left[ \lim_{S \rightarrow \infty} \frac{1}{S} \sum_{s=t+1}^{t+S} \mathbf{1}\{\widehat{y}_t(X_s) \neq Y_s\} = 0 \right]$$

and this probability is at least the probability of this event and of the event that $T \leq t$, i.e.,

$$\mathbf{Pr}\left[\mathop{\mathbf{Pr}}_{(x,y)\sim P}[\widehat{y}_t(x) \neq y] = 0\right] \geq \mathbf{Pr}\left[\lim_{S\to\infty} \frac{1}{S}\sum_{s=t+1}^{t+S} \mathbf{1}\{\widehat{y}_t(X_s) \neq Y_s\} = 0, T \leq t\right] = \mathbf{Pr}[T \leq t],$$

where the last inequality follows from the observation that since $s \geq t$ and $t$ is greater than the critical time $T$ then the first event occurs with probability one. This implies that

$$\mathbf{Pr}\left[\mathop{\mathbf{Pr}}_{(x,y)\sim P}[\widehat{y}_t(x) \neq y] = 0\right] \geq \mathbf{Pr}[T \leq t]$$

and so

$$\mathbf{Pr}\left[\mathop{\mathbf{Pr}}_{(x,y)\sim P}[\widehat{y}_t(x) \neq y] > 0\right] \leq \lim_{t\to\infty} 1 - \mathbf{Pr}[T \leq t] = 0\,.$$

$\square$

The above result guarantees that the expected error of the learning algorithm tends to zero as $t$ goes to infinity, i.e., we have established that that $\mathbf{E}[\mathrm{er}(\widehat{y}_t)] \to 0$ as $t \to \infty$. This means that the ordinal SOA is a consistent algorithm in the statistical setting. However, this fact is not enough to establish the exponential convergence rate. We follow the approach of [BHM$^+$21] to come up with an algorithm that achieves this guarantee. The first step is to observe that there is some distribution-dependent $t^\star$ such that $\mathbf{Pr}[\mathrm{er}(\widehat{y}_{t^\star}) > 0] < 1/4$. If we were to know this $t^\star$ one way to get the exponential rates is the following: We divide the training set into $\Theta(n/t^\star)$ batches and we get one classifier $\widehat{y}_{t^\star}^i$ for every batch. Afterwards, we output as classifier the multiclass majority vote among the classifiers.

**Theorem 10** (Exponential Rates). *Assume that class $\mathcal{H} \subseteq [k]^{\mathcal{X}}$ does not have an infinite multiclass Littlestone tree. Then, $\mathcal{H}$ admits a learning algorithm that achieves an exponentially fast rate.*

*Proof.* Consider the sequence $\widehat{t}_n$ which satisfies the properties of Lemma 6. Consider the collection of the learners $\mathbb{A}_{i,n} := \widehat{y}_{\widehat{t}_n}^i : \mathcal{X} \to [k]$ for any $1 \leq i \leq \lfloor \frac{n}{2\widehat{t}_n}\rfloor$ and $n \in \mathbb{N}$. Let us fix a time $t \in T^\star$, where $T^\star$ is the set of good estimates of the critical time $t^\star$ (see Lemma 6). We have that

$$\mathbf{Pr}\left[\frac{1}{\lfloor\frac{n}{2t}\rfloor}\sum_{i=1}^{\lfloor\frac{n}{2t}\rfloor} \mathbf{1}\left\{\mathop{\mathbf{Pr}}_{(x,y)\sim P}[\widehat{y}_t^i(x) \neq y] > 0\right\} > \frac{7}{16}\right] \leq \exp\left(-\left\lfloor n/(2t^\star)\right\rfloor/128\right),$$

using Hoeffding's inequality. The above probability is over the sequence of all the training sets and essentially states that the "bad" event that the misclassification error is non-zero holds for the majority of the trained algorithms $\widehat{y}_t^i$ with exponentially small probability. Conversely, except on an event of exponentially small probability, we have that $\mathbf{Pr}_{(x,y)\sim P}[\widehat{y}_t^i(x) \neq y] = 0$ for the majority of $i$. Recall that the above discussion holds for a chosen good $t$. We have to understand how well our learners $\mathbb{A}_{i,n}$ perform. To this end, we have that

$$\mathbf{Pr}\left[\mathop{\mathbf{Pr}}_{(x,y)\sim P}[\mathbb{A}_{i,n}(x) \neq y] > 0 \text{ for the majority of } i \leq \lfloor n/(2\widehat{t}_n)\rfloor\right]$$

$$\leq \mathbf{Pr}\left[\widehat{t}_n \notin T^\star\right] + \mathbf{Pr}\left[\exists t \in T^\star : \mathop{\mathbf{Pr}}_{(x,y)\sim P}[\widehat{y}_t^i(x) \neq y] > 0 \text{ for the majority of } i \leq \lfloor n/(2\widehat{t}_n)\rfloor\right]$$

$$\leq C(P)\exp(-c(P)\cdot n) + t^\star \cdot \exp\left(-\left\lfloor n/(2t^\star)\right\rfloor/128\right).$$

This implies that the majority of our learners $\mathbb{A}_{i,n}$ will not incur loss except on an event of exponentially small probability. As a result, the majority vote of these classifiers is almost surely correct on a random sample from the distribution $P$ over $\mathcal{X} \times [k]$. Hence, we have that

$$\mathbf{E}\left[\mathop{\mathbf{Pr}}_{(x,y)\sim P}[\mathrm{Maj}((\mathbb{A}_{i,n}(x))_i) \neq y]\right] \leq \mathbf{Pr}\left[\mathop{\mathbf{Pr}}_{(x,y)\sim P}[\mathrm{Maj}((\mathbb{A}_{i,n}(x))_i) \neq y] > 0\right]$$

$$\leq C(P)\exp(-c(P)\cdot n) + t^\star \cdot \exp\left(-\left\lfloor n/(2t^\star)\right\rfloor/128\right).$$

This concludes the proof. $\square$

**Lemma 6.** *For any $n \in \mathbb{N}$, there exists a universally measurable $\widehat{t}_n = \widehat{t}_n(X_1, Y_1, ..., X_n, Y_n)$ whose definition does not depend on $P$ so that the following holds. Set the critical time $t^\star \in \mathbb{N}$ be such that*

$$\mathbf{Pr}\left[\mathbf{Pr}_{(x,y)\sim P}[\widehat{y}_{t^\star}(x) \neq y] > 0\right] \leq 1/8\,,$$

*where the probability is over the training set of the algorithm $\widehat{y}_t$. There exist $C, c > 0$ that depend on $P, t^\star$ but not $n$ so that*

$$\mathbf{Pr}[\widehat{t}_n \in T^\star] \geq 1 - Ce^{-cn}\,.$$

*where the probability is over the training of the estimator $\widehat{t}_n$ and $T^\star$ is the set*

$$T^\star = \left\{1 \leq t \leq t^\star : \mathbf{Pr}\left[\mathbf{Pr}_{(x,y)\sim P}[\widehat{y}_t(x) \neq y] > 0\right] \leq 3/8\right\}\,,$$

*where the probability is over the training of $\widehat{y}_t$.*

*Proof.* We split the training set into two sets. The idea is to use the first one to train the learning algorithm and the other set to estimate the generalization error. For each $1 \leq t \leq \lfloor \frac{n}{2} \rfloor$ and $1 \leq i \leq \lfloor \frac{n}{2t} \rfloor$, we let

$$\widehat{y}_t^i(x) = \widehat{Y}_{t+1}(X_{(i-1)t+1}, Y_{(i-1)t+1}, X_{it}, Y_{it}, x)$$

be the output of the learning algorithm that is trained on batch $i$ of the data. For every fixed $t$, the data that the classifiers $\{\widehat{y}_t^i\}_{i \leq \lfloor n/2t \rfloor}$ are trained on are independent of each other and of the second half of the training set. This means that we can view every $\{\widehat{y}_t^i\}_{i \leq \lfloor n/2t \rfloor}$ as an independent draw of the distribution of $\widehat{y}_t$. To estimate the generalization error of the algorithm we use the second half of the training set. We let

$$\widehat{e}_t = \frac{1}{\lfloor n/2t \rfloor} \sum_{i=1}^{\lfloor n/2t \rfloor} \mathbf{1}\{\widehat{y}_t^i(X_s) \neq Y_s \text{ for some } n/2 \leq s \leq n\}\,.$$

Now observe that, with probability one,

$$\widehat{e}_t \leq e_t = \frac{1}{\lfloor n/2t \rfloor} \sum_{i=1}^{\lfloor n/2t \rfloor} \mathbf{1}\left\{\mathbf{Pr}_{(X,Y)\sim P}[\widehat{y}_t^i(X) \neq Y] > 0\right\}\,.$$

We define $\widehat{t}_n = \inf\{t \leq \lfloor n/2 \rfloor : \widehat{e}_t < 1/4\}$, where we assume that $\inf \emptyset = \infty$.

We now want to bound the probability that $\widehat{t}_n > t^\star$. Using Hoeffding's inequality we get that

$$\mathbf{Pr}\left[\widehat{t}_n > t^\star\right] \leq \mathbf{Pr}\left[\widehat{e}_{t^\star} \geq \frac{1}{4}\right] \leq \mathbf{Pr}\left[e_{t^\star} \geq \frac{1}{4}\right] =$$

$$= \mathbf{Pr}\left[e_{t^\star} - \frac{1}{8} \geq \frac{1}{8}\right] = \mathbf{Pr}\left[e_{t^\star} - \mathbf{E}[e_{t^\star}] \geq \frac{1}{8}\right] \leq e^{-\lfloor n/2t^\star \rfloor/32}\,.$$

This implies that $\widehat{t}_n \leq t^\star$ except for an event with exponentially small probability.

Moreover, for all $1 \leq t \leq t^\star$ that $\mathbf{Pr}\left[\mathbf{Pr}_{(x,y)\sim P}[\widehat{y}_t(x) \neq y] > 0\right] > \frac{3}{8}$, there is some $\epsilon > 0$ such that $\mathbf{Pr}\left[\mathbf{Pr}_{(x,y)\sim P}[\widehat{y}_t(x) \neq y] > \epsilon\right] > \frac{1}{4} + \frac{1}{16}$ (this holds by continuity). Now fix some $1 \leq t \leq t^\star$ such that $\mathbf{Pr}\left[\mathbf{Pr}_{(x,y)\sim P}[\widehat{y}_t(x) \neq y] > 0\right] > \frac{3}{8}$ (if it exists). Then, using Hoeffding's inequality again we get that

$$\mathbf{Pr}\left[\frac{1}{\lfloor n/2t \rfloor} \sum_{i=1}^{\lfloor n/2t \rfloor} \mathbf{1}\left\{\mathbf{Pr}_{(x,y)\sim P}[\widehat{y}_t^i(x) \neq y] > \epsilon\right\} < \frac{1}{4}\right] \leq e^{\lfloor n/2t^\star \rfloor/128}\,.$$

Whenever $f$ is a function such that $\mathbf{Pr}_{(x,y)\sim P}[f(x) \neq y] > \epsilon$, then

$$\mathbf{Pr}\left[f(X_s) \neq Y_s \text{ for some } n/2 \leq s \leq n\right] \geq 1 - (1 - \epsilon)^{n/2}\,.$$

As we mentioned before, $\{\widehat{y}_t^i\}_{i \leq \lfloor n/2t \rfloor}$ are independent of $(X_s, Y_s)_{s > n/2}$. Thus, applying a union bound we get that the probability that all $\widehat{y}_t^i$ that have $\mathbf{Pr}_{(x,y) \sim P}[\widehat{y}_t^i(x) \neq y] > \epsilon$ make at least one error on the second half of the training set is

$$\mathbf{Pr}\left[\sum_{i=1}^{\lfloor n/2t \rfloor} \mathbf{1}\left\{\Pr_{(x,y) \sim P}[\widehat{y}_t^i(x) \neq y] > \epsilon\right\} \leq \sum_{i=1}^{\lfloor n/2t \rfloor} \mathbf{1}\{\widehat{y}_t^i(X_s) \neq Y_s \text{ for some } n/2 < s \leq n\}\right] \geq 1 - \left\lfloor\frac{n}{2t}\right\rfloor(1-\epsilon)^{n/2}.$$

Thus, we get that

$$\mathbf{Pr}[\widehat{t}_n = t] \leq \mathbf{Pr}\left[\widehat{e}_t < \frac{1}{4}\right] \leq \left\lfloor\frac{n}{2}\right\rfloor(1-\epsilon)^{n/2} + e^{-\lfloor\frac{n}{2t^\star}\rfloor/128}.$$

Using the previous estimates and applying a union bound, we get that

$$\mathbf{Pr}[\widehat{t}_n \notin T^\star] \leq e^{-\lfloor n/2t^\star \rfloor/32} + t^\star \left\lfloor\frac{n}{2}\right\rfloor(1-\epsilon)^{n/2} + t^\star e^{-\lfloor n/2t^\star \rfloor/128} \leq Ce^{-cn},$$

for some constants $C, c > 0$. Note that $C = C(P, t^\star)$ and $c = c(P, t^\star)$. $\qquad\square$

## C.2 Infinite Multiclass Littlestone Trees and Rates

We next show that if $\mathcal{H}$ has an infinite mutliclass Littlestone tree, then there exists a significant drop in the rate: any learning algorithm $\mathcal{H}$ cannot be faster than linear. Our proof follows the approach in [BHM+21].

**Theorem 11.** *Assume that $\mathcal{H} \subseteq \{0, 1, ..., k\}^{\mathcal{X}}$ has an infinite multiclass Littlestone tree. Then, for any learning algorithm $\widehat{h}_n$, there exists a realizable distribution $P$ over $\mathcal{X} \times [k]$ such that $\mathbf{E}[\mathrm{er}(\widehat{h}_n)] \geq \Omega(1/n)$ for infinitely many $n$. This means that $\mathcal{H}$ is not learnable at rate faster than linear, i.e., $R(n) \geq 1/n$.*

Let us provide some intuition. We will make use of the probabilistic method. We are going to define a distribution over probability distributions so that, with positive probability over this choice of the random object, any learning algorithm will have an expected error of order $\Omega(1/n)$. This positive probability implies that there exists such a distribution and hence the above result holds true. The key idea is that we are going to associate any random distribution $P_{\boldsymbol{y}}$ with a branch $\boldsymbol{y}$ of the infinite multiclass Littlestone tree. Given a finite number of samples from this distribution, only a finite part of this infinite path will be discovered; hence any algorithm after the "revealed path" must guess whether this random path goes left or right. This implies that the algorithm will err with probability $1/2$ when it observes a point that lies deeper in the branch than the training examples.

*Proof.* Fix any learner $\widehat{h}_n$ and an infinite multiclass Littlestone tree for $\mathcal{H}$. Fix also a random branch $\boldsymbol{y} = (y_1, y_2, ...)$ of this tree, where the sequence is an i.i.d. sequence of fair Bernoulli coins. We introduce the random distribution over $\mathcal{X} \times \{0, 1, ..., k\}$ as

$$P_{\boldsymbol{y}}((x_{\boldsymbol{y}_{\leq \ell}}, z_{\ell+1})) = \frac{1}{2^{\ell+1}}, \ell \geq 0,$$

where $z_{\ell+1} \in \{0, 1, ..., k\}$ is the label of the edge connecting $x_{\boldsymbol{y}_{\leq \ell}}$ to its child according to the chosen path $\boldsymbol{y}$. For any $n < \infty$, there exists a hypothesis $h \in \mathcal{H}$ so that

$$h(x_{\boldsymbol{y}_{\leq \ell}}) = z_{\ell+1}$$

for $0 \leq \ell \leq n$. This is due to the construction of a multiclass Littlestone tree. We have that

$$\mathrm{er}_{\boldsymbol{y}}(h) = \Pr_{(x,z) \sim P_{\boldsymbol{y}}}[h(x) \neq z] \leq \sum_{\ell > n} 2^{-\ell-1},$$

which goes to 0 as $n \to \infty$. This implies that $P_{\boldsymbol{y}}$ is realizable for every infinite branch $\boldsymbol{y} \in \{0, 1\}^\infty$. Let us draw $(X, Z), (X_1, Z_1), (X_2, Z_2), ...$ i.i.d. samples from $P_{\boldsymbol{y}}$. Moreover, the mapping $y \to P_y$ is measurable. The first sample corresponds to the test sample and the other samples deal with the training phase. Moreover, let $T, T_1, T_2, ...$ be i.i.d. Geometric random variables with success probability $1/2$ starting at 0. We can set

1. $X = x_{\boldsymbol{y} \leq T}, Z = z_{T+1}$ and

2. $X_i = x_{\boldsymbol{y} \leq T_i}, Z_i = z_{T_i+1}$.

Crucially, on the event that $\{T = \ell, \max\{T_1, ..., T_n\} < \ell\}$, the value of $\widehat{h}_n(X)$ is conditionally independent of $z_{\ell+1}$ given $X, (X_1, Z_1), ..., (X_n, Z_n)$. We next have that

$$\mathbf{Pr}[\widehat{h}_n(X) \neq Z, T = \ell, \max\{T_1, ..., T_n\} < \ell] = \mathbf{Pr}[\widehat{h}_n(X) \neq Z_{\ell+1}, T = \ell, \max\{T_1, ..., T_n\} < \ell].$$

This is equal to

$$\mathbf{E}[\mathbf{Pr}_Z[\widehat{h}_n(X) \neq Z | X, (X_1, Z_1), ..., (X_n, Z_n)]\mathbf{1}\{T = \ell, \max\{T_1, ..., T_n\} < \ell\}]$$

Now conditional on this event, any algorithm will err with probability $1/2$ (since it will guess the value at random). Hence, this quantity is lower bounded by

$$\frac{1}{2}\mathbf{Pr}[T = \ell, \max\{T_1, ..., T_n\} < \ell] = 2^{-\ell-2}(1 - 2^{-\ell})^n.$$

We are free now to pick $\ell$. Choosing $\ell = \ell_n := \lceil 1 + \log(n) \rceil$, we have that $1/2^\ell > 1/(4n)$ and $(1 - 2^{-\ell})^n \geq 1/2$. Our goal is to apply the reverse Fatou lemma. This can be done since almost surely, we have that

$$n\mathbf{Pr}[\widehat{h}_n(X) \neq Z, T = \ell_n | \boldsymbol{y}] \leq n\mathbf{Pr}[T = \ell_n] = n2^{-\ell_n-1} \leq 1/4.$$

Hence, we can apply the reverse Fatou lemma and get

$$\mathbf{E}\left[\limsup_{n \to \infty} n\mathbf{Pr}[\widehat{h}_n(X) \neq Z, T = \ell_n | \boldsymbol{y}]\right] \geq \limsup_{n \to \infty} n\mathbf{Pr}[\widehat{h}_n(X) \neq Z, T = \ell_n] > 1/32.$$

But, almost surely, it holds that

$$\mathbf{E}[\mathrm{er}_{\boldsymbol{y}}(\widehat{h}_n) | \boldsymbol{y}] = \mathbf{Pr}[\widehat{h}_n(X) \neq Z | \boldsymbol{y}] \geq \mathbf{Pr}[\widehat{h}_n(X) \neq Z, T = \ell_n | \boldsymbol{y}].$$

So, combining the above inequalities

$$\mathbf{E}\left[\limsup_{n \to \infty} n\mathbf{E}[\mathrm{er}_{\boldsymbol{y}}(\widehat{h}_n)]\right] > \Omega(1).$$

Hence, there must exist a realization of $\boldsymbol{y}$ so that $\mathbf{E}[\mathrm{er}_{\boldsymbol{y}}(\widehat{h}_n)] = \Omega(1/n)$ infinitely often. Choosing $P = P_{\boldsymbol{y}}$ completes the proof. $\qquad\square$

The above result states that a class which does not have exponential rates cannot be learned faster than linearly. However, it is not clear if even linear rates are achievable. The next section deals with this case.

### C.3 Linear Learning Rates and Natarajan-Littlestone Trees

We introduce a novel combinatorial measure, the Natarajan-Littlestone (NL) tree, which essentially combines the structure of the Natarajan dimension and the Littlestone dimension.

**Definition 12.** *A Natarajan-Littlestone (NL) tree for $\mathcal{H} \subseteq [k]^{\mathcal{X}}$ of depth $d \leq \infty$ consists of a tree*

$$\bigcup_{0 \leq \ell < d} \{x_u \in \mathcal{X}^{\ell+1}, u \in \{0, 1\} \times \{0, 1\}^2 \times ... \times \{0, 1\}^\ell\}$$

*and two colorings $s^{(0)}, s^{(1)}$ mapping each position $u^i \in u$ for any node with pattern $u \in \{0, 1\} \times ... \times \{0, 1\}^\ell$ for $i \in \{0, 1, ..., \ell\}$ and $\ell \in \{0, 1, ..., d-1\}$ of the tree to some color $\{0, 1, ..., k\}$ such that for every finite level $n < d$, the subtree $T_n = \cup_{0 \leq \ell \leq n}\{x_u = (x_u^0, ..., x_u^\ell) : u \in \{0, 1\} \times \{0, 1\}^2 \times ... \times \{0, 1\}^\ell\}$ satisfies the following:*

1. *At any point $x_u^i \in x_u \in T_n$, it holds $s^{(0)}(x_u^i) \neq s^{(1)}(x_u^i)$ and*

2. *for any path $\boldsymbol{y} \in \{0,1\} \times ... \times \{0,1\}^{n+1}$, there exists a concept $h \in \mathcal{H}$ so that $h(x^i_{\boldsymbol{y}_{\leq \ell}}) = s^{(0)}(x^i_{\boldsymbol{y}_{\leq \ell}})$ if $y^i_{\ell+1} = 1$ and $h(x^i_{\boldsymbol{y}_{\leq \ell}}) = s^{(1)}(x^i_{\boldsymbol{y}_{\leq \ell}})$ otherwise, for all $0 \leq i \leq \ell$ and $0 \leq \ell \leq n$, where*

$$\boldsymbol{y}_{\leq \ell} = (y^0_1, (y^0_2, y^1_2), ..., (y^0_\ell, ..., y^{\ell-1}_\ell)), x_{\boldsymbol{y}_{\leq \ell}} = (x^0_{\boldsymbol{y}_{\leq \ell}}, ..., x^\ell_{\boldsymbol{y}_{\leq \ell}}).$$

*We say that $\mathcal{H}$ has an infinite NL tree if it has a NL tree of depth $d = \infty$.*

We note that in the above definition we identify the color $s^{(0)}(x^i_{\boldsymbol{y}_{\leq \ell}})$ with the (unique) position of this point $x^i_{\boldsymbol{y}_{\leq \ell}}$ (since typically the coloring is over positions). For short, we will call the colorings $s^{(0)}, s^{(1)}$ used in the above definition *everywhere different* and denote by $s^{(0)} \neq s^{(1)}$. As a sanity check, one can verify that if $\mathcal{H}$ has an infinite NL tree, then it has an infinite multiclass Littlestone tree. In fact, one can construct the latter tree by choosing only one point from any node of the infinite NL tree.

The main result of this section is the next theorem.

**Theorem 12.** *Assume that $\mathcal{H} \subseteq [k]^{\mathcal{X}}$ does not have an infinite Natarajan-Littlestone tree. Then, there exists an algorithm that learns $\mathcal{H}$ at a linear rate.*

### C.3.1 The Natarajan-Littlestone Game

In a similar manner as in the exponential rates setting, we need to come up with a function that is correct after a finite number of steps $n^*$ and this implies that some appropriate data-dependent complexity measure of $\mathcal{H}$ is finite. The first step towards establishing the desired result is to introduce the following two-player game between an adversary and a learning player.

---

1. The adversary picks $t$ points and two everywhere different colorings of these $t$ points $\xi_t = \left( \xi^{(0)}_t, ..., \xi^{(t-1)}_t, s^{(0)}_t, s^{(1)}_t \right) \in \mathcal{X}^t \times [k]^t \times [k]^t$ and reveals them to the learner.

2. The learner chooses a pattern $\eta_t = (\eta^{(0)}_t, ..., \eta^{(t-1)}_t) \in \{0,1\}^t$.

---

Figure 6: Adversarial Setting - 2-Player NL Game

Let $\Xi_t = \{\xi^{(0)}_1, \xi^{(0)}_2, \xi^{(1)}_2, ..., \xi^{(0)}_t, ..., \xi^{(t-1)}_t, s^{(0)}_1, s^{(1)}_1, ..., s^{(0)}_t, s^{(1)}_t\}$ be the sets of all points and colorings chosen by the adversary after $t$ rounds, for some finite integer $t$. The learning player wins the game in round $t$ if the class $\mathcal{H}_t = \mathcal{H}_{\xi_1, \eta_1, ..., \xi_t, \eta_t} = \emptyset$, where $\mathcal{H}_0 = \mathcal{H}$ and $\mathcal{H}_t = \mathcal{H}_{\xi_1, \eta_1, ..., \xi_t, \eta_t}$ is the set

$$\left\{ h \in \mathcal{H} \text{ so that } \begin{array}{l} h(\xi^{(i)}_z) = s^{(0)}_z(\xi^{(i)}_z) \text{ if } \eta^{(i)}_z = 0 \\ h(\xi^{(i)}_z) = s^{(1)}_z(\xi^{(i)}_z) \text{ if } \eta^{(i)}_z = 1 \end{array} \text{ for } 0 \leq i < z, z \in [1..t] \right\}.$$

Note that the collection of valid functions decreases as the game proceeds, i.e., $\mathcal{H}_t \subseteq \mathcal{H}_{t-1}$. The next lemma guarantees the existence of a winning strategy for the learner in the game.

**Lemma 7.** *If $\mathcal{H} \subseteq [k]^{\mathcal{X}}$ has no infinite NL tree, then there is a universally measurable winning strategy for the learning player in the game $\mathcal{G}$.*

*Proof.* The first point is that the 2-player game of Figure 6 is Gale-Stewart. This follows by the observation that the set of winning sequences of the learning player is finitely decidable, since the membership of $(\boldsymbol{\xi}, \boldsymbol{\eta})$ in the set is witnessed by a finite subsequence. Hence, exactly one of the two players has a winning strategy in the game (see Appendix B.6).

The second point is that the class $\mathcal{H}$ has an infinite NL tree if and only if the adversary player has a winning strategy in the game. Suppose that $\mathcal{H}$ has an infinite NL tree. The adversary can adopt the strategy iteratively by setting $\xi_t(\eta_1, ..., \eta_{t-1}) = (x_{\eta_1, ..., \eta_{t-1}}, s^{(0)}_{\eta_1, ..., \eta_{t-1}}, s^{(1)}_{\eta_1, ..., \eta_{t-1}}) \in \mathcal{X}^t \times [k]^t \times [k]^t$. Hence, the adversary traverses the infinite tree and, by construction of the NL tree, the set $\mathcal{H}_{\xi_1, \eta_1, ..., \xi_t, \eta_t}$ never gets empty for any sequence of patterns $\boldsymbol{\eta}$ and $t < \infty$. Thus, this is a winning strategy for the adversary player. In the opposite direction, assume that the adversary has a winning

strategy. Then, she can construct an infinite NL tree by setting $(x_{\eta_1,...,\eta_{t-1}}, s^{(0)}_{\eta_1,...,\eta_{t-1}}, s^{(1)}_{\eta_1,...,\eta_{t-1}}) = \xi_t(\eta_1,...,\eta_{t-1})$ for any possible binary pattern. Note that since the strategy is winning, the class $\mathcal{H}_{\xi_1,\eta_1,...,\xi_t,\eta_t} \neq \emptyset$ for any $t \in \mathbb{N}$ and this means that for any level of the tree, there exists two everywhere different colorings that witness the patterns.

Using the second point, we have that the learning player has a winning strategy in the game if the class $\mathcal{H}$ has no infinite NL tree. To show that this strategy is also universally measurable, it suffices to show that the set of winning sequences for the learning player is coanalytic. The set of winning strategies for the adversary is

$$\mathcal{W}^c = \left\{ ((\boldsymbol{\xi}, \boldsymbol{s}^{(0)}, \boldsymbol{s}^{(1)}), \boldsymbol{\eta}) \in \bigcup_{t=1}^{\infty} ((\mathcal{X}^t \times [k]^t \times [k]^t) \times \{0,1\}^t) : \mathcal{H}_{\xi_1,\eta_1,...,\xi_t,\eta_t} \neq \emptyset \text{ for all } t < \infty \right\} .$$

The set $\mathcal{W}^c$ is equal to

$$\bigcap_{1 \leq t < \infty} \bigcup_{\theta \in \Theta} \bigcap_{1 \leq \ell \leq t} \left\{ ((\boldsymbol{\xi}, \boldsymbol{s}^{(0)}, \boldsymbol{s}^{(1)}), \boldsymbol{\eta}) : \begin{array}{ll} h(\theta, \xi_z^{(i)}) = s_z^{(0)}(\xi_z^{(i)}) & \text{if } \eta_z^{(i)} = 0 \\ h(\theta, \xi_z^{(i)}) = s_z^{(1)}(\xi_z^{(i)}) & \text{if } \eta_z^{(i)} = 1 \end{array} \text{ for } 0 \leq i < z, z \in [\ell] \right\} .$$

In words, this set contains all the states of the game, for any timestep $t$, so that there exists a parameter in $\Theta$ and some $k$-colorings $s^{(0)}, s^{(1)}$ which are everywhere different, that are $N$-consistent with some hypothesis in the class. Also, note that the set

$$\left\{ ((\boldsymbol{\xi}, \boldsymbol{s}^{(0)}, \boldsymbol{s}^{(1)}), \boldsymbol{\eta}) : \begin{array}{ll} h(\theta, \xi_z^{(i)}) = s_z^{(0)}(\xi_z^{(i)}) & \text{if } \eta_z^{(i)} = 0 \\ h(\theta, \xi_z^{(i)}) = s_z^{(1)}(\xi_z^{(i)}) & \text{if } \eta_z^{(i)} = 1 \end{array} \text{ for } 0 \leq i < z, z \in [\ell] \right\} =$$

$$\bigcap_{1 \leq z \leq \ell} \bigcap_{0 \leq i < z} \left\{ ((\boldsymbol{\xi}, \boldsymbol{s}^{(0)}, \boldsymbol{s}^{(1)}), \boldsymbol{\eta}) : h(\theta, \xi_z^{(i)}) = s_z^{(0)}(\xi_z^{(i)}) \text{ if } \eta_z^{(i)} = 0 \right\} \cap$$

$$\left\{ ((\boldsymbol{\xi}, \boldsymbol{s}^{(0)}, \boldsymbol{s}^{(1)}), \boldsymbol{\eta}) : h(\theta, \xi_z^{(i)}) = s_z^{(1)}(\xi_z^{(i)}) \text{ if } \eta_z^{(i)} = 1 \right\} =$$

$$\bigcap_{1 \leq z \leq \ell} \bigcap_{0 \leq i < z} \left\{ ((\boldsymbol{\xi}, \boldsymbol{s}^{(0)}, \boldsymbol{s}^{(1)}), \boldsymbol{\eta}) : h(\theta, \xi_z^{(i)}) = s_z^{(\eta_z^{(i)})}(\xi_z^{(i)}) \right\} .$$

Recall that a Borel set is any set in a topological space that can be formed from open sets through the operations of countable union, countable intersection, and relative complement. By the measurability assumption of Definition 9, the set $\left\{ (\theta, (\boldsymbol{\xi}, \boldsymbol{s}^{(0)}, \boldsymbol{s}^{(1)}), \boldsymbol{\eta}) : h(\theta, \xi_z^{(i)}) = s_z^{(\eta_z^{(i)})}(\xi_z^{(i)}) \right\}$ which corresponds to the subspace of the product space of $\Theta$ and the space of all infinite sequences $((\boldsymbol{\xi}, \boldsymbol{s}^{(0)}, \boldsymbol{s}^{(1)}), \boldsymbol{\eta})$ whose $(z, i)$-th term is consistent with some hypothesis $h$ is Borel measurable (note that the sequence $(\boldsymbol{s}^{(0)}, \boldsymbol{s}^{(1)}), \boldsymbol{\eta})$ uniquely induces a sequence of labels over $[k]$ on which we apply the measurability definition). Moreover, the following hold: (i) the two intersections over $t$ and $\ell$ are countable, (ii) the union over $\theta$ is a projection of a Borel set and (iii) the union over the colorings is countable (it is even finite given $t$). This implies that $\mathcal{W}$ is coanalytic. $\square$

### C.3.2 Pattern Avoidance Algorithm

Our learning algorithm for the linear rates case will be built over the notion of data patterns and pattern avoidance functions. In the exponential rates setting, we focused on the case where the learner is successful if she makes a finite number of mistakes. In the case of linear rates, this is replaced with controlling the *model complexity*, i.e., we have to understand how expressive the class $\mathcal{H}$ is. We will say that a sequence $x_1, y_1, x_2, y_2, ...$ in $(\mathcal{X} \times [k])^\infty$ is *consistent* with $\mathcal{H} \subseteq [k]^{\mathcal{X}}$ if, for every finite $t < \infty$, there exists $h \in \mathcal{H}$ such that $h(x_i) = y_i$ for all $i \leq t$.

We next introduce the crucial notion of pattern. Intuitively, the expressivity of a hypothesis class $\mathcal{H}$ is proportional to the length of the "patterns" realized by concepts in $\mathcal{H}$. Hence, by controlling the length of the realizable patterns, we can obtain learners for $\mathcal{H}$.

**Definition 13.** *Given a sequence $S = (x_1, y_2, x_2, y_2, ...) \in (\mathcal{X} \times [k])^\infty$ that is consistent with $\mathcal{H}$, the binary string $b_t \in \{0,1\}^t$ of length $t$ is a Natarajan-Littlestone pattern (or simply a pattern) if there exists a subsequence of consecutive terms $S' = (x_{z+i}, y_{z+i})_{i \in [t]}$ of length $t$ for some $z \in \mathbb{N}$ so that there exist two $k$-colorings everywhere different $s^{(0)}, s^{(1)}$ of the elements $(x_{z+i})_{i \in [t]}$ with $s^{(0)}(x_{z+i}) = y_{z+i}$ if $b_t = 0$ and $s^{(1)}(x_{z+i}) = y_{z+i}$ if $b_t = 1$.*

We are going to use the winning strategy of the learning player of the game of Figure 6 in order to design an algorithm that *avoids NL patterns in the data*. This is exactly the intuition behind the design of the Gale-Stewart game of Figure 6. The learner predicts a binary string $\eta_t \in \{0,1\}^t$ with goal that this pattern is not realized by the class $\mathcal{H}$, i.e., it is not a NL pattern with respect to a sequence consistent with $\mathcal{H}$. Intuitively, the reason we are aiming for forbidden patterns is the following: before witnessing the realizable sequence we have no control over the complexity of the hypothesis class $\mathcal{H}$, since its Natarajan dimension can be infinite. However, leveraging the fact that there is no infinite NL-tree we show that for every realizable sequence $S$ there is some (sequence-dependent) $n^*(S)$ such that the class $\mathcal{H}_{x_1,y_1,...,x_{n^*},y_{n^*}}$ of the concepts that agree on the first $n^*$ terms has Natarajan dimension that is bounded by $O(n^*)$. This is because if we can produce a forbidden pattern for every $x$-tuple of length $n^*$ we know that the Natarajan dimension of this class is bounded by $n^*$.

The algorithm will get as input a data sequence consistent with $\mathcal{H}$ and will identify patterns in the data. In particular, the algorithm works as follows: For any finite pattern length $t$, the algorithm traverses the data sequence in consecutive blocks of length $t$ and in each such block it tries all the everywhere different colorings for the points and uses the universally measurable winning strategy of the learning player to obtain a guess for a forbidden NL pattern for each of these colorings. The algorithm then checks if at least one of the pairs of colorings and forbidden patterns are actually realized by the data. If this is the case (i.e., the guess was false), the algorithm adds a new point $\xi_t$ in its list of "bad points" and continues searching for patterns of larger length. If the guess was correct and the pattern is not realized in this block, it continues with the same pattern size in the next block.

The algorithm operates as follows. We remark that we can construct the strategy $\eta_t$ of the learner by invoking a notion of ordinal NL dimension in a similar manner as in the exponential rates case.

---

1. Initialize the pattern length $\ell_0 = 1$.
2. At every time step $t \geq 1$:
   - (a) Set $L := \ell_{t-1}$ and create a list $S_L$ of all the possible pairs of everywhere different colorings of $L$ elements.
   - (b) Set $j := 1$ be the index that traverses the list $S_L$.
   - (c) Set $s^{(0)} := S_L[j](0), s^{(1)} := S_L[j](1)$.
   - (d) Compute pattern $b_L = \eta_L(\xi_1, ..., \xi_{L-1}, x_{t-L+1}, ..., x_t, s^{(0)}, s^{(1)}) \in \{0,1\}^L \times [k]^L \times [k]^L$.
   - (e) If it holds that, $s_z^{(0)}(x_{t-L+1+z}) = y_{t-L+1+z}$, if $b_L(z) = 0$, and $s_z^{(1)}(x_{t-L+1+z}) = y_{t-L+1+z}$, otherwise, for $z \in \{0, 1, ..., L-1\}$, then
     - i. Set $\xi_L = (x_{t-L+1}, ..., x_t, s^{(0)}, s^{(1)})$ //The pattern is realized (not forbidden).
     - ii. Update $\ell_t = L + 1$ //Look for larger patterns.
   - (f) Else:
     - i. If $j < L$ then update $j := j + 1$ and go to step (c).
     - ii. Else shift the block, and move to the next timestep, i.e. $\ell_t = L, t := t + 1$.

Figure 7: Pattern Avoidance Algorithm

---

Hence, the above procedure defines a pattern avoidance function

$$\widehat{y}_{t-1}(x_1, ..., x_{\ell_{t-1}}, s^{(0)}, s^{(1)}) := \eta_{\ell_{t-1}}(\xi_1, ..., \xi_{\ell_{t-1}-1}, x_1, ..., x_{\ell_{t-1}}, s^{(0)}, s^{(1)}) \in \{0,1\}^{\ell_{t-1}} \times [k]^{\ell_{t-1}} \times [k]^{\ell_{t-1}}.$$

(1)

The intuition is that as long as the class $\mathcal{H}$ does not have an infinite NL tree and a consistent sequence is provided to the algorithm, then whenever the "if" statement gets True, the game of Figure 6 progresses and the learner gets closer to winning the game. The finiteness of the NL tree gives the next key lemma.

**Lemma 8.** *For any sequence $x_1, y_1, x_2, y_2, ...$ that is consistent with $\mathcal{H}$, the pattern avoidance algorithm of Figure 7, in a finite number of steps, rules out NL patterns in the data sequence, in the sense that the "if" statement in Line (c) is false and $\ell_t = \ell_{t-1} < \infty, \widehat{y}_t = \widehat{y}_{t-1}$ for all sufficiently large $t$. Moreover, the mappings $\ell_t$ and $\widehat{y}_t$ are universally measurable.*

*Proof.* Consider an infinite sequence of times $t = t_1, t_2, ...$ and assume that the "if" condition is true for any time-step in this sequence. Since $\eta_t$ is a winning strategy for the learning player in the Gale-Stewart game $\mathcal{G}$, we have that there exists some finite time $t^\star$ so that $\mathcal{H}_{\xi_1, \eta_1, ..., \xi_{t^\star}, \eta_{t^\star}} = \emptyset$. By the structure of the "if" condition, we have that $\xi_i = (x_{t_i - \ell_{t_i} - 1} + 1, ..., x_{t_i})$ and $\eta_i = (y_{t_i - \ell_{t_i} - 1} + 1, ..., y_{t_i})$. But since the sequence is consistent with $\mathcal{H}$, the class $\mathcal{H}_{\xi_1, \eta_1, ..., \xi_{t^\star}, \eta_{t^\star}}$ must contain the hypothesis that makes it consistent which is a contradiction since the class is empty. For the measurability of the strategies, we refer to Remark 5.4 of [BHM+21]. $\square$

### C.3.3   Asymptotic Behavior of the Algorithm

As in the case of exponential rates, our first goal towards the design of our algorithms is an asymptotic result. We first have to move from the adversarial to a probabilistic setting. Fix a realizable distribution $P$ over $\mathcal{X} \times [k]$ and let $(X_1, Y_1), (X_2, Y_2), ...$ be i.i.d. samples from $P$. As it is clear in this section for the upper bounds, we assume that $\mathcal{H} \subseteq [k]^{\mathcal{X}}$ does not have an infinite NL tree.

For any integer $\ell \in \mathbb{N}$ and any universally measurable pattern avoidance function $g : \mathcal{X}^\ell \times [k]^\ell \times [k]^\ell \to \{0, 1\}^\ell$, we consider the error

$$\mathrm{per}(g) := \mathrm{per}_\ell(g) = \Pr_{(x_1, y_1, ..., x_\ell, y_\ell)} [g \text{ fails to avoid some NL pattern realized by the data } x_1, y_1, ..., x_\ell, y_\ell],$$

i.e., there exist some colorings $s^{(0)}, s^{(1)}$ everywhere different that witness the pattern $g(x_1, ..., x_\ell, s^{(0)}, s^{(1)})$ with colors $(y_1, ..., y_\ell)$. The following lemma establishes this result. It follows the approach in [BHM+21].

**Lemma 9.** *For the algorithm of Equation* (1)*, it holds that*

$$\mathbf{Pr}[\mathrm{per}(\widehat{y}_t) > 0] \to 0 \text{ as } t \to \infty,$$

*where the probability is over the random data that are used to train the algorithm.*

*Proof.* As in the proof of the asymptotic result in the exponential rates, since the distribution $P$ is realizable, we can get that the data sequence $X_1, Y_1, X_2, Y_2, ...$ is consistent with $\mathcal{H}$ almost surely. We can now employ Lemma 8 and get that the random variable

$$T = \sup\{s \geq 1 : \text{the Pattern Avoidance Algorithm of Figure 7 gets in the "if" statement}\}$$

is finite almost surely and $\widehat{y}_s = \widehat{y}_t$ and $\ell_s = \ell_t$ for all $s \geq t \geq T$. We can apply the law of large nymber for $m$-dependent sequences and get that the quantity $\mathbf{Pr}[\mathrm{per}_{\ell_t}(\widehat{y}_t) = 0]$ is equal to

$$\mathbf{Pr}\left[\lim_{S \to \infty} \frac{1}{S} \sum_{s=t+1}^{t+S} 1\{\widehat{y}_t \text{ fails to avoid some NL pattern realized by } (X_s, Y_s, ..., X_{s+\ell_t-1}, Y_{s+\ell_t-1})\} = 0\right].$$

We can decrease the probability of the right-hand side by taking the intersection of the right-hand side event with the event $\{T \leq t\}$. Hence, we have that

$$\mathbf{Pr}[\mathrm{per}_{\ell_t}(\widehat{y}_t) = 0] \geq \mathbf{Pr}[T \leq t] \to 1, \text{ as } t \to \infty,$$

since $T$ is finite with probability one. $\square$

### C.3.4   Linear Learning Rates

Given some pattern avoidance function that is correct on *any* tuple of size $t$, we can essentially use the 1-inclusion graph in a similar manner as if the Natarajan dimension of $\mathcal{H}$ was bounded by $t$. This is established in the following lemma.

**Lemma 10** (Learning NL pattern classes)**.** *Fix $t \geq 1$. Let $g : \mathcal{X}^t \times [k]^t \times [k]^t \to \{0, 1\}^t$ be the universally measurable NL pattern avoidance function of Figure 7. For any $n \geq 1$, there exists a universally measurable classifier $\widehat{Y}_n^g : (\mathcal{X} \times [k])^{n-1} \times \mathcal{X} \to [k]$ such that for every training set $(x_1, y_1, ..., x_n, y_n) \in (\mathcal{X} \times [k])^n$ where $g(x_{i_1}, ..., x_{i_t}, s^{(0)}, s^{(1)})$ is an avoiding NL pattern for all pairwise distinct indices $1 \leq i_1, ..., i_t \leq n$ and everywhere different colorings $s^{(0)}, s^{(1)}$, the classifier achieves a linear permutation bound, i.e.,*

$$\Pr_{\sigma \sim \mathcal{U}(\mathbb{S}_n)} \left[\widehat{Y}_n^g(x_{\sigma(1)}, y_{\sigma(1)}, ..., x_{\sigma(n-1)}, y_{\sigma(n-1)}, x_{\sigma(n)}) \neq y_{\sigma(n)}\right] \leq \frac{t \log(k)}{n}.$$

Note that the above algorithm is transductive. This hints the use of the one-inclusion learning algortihm.

*Proof.* Fix $n \geq 1$ and set $X = [n]$. Also let $F$ be a set of colorings $f : X \to [k]$. We can apply the one-inclusion hypergraph algorithm (see Lemma 1) to get that there exists an algorithm $\mathbb{A}$ such that

$$\Pr_{\sigma \sim \mathcal{U}(\mathbb{S}_n)}[\mathbb{A}(F, \sigma(1), f(\sigma(1)), ..., \sigma(n-1), f(\sigma(n-1)), \sigma(n)) \neq f(\sigma(n))] \leq \frac{\mathrm{Ndim}(F) \log(k)}{n},$$

for any coloring $f \in F$ and $F \in 2^{[k]^X}$. By construction of the algorithm $\mathbb{A}$, the output of the mapping is preserved under relabeling of $X$, i.e.,

$$\mathbb{A}(F, \sigma(1), y_1, ..., \sigma(n-1), y_{n-1}, \sigma(n)) = \mathbb{A}(F \circ \sigma, 1, y_1, ..., n-1, y_{n-1}, n),$$

where $F \circ \sigma = \{f \circ \sigma : f \in F\}$. Moreover, the mapping $\mathbb{A}$ is measurable since its domain is finite (since the number of colors $k$ is a constant).

We will make use of the next result.

**Claim 2.** *Consider the pattern avoidance mapping $g : \mathcal{X}^t \times [k]^t \times [k]^t \to \{0,1\}^t$ of Figure 7. Consider the class $F$ of $k$-colorings of $[n]$ and fix a sequence $\boldsymbol{x} = (x_1, ..., x_n) \in \mathcal{X}^n$. Fix $0 < t \leq n$. Define the subset $F_{\boldsymbol{x}}$ that contains the colorings $f : [n] \to [k]$ that satisfy the following property: For all subsets $(i_1, ..., i_t)$ of $[n]$ of size $t$ and for all colorings $s^{(0)}, s^{(1)} : [n] \to [k]$ which are everywhere different, at least one of the next $t$ conditions is violated:*

$$f(i_j) = s^{(0)}(x_{i_j}) \text{ if } g(x_{i_1}, \ldots, x_{i_t}, s^{(0)}, s^{(1)})[x_{i_j}] = 0,$$

$$f(i_j) = s^{(1)}(x_{i_j}) \text{ if } g(x_{i_1}, \ldots, x_{i_t}, s^{(0)}, s^{(1)})[x_{i_j}] = 1.$$

*Then, we have that $\mathrm{Ndim}(F_{\boldsymbol{x}}) < t$.*

*Proof.* It must be the case that the Natarajan dimension cannot be more than $t - 1$ since we cannot $N$-shatter any $t$-subset of $[n]$. $\square$

Given any input sequence $(x_1, y_1, ..., x_n, y_n)$, we introduce the concept class $F_x$ as in Claim 2. Moreover, consider the mapping $g : \mathcal{X}^t \times [k]^t \times [k]^t \to \{0,1\}^t$ defined to be the pattern avoidance function generated by Figure 7. We can introduce a data-dependent classifier

$$\widehat{Y}_n(g; x_1, y_1, ..., x_{n-1}, y_{n-1}, x_n) = \mathbb{A}(F_x, 1, y_1, ..., n-1, y_{n-1}, n).$$

Note that the above mapping is universally measurable due to Lemma 8. By relabeling we have that

$$\widehat{Y}_n(g; x_{\sigma(1)}, y_{\sigma(1)}, ..., x_{\sigma(n-1)}, y_{\sigma(n-1)}, x_{\sigma(n)}) = \mathbb{A}(F_x, \sigma(1), y_{\sigma(1)}, ..., \sigma(n-1), y_{\sigma(n-1)}, \sigma(n)).$$

By the assumption of the lemma about the input $(x_1, y_1, ..., x_n, y_n)$, we have that the coloring $y(i) = y_i \in [k]$ satisfies $y \in F_x$ (due to the construction of Claim 2). Hence, for such a sequence, we get that

$$\Pr_{\sigma \sim \mathcal{U}(\mathbb{S}_n)}\left[\widehat{Y}_n^g(x_{\sigma(1)}, y_{\sigma(1)}, ..., x_{\sigma(n-1)}, y_{\sigma(n-1)}, x_{\sigma(n)}) \neq y_{\sigma(n)}\right] \leq \frac{\mathrm{Ndim}(F_x) \log(k)}{n}.$$

However, due to the conclusion of Claim 2, we have that the above rate is of order $\log(k) \cdot t/n$, since by construction $\mathrm{Ndim}(F_x) < t$. $\square$

To make use of the results we just stated, we need to come up with a pattern avoidance function that is correct on *any* tuple of size $t$. However, we can only establish that our pattern avoidance function is *eventually* correct, so there is no bound on $t$ that is given to the learner. To deal with this problem, we keep track of some $\widehat{t}_n$ such that, with high probability, the error of the pattern avoidance function is small. Then, we divide our sample into $n/\widehat{t}_n$ batches of size $\widehat{t}_n$. For each batch $i$ we generate some pattern avoidance function $g^i$, whose error probability is small. To achieve the linear rates we run the algorithm from Lemma 10 for every different batch $i$ and function $g^i$ to get a classifier $\widehat{h}^i$. We output the majority vote of the classifiers $\widehat{h}^i$.

**Lemma 11.** *For any $n \in \mathbb{N}$, Consider a training set $\{(X_i, Y_i)\}$ consisting of $n$ points i.i.d. drawn from $P$. Then there exists a universally measurable $\widehat{t}_n = \widehat{t}_n(X_1, Y_1, ..., X_{\lfloor n/2 \rfloor}, Y_{\lfloor n/2 \rfloor})$ whose definition does not depend on $P$ so that the following holds. Set the critical time $t^\star \in \mathbb{N}$ be such that*

$$\mathbf{Pr}[\mathrm{per}(\widehat{y}_{t^\star}) > 0] \le 1/8 \,,$$

*where the probability is over the training set of the algorithm $\widehat{y}_t$. Then, there exist $C, c > 0$ that depend on $P, t^\star$ but not $n$ so that*

$$\mathbf{Pr}[\widehat{t}_n \in T^\star] \ge 1 - Ce^{-cn} \,.$$

*where the probability is over the training of the estimator $\widehat{t}_n$ and $T^\star$ is the set*

$$T^\star = \{1 \le t \le t^\star : \mathbf{Pr}[\mathrm{per}(\widehat{y}_{t^\star}) > 0] \le 3/8\} \,,$$

*where the probability is over the training of $\widehat{y}_t$.*

*Proof.* We consider the strategies

$$\ell_t = L_t(X_1, Y_1, ..., X_t, Y_t)$$

and

$$\widehat{y}_t(z_1, ..., z_{\ell_t}, s^{(0)}, s^{(1)}) = \widehat{Y}_t(X_1, Y_1, ..., X_t, Y_t, z_1, ..., z_{\ell_t}, s^{(0)}, s^{(1)}) \,,\,^{[6]}$$

which can be obtained by the pattern avoidance strategies. We remark that the algorithms behind $L_t$ and $\widehat{Y}_t$ try out all the possible colorings as described in the pattern avoidance algorithm. As a first step we decompose the training set into four parts. We will use the first quarter of the training examples as follows: For any $1 \le t \le \lfloor n/4 \rfloor$ and $1 \le i \le \lfloor n/(4t) \rfloor =: \widehat{N}$, we invoke the above two strategies to get

$$\ell_t^i = L_t(X_{(i-1)t+1}, Y_{(i-1)t+1}, ..., X_{it}, Y_{it})$$

and

$$\widehat{y}_t^i(z_1, ..., z_{\ell_t}, s^{(0)}, s^{(1)}) = \widehat{Y}_t(X_{(i-1)t+1}, Y_{(i-1)t+1}, ..., X_{it}, Y_{it}, z_1, ..., z_{\ell_t}, s^{(0)}, s^{(1)}) \,.$$

For each $t$ in the decomposition, we estimate the value $\mathbf{Pr}[\mathrm{per}(\widetilde{y}_t) > 0]$ using our estimates as

$$\widehat{e}_t = \frac{1}{\widehat{N}} \sum_{i \in [\widehat{N}]} \mathbf{1}\left\{\widehat{y}_t^i \text{ fails to avoid some NL pattern realized by } (X_{s+1}, Y_{s+1}, ..., X_{s+\ell_t^i}, Y_{s+\ell_t^i}) \text{ for some } n/4 \le s \le n/2 - \ell_t^i\right\}$$

We note that almost surely $\widehat{e}_t \le e_t = \sum_{i \in [\widehat{N}]} \mathbf{1}\{\mathrm{per}(\widehat{y}_t^i) > 0\}/\widehat{N}$. Moreover, we set

$$\widehat{t}_n = \inf\{t \le \lfloor n/4 \rfloor : \widehat{e}_t < 1/4\} \,,$$

and we set $\inf \emptyset = \infty$. Set the critical time $t^\star \in \mathbb{N}$ be such that

$$\mathbf{Pr}[\mathrm{per}(\widehat{y}_{t^\star}) > 0] \le 1/8 \,,$$

where the probability is over the training set of the algorithm $\widehat{y}_t$. It holds that

$$\mathbf{Pr}[\widehat{t}_n > t^\star] \le \mathbf{Pr}[\widehat{e}_{t^\star} \ge 1/4] \le \mathbf{Pr}[e_{t^\star} - \mathbf{E}[e_{t^\star}] \ge 1/8] \le \exp(-N^\star/32) \,,$$

where $N^\star = \lfloor n/(4t^\star) \rfloor$. By continuity, there exists $\epsilon > 0$ such that for all $1 \le t \le t^\star$ such that $\mathbf{Pr}[\mathrm{per}(\widehat{y}_t)] > 3/8$, we have that $\mathbf{Pr}[\mathrm{per}(\widehat{y}_t) > \epsilon] > 1/4 + 1/16$. Fix $1 \le t \le t^\star$ with $\mathbf{Pr}[\mathrm{per}(\widehat{y}_t) > 0] > 3/8$ (if such an index exists). Standard concentration inequalities yield

$$\mathbf{Pr}\left[\sum_{i \in [\widehat{N}]} \mathbf{1}\{\mathrm{per}(\widehat{y}_t^i) > \epsilon\} < \frac{\widehat{N}}{4}\right] \le \exp(-N^\star/128) \,.$$

We remark that any NL pattern avoidance function $g$ which satisfies $\mathrm{per}(g) > \epsilon$, we have that

$$\mathbf{Pr}[g \text{ fails to avoid some NL pattern realized by } (X_{s+1}, Y_{s+1}, ..., X_{s+\ell}, Y_{s+\ell}) \text{ for some } n/4 \le s \le n/2 - \ell] \ge 1 - p \,,$$

---

[6]Notice that $L_t, \widehat{Y}_t$ depend also on the colorings that were used throughout the execution of the algorithm. We try the colorings in some fixed order and we slightly abuse the notation to drop that dependence.

where $p = p(\epsilon, n, \ell) = (1 - \epsilon)^{\lfloor (n-4)/(4\ell) \rfloor}$ since there are $\lfloor (n-4)/(4\ell) \rfloor$ disjoint intervals of length $\ell$ in $[n/4 + 1, n/2] \cap \mathbb{N}$.

We now pass to the second quarter of the dataset to test our guesses using the above results. First, the estimates $(\ell_t^i, \widehat{y}_t^i)$ for $1 \leq i \leq \widehat{N}$ are independent of $(X_s, Y_s)$ for $s > n/4$. Second, using a union bound conditionally on the first quarter of the dataset gives that the probability that every guess function $\widehat{y}_t^i$ with $\mathrm{per}^{\ell_t^i}(\widehat{y}_t^i) > \epsilon$ (we let $\mathrm{per}^{\ell}$ denote that the probability is over a sample from the distribution $P^{\otimes \ell}$) makes an error on the second quarter of the dataset is

$$\mathbf{Pr}[\, (\forall i) \, \mathbf{1}\{\mathrm{per}^{\ell_t^i}(\widehat{y}_t^i) > \epsilon\} \leq \mathbf{1}\{E_{i,t}\}\}] \geq 1 - \widehat{N}(1 - \epsilon)^{\lfloor (n-4)/(4t^\star) \rfloor} \,,$$

since $\ell_t^i \leq t^\star$, where $E_{i,t}$ is the event that the $(i,t)$-pattern avoidance estimate $\widehat{y}_t^i$ fails to avoid some NL pattern realized by the data sequence $(X_{s+1}, Y_{s+1}, ..., X_{s+\ell_t^i}, Y_{s+\ell_t^i})$ for some $n/4 \leq s \leq n/2 - \ell_t^i$. This yields that

$$\mathbf{Pr}[\widehat{t}_n = t] \leq \mathbf{Pr}[\widehat{e}_t < 1/4] \leq \lfloor n/4 \rfloor (1 - \epsilon)^{\lfloor (n-4)/(4t^\star) \rfloor} + \exp(-N^\star/32) \,.$$

Taking a union bound over the elements of $T^\star$, where

$$T^\star = \{1 \leq t \leq t^\star : \mathbf{Pr}[\mathrm{per}(\widehat{y}_{t^\star}) > 0] \leq 3/8\} \,,$$

we obtain that

$$\mathbf{Pr}[\widehat{t}_n \notin T^\star] \leq \exp(-N^\star/32) + t^\star \cdot (\lfloor n/4 \rfloor (1 - \epsilon)^{\lfloor (n-4)/(4t^\star) \rfloor} + \exp(-N^\star/32)) \,.$$

This concludes the proof since there exist $C, c > 0$ so that

$$\mathbf{Pr}[\widehat{t}_n \in T^\star] \geq 1 - C \exp(-cn) \,.$$

$\qquad\qquad\qquad\qquad\qquad\qquad\qquad\qquad\qquad\qquad\qquad\qquad\qquad\qquad\qquad\qquad\qquad\quad\square$

**Theorem 13** (Linear Rates). *Assume that class $\mathcal{H} \subseteq [k]^{\mathcal{X}}$ does not have an infinite Natarajan-Littlestone tree. Then, $\mathcal{H}$ admits a learning algorithm that achieves an optimal linear rate.*

*Proof.* The learning algorithm works as follows: It first computes the estimate $\widehat{t}_n$ introduced in Lemma 11. Then it splits the data into two halves: the first half is used to compute the pattern avoidance functions $g^i := \widehat{y}_{\widehat{t}_n}^i$ for $1 \leq i \leq \lfloor n/(4\widehat{t}_n) \rfloor$ and the second half is used in order to apply Lemma 10 and get classifiers $\widehat{y}^i$ with $\widehat{y}^i(x) := \widehat{Y}_{\lfloor n/2 \rfloor + 2}^{g^i}(X_{\lceil n/2 \rceil}, Y_{\lceil n/2 \rceil}, ..., X_n, Y_n, x)$. Finally, the algorithm outputs

$$\widehat{h}_n = \mathrm{Maj}\left(\widehat{y}^1, \widehat{y}^2, ..., \widehat{y}^{\lfloor n/(4\widehat{t}_n) \rfloor}\right) \,.$$

and aims to get $\mathbf{E}[\mathbf{Pr}_{(X,Y)}[\widehat{h}_n(X) \neq Y]] \leq C/n$ for some constant $C$, where the expectation is over the training set used for the predescribed steps. Set $\widehat{N} = \lfloor n/(4\widehat{t}_n) \rfloor$ and $N^\star = \lfloor n/(4t^\star) \rfloor$. With the notation of Lemma 11, we get

$$\mathbf{Pr}\left[\frac{1}{\widehat{N}} \sum_{i \in [\widehat{N}]} \mathbf{1}\{\mathrm{per}(\widehat{y}_{\widehat{t}_n}^i) > 0\} > \frac{1}{100k}, \widehat{t}_n \in T^\star\right] \leq t^\star \exp(-C_1 \cdot N^\star) \,,$$

i.e., the strict majority of the pattern avoidance functions have zero error with high probability where $C_1$ is a fixed constant depending on the (uniformly bounded) number of labels $k \in \mathbb{N}$. Using Lemma 11, we get

$$\mathbf{E}[\mathbf{Pr}[\widehat{h}_n(X) \neq Y]] \leq \mathbf{Pr}\left[\mathrm{Maj}\left(\widehat{y}^1(X), \widehat{y}^2(X), ..., \widehat{y}^{\lfloor n/(4\widehat{t}_n) \rfloor}(X)\right) \neq Y\right]$$

$$\leq C \exp(-cn) + t^\star \exp(-C_1 \cdot n^\star) + p \,,$$

where

$$p = \mathbf{Pr}\left[\widehat{t}_n \in T^\star, \sum_{i \in [\widehat{N}]} \mathbf{1}\{\mathrm{per}(\widehat{y}_{\widehat{t}_n}^i) = 0\} \geq (100k - 1)\widehat{N}/(100k), \mathrm{Maj}\left(\widehat{y}^1(X), \widehat{y}^2(X), ..., \widehat{y}^{\lfloor n/(4\widehat{t}_n) \rfloor}(X)\right) \neq Y\right] \,.$$

We have that

$$\mathbf{Pr}\left[\mathrm{Maj}\left(\widehat{y}^1(X),\widehat{y}^2(X),...,\widehat{y}^{\lfloor n/(4\widehat{t}_n)\rfloor}(X)\right) \neq Y\right] \leq \mathbf{Pr}\left[\sum_{i\in[\widehat{N}]} \mathbf{1}\{\widehat{y}^i(X)\neq Y\} \geq \widehat{N}/k\right],$$

since whenever the majority makes a mistake, the right-hand side event occurs. Since $k$ is a fixed constant, any two sets containing at least $1/k$ and $1-1/(100k)$ fractions of $\{1,2,...,\lceil n/\widehat{t}_n\rceil\}$, must have at least a $\Theta(1)$ fraction in their intersection. Hence, we get that

$$p \leq \mathbf{Pr}\left[\widehat{t}_n \in T^\star, \frac{1}{\widehat{N}}\sum_{i\in[\widehat{N}]} \mathbf{1}\{\widehat{y}^i(X)\neq Y\}\cdot\mathbf{1}\{\mathrm{per}(\widehat{y}^i_{\widehat{t}_n})=0\} \geq \Theta(1)\right]$$

and using Markov's inequality

$$p \leq \Theta(1)\cdot\mathbf{E}\left[\frac{1}{\widehat{N}}\sum_{i\in[\widehat{N}]} \mathbf{1}\{\widehat{t}_n\in T^\star\}\cdot\mathbf{1}\{\widehat{y}^i(X)\neq Y\}\cdot\mathbf{1}\{\mathrm{per}(\widehat{y}^i_{\widehat{t}_n})=0\}\right]$$

We can now apply Lemma 10 conditionally on the first half of the data and get

$$\mathbf{E}[\mathbf{Pr}[\widehat{h}_n(X)\neq Y]] \leq C\exp(-cn)+t^\star\exp(-n^\star/128)+\Theta(1)\cdot\log(k)\cdot\mathbf{E}\left[\mathbf{1}\{\widehat{t}_n\in T^\star\}\cdot\frac{1}{\widehat{N}}\sum_{i\in[\widehat{N}]}\frac{\ell^i_{\widehat{t}_n}}{\lfloor n/2\rfloor+2}\right].$$

Now we have that $\Theta(1)\log(k)=\Theta(1)$ and $\ell^i_{\widehat{t}_n} \leq \widehat{t}_n+1 \leq t^\star+1$, since $\widehat{t}_n\in T^\star$. This gives

$$\mathbf{E}[\mathbf{Pr}[\widehat{h}_n(X)\neq Y]] \leq C/n.$$

$\square$

## C.4 Arbitrarily Slow Rates

The last step that is needed to establish the characterization of learnability in the multiclass setting is to show that if $\mathcal{H}$ has an infinite Natarajan Littlestone tree it is learnable at an arbitrarily slow rate. The following theorem establishes that. The idea of the proof is similar as in the setting with the infinite multiclass Littlestone tree. Intuitively, the reason that in this regime we get arbitrarily slow rates whereas in the other one we get linear rates is that the branching factor now is *exponential* in the depth of the tree, whereas before it remained constant.

**Theorem 14.** *Assume that $\mathcal{H} \subseteq [k]^{\mathcal{X}}$ has an infinite Natarajan-Littlestone tree. Then, any algorithm that learns $\mathcal{H}$ requires arbitrarily slow rates.*

*Proof.* For the proof, fix a vanishing rate $R(t)\to 0$, fix any learning algorithm $\widehat{h}_n$ for $\mathcal{H}$ and let $\{x_u\}$ be an infinite NL tree for $\mathcal{H}$. Consider a random branch of the tree $y=(y_1,y_2,...)$ where the pattern $y_\ell = (y^0_\ell,...,y^{\ell-1}_\ell) \in \{0,1\}^\ell$ is chosen uniformly at random from the $\ell$-dimensional Boolean hypercube for any $\ell\in\mathbb{N}$. Fix a finite level $n\in\mathbb{N}$. By the structure of the NL tree, we know that there exist two different colorings $s_1,s_2$ and a hypothesis $h\in\mathcal{H}$ so that $h$ agrees either with $s_1$ if the pattern bit says 1 or with $s_2$ otherwise. Our first goal in this universal lower bound is to construct a realizable distribution. We define the random distribution that assigns non-zero mass to the points of the branch $y$ with labels consistent with $h$, i.e.,

$$P_y((x^i_{y\leq\ell-1},h(x^i_{y\leq\ell-1}))) = \frac{p_\ell}{\ell} \text{ for } 0\leq i<\ell, \ell\in\mathbb{N},$$

where $p_\ell$ is a sequence of probabilities so that $\sum_{\ell\in\mathbb{N}}p_\ell=1$ that we will select later in the proof. Intuitively, the distribution $P_y$ chooses the node of the infinite branch at level $\ell$ with probability $p_\ell$; this node contains $\ell$ points of $\mathcal{X}$ and one of them is chosen uniformly at random.

By the structure of the infinite NL tree, we get that such a labeling $h\in\mathcal{H}$ exists for any level $n\in\mathbb{N}$ and this labeling is consistent with all the previous levels $1\leq n'<n$. Thus, we get that

$$\mathbf{Pr}_{(x,z)\sim P_y}[h(x)\neq z] \leq \sum_{\ell>n}p_\ell.$$

Hence, as $n \to \infty$, we get that $P_y$ is realizable for any realization of the random branch $y$. Moreover, the mapping $y \to P_y$ is measurable.

We now have to lower bound the loss of the potential learner $\widehat{h}_n$ using the probability measure $P_y$. Let $(X, Z), (X_1, Z_1), (X_2, Z_2), ... \in (\mathcal{X} \times [k])^\infty$ be a collection of i.i.d. samples from $P_y$. Equivalently, we can write

1. $X = x^I_{y \leq T-1}$ and $Z = z^I_T$ for two random variables $(T, I)$ with joint distribution $\mathbf{Pr}[T = \ell, I = i] = \frac{p_\ell}{\ell}$ for $0 \leq i < \ell, \ell \in \mathbb{N}$.

2. For $j \in \mathbb{N}$, set $X_j = x^{I_j}_{y \leq T_j - 1}$ and $Z_j = z^{I_j}_{T_j}$ for two random variables $(T_j, I_j)$ with joint distribution $\mathbf{Pr}[T_j = \ell, I_j = i] = \frac{p_\ell}{\ell}$ for $0 \leq i < \ell, \ell \in \mathbb{N}$.

We underline that in the above the random variables $(T, I), (T_1, I_1), (T_2, I_2)$ are i.i.d. and independent of the random branch $y$. Our goal is to lower bound the error of any learning algorithm: For all $n$ and $\ell$,

$$\mathbf{Pr}[\widehat{h}_n(X) \neq Z, T = \ell] \geq \sum_{i=0}^{\ell-1} \mathbf{Pr}[\widehat{h}_n(x^i_{y \leq \ell - 1}) \neq z^i_\ell, T = \ell, I = i, T_1, ..., T_n \leq \ell, (T_1, I_1), ..., (T_n, I_n) \neq (\ell, i)],$$

where in the right hand side the probability is decreased by additionally requiring that the whole training set is concentrated before the level $\ell + 1$ and the testing example is not contained in the training set. Consider this event $E_{n,\ell,i} = \{T = \ell, I = i, T_1, ..., T_n \leq \ell, (T_1, I_1), ..., (T_n, I_n) \leq (\ell, i)\}$. If we condition on $E_{n,\ell,i}$, we have that the prediction of $\widehat{h}_n(X) = \widehat{h}_n(x^i_{y \leq \ell - 1})$ is independent of the label $z^i_\ell$. Hence, we have that

$$\mathbf{Pr}[\widehat{h}_n(X) \neq Z, T = \ell] \geq \sum_{i=0}^{\ell-1} \mathbf{Pr}[\widehat{h}_n(x^i_{y \leq \ell - 1}) \neq z^i_\ell | E_{n,\ell,i}] \, \mathbf{Pr}[E_{n,\ell,i}] \geq \frac{1}{2} \sum_{i=0}^{\ell-1} \mathbf{Pr}[E_{n,\ell,i}].$$

By the choice of the randomness over $T, I, T_1, I_1, ...$, we get that

$$\mathbf{Pr}[\widehat{h}_n(X) \neq Z, T = \ell] \geq \frac{p_\ell}{2} \left( 1 - \sum_{m > \ell} p_m - \frac{p_\ell}{\ell} \right)^n.$$

To conclude the proof we have to choose the sequence of probabilities $(p_\ell)$ and relate it to the vanishing rate $R$. By combining Lemma 5.12 of [BHM$^+$21] and by applying the reverse Fatou's lemma (see e.g., the end of the proof of Theorem 5.11 of [BHM$^+$21]), the proof is concluded. $\square$

### C.5 A Sufficient Condition for Linear Rates for Multiclass Learning

Another approach to come up with an algorithm that works in the multiclass setting is to use the algorithm that was developed in [BHM$^+$21] for the binary setting. Towards this end, we define a slightly different combinatorial measure for a class $\mathcal{H}$.

**Definition 14.** *A Graph-Littlestone (GL) tree for $\mathcal{H} \subseteq [k]^\mathcal{X}$ of depth $d \leq \infty$ consists of a tree*

$$\bigcup_{0 \leq \ell < d} \{x_u \in \mathcal{X}^{\ell+1}, u \in \{0,1\} \times \{0,1\}^2 \times ... \times \{0,1\}^\ell\}$$

*and a coloring $s$ mapping each position $u^i \in u$ for any node with pattern $u \in \{0,1\} \times ... \times \{0,1\}^\ell$ for $i \in \{0, 1, ..., \ell\}$ and $\ell \in \{0, 1, ..., d-1\}$ of the tree to some color $\{0, 1, ..., k\}$ such that for every finite level $n < d$, the subtree $T_n = \cup_{0 \leq \ell \leq n}\{x_u : u \in \{0,1\} \times \{0,1\}^2 \times ... \times \{0,1\}^\ell\}$ satisfies the following:*

1. *For any path $y \in \{0,1\} \times ... \times \{0,1\}^{n+1}$, there exists a concept $h \in \mathcal{H}$ so that $h(x^i_{y \leq \ell}) = s(x^i_{y \leq \ell})$ if $y^i_{\ell+1} = 1$ and $h(x^i_{y \leq \ell}) \neq s(x^i_{y \leq \ell})$ otherwise, for all $0 \leq i \leq \ell$ and $0 \leq \ell \leq n$, where*

$$\boldsymbol{y}_{\leq \ell} = (y^0_1, (y^0_2, y^1_2), ..., (y^0_\ell, ..., y^{\ell-1}_\ell)), x_{\boldsymbol{y}_{\leq \ell}} = (x^0_{\boldsymbol{y}_{\leq \ell}}, ..., x^\ell_{\boldsymbol{y}_{\leq \ell}}).$$

*We say that $\mathcal{H}$ has an infinite GL tree if it has a GL tree of depth $d = \infty$.*

We note that in the above definition we identify the color $s(x^i_{\boldsymbol{y} \leq \ell})$ with the (unique) position of this point $x^i_{\boldsymbol{y} \leq \ell}$ (since typically the coloring is over positions). One can again verify that if $\mathcal{H}$ has an infinite GL tree, then it has an infinite multiclass Littlestone tree. We show that if a class $\mathcal{H}$ does not have an infinite GL tree, then it is possible to learn $\mathcal{H}$ at a linear rate. The proof of this statement is via a reduction to the binary setting. In fact, we invoke the well-studied "one versus all" approach, where we are trying to learn $k$ different classes $\mathcal{H}_i$ that distinguish between points that belong to the $i$-th class and points that belong to some class $j \neq i$.

**Theorem 15.** *Assume that $\mathcal{H} \subseteq [k]^{\mathcal{X}}$ does not have an infinite Graph-Littlestone tree. Then, there exists an algorithm that learns $\mathcal{H}$ at a linear rate.*

*Proof.* Consider $k$ binary classes $\mathcal{H}_i$ induced by the hypothesis class $\mathcal{H}$ where $\mathcal{H}_i = \{x \mapsto 1\{h(x) = i\} : h \in \mathcal{H}\}$, i.e., for $h \in \mathcal{H}, \widetilde{h} \in \mathcal{H}_i$ we have that $\widetilde{h}(x) = 1$ if and only if $h(x) = i$.

**Claim 3.** *Assume that $\mathcal{H}$ does not have an infinite GL tree. Then for any $i \in [k]$ the class $\mathcal{H}_i$ is learnable at a linear rate.*

*Proof.* Fix $i \in [k]$. It suffices to show that the class $\mathcal{H}_i$ does not have an infinite VCL tree (VCL trees are introduced in [BHM$^+$21] and, intuitively, a VCL tree is a GL tree with $k = 1$ and $s = 1$ everywhere). Towards contradiction, assume that $\mathcal{H}_i$ admits an infinite VCL tree $\mathcal{T} = \{x_u\}$. We can construct the following Graph-Littlestone tree: we use the same nodes as in $\mathcal{T}$ and for any node in the tree we use the coloring that colors each point with the color $i$. Fix an arbitrary level $n$ and a path $y$. We know that there exists a binary hypothesis $\widetilde{h} \in \mathcal{H}_i$ that realizes this path in the VCL tree. Due to the construction of the class $\mathcal{H}_i$, we have that there exists a mapping $h \in \mathcal{H}$ (where $\widetilde{h} = 1\{h(\cdot) = i\}$) that realizes the path in the constructed tree. This property holds for any path and any level. Hence, we have constructed an infinite GL tree for $\mathcal{H}$ which yields a contradiction. $\qquad \square$

Assume that we get $k$ binary classifiers $\widehat{h}_n^{(i)}$, one for each class $\mathcal{H}_i$. Finally, we set $\widehat{h}_n(x) = \operatorname{argmax}_i \widehat{h}_n^i(x)$ for any $x \in \mathcal{X}$. The above claim concludes the proof since

$$\mathbf{E} \Pr_{(x,y) \sim P}[\widehat{h}_n(x) \neq y] = \mathbf{E} \Pr_{(x,y) \sim P}[\exists i \in [k] : \widehat{h}_n^{(i)}(x) \neq \mathbf{1}\{y = i\}] \leq \sum_{i \in [k]} \mathbf{E}[\operatorname{err}(\widehat{h}_n^{(i)})] \leq 1/n \,,$$

since $k$ is a fixed constant. $\qquad \square$

We close this section with an open question. Is the GL tree roughly speaking equivalent to the NL tree? This should remind the reader the connection between the Graph and the Natarajan dimension in the uniform multiclass PAC learning. In fact, it holds that $\operatorname{Ndim}(\mathcal{H}) \leq \operatorname{Gdim}(\mathcal{H}) \leq \operatorname{Ndim}(\mathcal{H}) \cdot O(\log(k))$.

**Open Question 1.** *Let $\mathcal{H} \subseteq [k]^{\mathcal{X}}$ for some fixed constant $k \in \mathbb{N}$.*

1. *Is it true that $\mathcal{H}$ has an infinite NL tree if and only if it has an infinite GL tree?*

2. *Is it possible to obtain an analogue of the inequality between the Graph and the Natarajan dimensions for the ordinal GL and NL dimensions?*

One approach to tackle this problem is to show that if $\mathcal{H}$ has an infinite GL tree, then it is learnable at an arbitrarily slow rate. It is not clear to us that the current proof can be modified to work in this setting. If we try to follow the same steps we cannot guarantee the realizability of the sequence. If we pick an infinite random path and let $p_i$ be the path up to depth $i$, we know that for each such depth there exists some $h_i$ that realizes this path, i.e., if $b_i = 1 \implies h(x_i) = y^{h_i} = s(x_i)$ and if $b_i = 0 \implies h(x_i) = y^{h_i} \neq s(x_i)$. Now if consider some depth $j > i$ then the hypothesis $h_j$ will agree with $h_i$ on every $x_i$ with $b_i = 1$. However, we cannot guarantee that $h_j(x_i) = h_i(x_i)$ if $b_i = 0$. Hence, the straightforward way to modify the proof to account for that would be to consider a *family* of different distributions. Note that this is not allowed in the universal learning setting, since we fix the distribution.

# D  Multiclass Learning for Partial Concepts

In this section, we provide our results on multiclass learnability in the context of partial concept classes.

## D.1  PAC Multiclass Learnability for Partial Concepts: The Proof of Theorem 6

We show the following theorem, which implies Theorem 3. For reader's convenience, we restate Theorem 6.

**Theorem.** *For any partial concept class* $\mathcal{H} \subseteq \{0, 1, ..., k, \star\}^{\mathcal{X}}$ *with* $\mathrm{Ndim}(\mathcal{H}) \leq \infty$, *the sample complexity of PAC learning the class* $\mathcal{H}$ *satisfies*

$$\mathcal{M}(\epsilon, \delta) = O\left(\frac{\mathrm{Ndim}(\mathcal{H}) \log(k)}{\epsilon} \log(1/\delta)\right) \text{ and } \mathcal{M}(\epsilon, \delta) = \Omega\left(\frac{\mathrm{Ndim}(\mathcal{H}) + \log(1/\delta)}{\epsilon}\right).$$

*In particular, if* $\mathrm{Ndim}(\mathcal{H}) = \infty$, *then* $\mathcal{H}$ *is not PAC learnable.*

*Proof.* Our algorithm will make use of the one-inclusion hypergraph algorithm whose utility is provided by Lemma 1 for total concepts. We first show the next lemma for the one-inclusion hypergraph predictor for partial concepts.

**Lemma 12.** *Fix a positive constant* $k$. *For any partial concept class* $\mathcal{H} \subseteq \{0, 1, ..., k, \star\}^{\mathcal{X}}$ *with* $\mathrm{Ndim}(\mathcal{H}) < \infty$, *there exists an algorithm* $\mathbb{A} : (\mathcal{X} \times [k])^* \times \mathcal{X} \to [k]$ *such that, for any* $n \in \mathbb{N}$ *and any sequence* $\{(x_1, y_1), ...., (x_n, y_n)\} \in (\mathcal{X} \times [k])^n$ *that is realizable with respect to* $\mathcal{H}$,

$$\Pr_{\sigma \sim \mathcal{U}(\mathbb{S}_n)}[\mathbb{A}(x_\sigma(1), y_\sigma(1), ..., x_\sigma(n-1), y_\sigma(n-1), x_\sigma(n)) \neq y_\sigma(n)] \leq \frac{\mathrm{Ndim}(\mathcal{H}) \log(k)}{n}.$$

*Proof.* Fix $n \in \mathbb{N}$. Consider a set of points $S = \{x_1, ..., x_n\}$ and let $S_d$ be the set of distinct elements of the sequence $S$. Define the hypothesis class $\mathcal{H}_{S_d}$ that contains all the total functions $h : S_d \to [k]$ such that the sequence $\{(x, h(x)) : x \in S_d\}$ is realizable with respect to $\mathcal{H}$.

CASE A: Assume that $\mathcal{H}_{S_d} \neq \emptyset$. This is a total concept class and so let $\mathbb{A}_{S_d}$ be the algorithm guaranteed to exist by Lemma 1 with $X = S_d$ and $\mathcal{H} = \mathcal{H}_{S_d}$. For any $y_1, ..., y_n \in [k]$ so that the training sequence $(x_1, y_1), ..., (x_n, y_n)$ is realizable with respect to $\mathcal{H}$ (and so realizable with respect to $\mathcal{H}_{S_d}$), define

$$\mathbb{A}(x_1, y_1, ..., x_{n-1}, y_{n-1}, x_n) = \mathbb{A}_{S_d}(\mathcal{H}_{S_d}, x_1, y_1, ..., x_{n-1}, y_{n-1}, x_n).$$

Moreover, we can consider any permutation of the sequence $x_1, ..., x_n$ and let the feature space $S_d$ and the hypothesis class $\mathcal{H}_{S_d}$ the same. Finally, we have that $\mathrm{Ndim}(\mathcal{H}_{S_d}) \leq \mathrm{Ndim}(\mathcal{H})$. This gives the desired bound.

CASE B: Assume that $\mathcal{H}_{S_d}$ is empty. In this case, set $\mathcal{A}(x_1, y_1, ..., x_{n-1}, y_{n-1}, x_n) = 0$ for all sequences $(x_1, ..., x_n) \in \mathcal{X}^n$ and $(y_1, ..., y_{n-1}) \in [k]^{n-1}$ so that $\{h \in \mathcal{H} : h(x_i) = y_i \text{ with } i < n \text{ and } h(x_n) \in [k]\} = \emptyset$. □

Let us now focus on the upper bound given that $\mathrm{Ndim}(\mathcal{H}) < \infty$. For any distribution $P$ realizable with respect to $\mathcal{H}$ and for a sequence of $n$ labeled i.i.d. examples from $P$, we define the strategy $\widehat{h}_n(\cdot) = \mathbb{A}(X_1, Y_1, ..., X_n, Y_n, \cdot)$ and so

$$\mathbf{E}[\mathrm{er}_P(\widehat{h}_n)] = \mathop{\mathbf{E}}_{(X_i, Y_i)_{i \leq n}}\left[\mathop{\Pr}_{(X_{n+1}, Y_{n+1})}[\mathbb{A}(X_1, Y_1, ..., X_n, Y_n, X_{n+1}) \neq Y_{n+1}]\right] \leq \frac{\mathrm{Ndim}(\mathcal{H})\Theta(\log(k))}{n+1}.$$

We next have to convert this algorithm which guarantees an expected error bounded by $\mathrm{Ndim}(\mathcal{H})\Theta(\log(k))/(n + 1)$ into an algorithm that guarantees a bound on the error with probability at least $1 - \delta$. In order to boost the algorithm, we use a standard boosting algorithm by decomposing the dataset into $\log(1/\delta)$ parts and using Chernoff bounds. For the details we refer to the boosting trick of [HLW94] and the proof of Theorem 34.(i) of [AHHM22].

Let $\mathrm{Ndim}(\mathcal{H}) = \infty$. We will show that $\mathcal{H}$ is not PAC learnable. For any $\ell \leq \mathrm{Ndim}(\mathcal{H})$, let $\mathcal{X}_\ell = \{x_1, ..., x_\ell\}$ be a set $N$-shattered by $\mathcal{H}$ using the function $f$. Let $\mathcal{H}_\ell$ be the class of all total functions $\mathcal{X}_\ell \to \{0, 1, ..., \ell\}$, any distribution $P$ on $\mathcal{X}_\ell \times \{0, 1, ..., \ell\}$ realizable with respect

to $\mathcal{H}_\ell$ can be extended to a distribution on $\mathcal{X} \times \{0, 1, ..., \ell\}$ realizable with respect to $\mathcal{H}$ with $P((\mathcal{X} \setminus \mathcal{X}_k) \times \{0, 1, ..., \ell\}) = 0$. Thus, any lower bound on the sample complexity of PAC learning the total concept class $\mathcal{H}_\ell$ is also a lower bound on the sample complexity of learning the partial class $\mathcal{H}$. This gives the desired lower bound. Also, the partial concept classes with infinite Natarajan dimension are not PAC learnable.

$\square$

## D.2 The Proof of Proposition 2

Our goal is to understand *when the ERM principle succeeds in the partial setting*. To address this natural and important task, we pose the following question: What is the difference between learning partial concepts with output in $\{0, 1, ..., k, \star\}$ and learning total concepts with labels in $\{0, 1, ..., k, k+1\}$, where $k$ is a positive integer? Conceptually, the key difference between partial concepts with $k + 1$ labels and $(k + 2)$-label multiclass classification has to do with the support of the distribution: In the latter, the learning problem is a distribution over $\mathcal{X} \times \{0, 1, ..., k, k+1\}$, while in the former we have a distribution only over $\mathcal{X} \times \{0, 1, ..., k\}$ (recall Definition 4). We address this question in Proposition 2. We show that any partial concept class $\mathcal{H}$ is learnable in the $(k + 2)$-label setting if and only if $\mathcal{H}$ is learnable in the partial setting and the VC dimension of the set family $\{\mathrm{supp}(h) : h \in \mathcal{H}\}$ is finite. This result is helpful since it guarantees that when the VC dimension of the above family is bounded, the ERM principle provably holds and can be applied in the partial setting.

The next result gives a formal connection between the two settings.

**Proposition 2.** *Any partial class $\mathcal{H} \subseteq \{0, 1, \ldots, k, \star\}^\mathcal{X}$ is PAC learnable in the $(k + 2)$-label multiclass setting if and only if $\mathcal{H}$ is PAC learnable in the partial concepts setting **and** $\mathrm{VCdim}(\{\mathrm{supp}(h) : h \in \mathcal{H}\}) < \infty$.*

The following proof is an adaptation of Proposition 23 of [AHHM22] to the multiclass setting.

*Proof.* Let us first assume that $\mathcal{H}$ is PAC learnable in the $(k+2)$-label multiclass setting. This implies that $\mathrm{Ndim}_{k+2}(\mathcal{H}) \leq \mathrm{Gdim}_{k+2}(\mathcal{H}) \leq O(\log(k + 2)) \cdot \mathrm{Ndim}_{k+2}(\mathcal{H}) < \infty$. This implies that $\mathcal{H}$ is also learnable in the partial concepts setting by Theorem 3 (by the definitions of the extended Graph and Natarajan dimensions; intuitively in the partial setting, we can use one less color). Now consider the collection of sets $\mathbb{S} = \{\mathrm{supp}(h) : h \in \mathcal{H}\} = \{\{x \in \mathcal{X} : h(x) \neq \star\} : h \in \mathcal{H}\}$. For any sequence of $d$ points $x_1, \ldots, x_d$ shattered by $\mathbb{S}$, let us take the hypothesis $f : \mathcal{X} \to \{0, 1, ..., k, \star\}$ so that $f(x_i) = \star$ for any $i \in [d]$. As an implication, this sequence $x_1, ..., x_d$ is also shattered by $\{x \mapsto \mathbf{1}\{h(x) = f(x)\} : h \in \mathcal{H}\}$. By the definition of the graph dimension, we get that $\mathrm{Gdim}_{k+2}(\mathcal{H}) := \sup_{f:\mathcal{X}\to\{0,1,...,k,\star\}} \mathrm{VCdim}(x \mapsto \mathbf{1}\{h(x) = f(x)\} : h \in \mathcal{H}) \geq \mathrm{VCdim}(\mathbb{S})$. Since $\mathcal{H}$ is PAC learnable in the $(k + 2)$-label multiclass setting, we get that $\mathrm{VCdim}(\{\mathrm{supp}(h) : h \in \mathcal{H}\}) < \infty$.

Let us now assume that $\mathcal{H}$ is PAC learnable in the partial concepts setting and $\mathrm{VCdim}(\{\mathrm{supp}(h) : h \in \mathcal{H}\}) < \infty$. We are going to show that the Natarajan dimension of $\mathcal{H}$ is not infinite in the $(k + 2)$-label multiclass setting. Fix a sequence $(x_i, y_i^{(0)}, y_i^{(1)})_{i\in[d]} \in (\mathcal{X} \times \{0, 1, ..., k, \star\} \times \{0, 1, ..., k, \star\})^d$ for some $d \in \mathbb{N}$, as in the definition of the $(k + 2)$-label Natarajan dimension. For this sequence, consider the set $\{x_i : \star \notin \{y_i^{(0)}, y_i^{(1)}\}\}$ with $y_i^{(0)} \neq y_i^{(1)}$. This set is shattered by the partial concept class $\mathcal{H}$ by the extension of the Natarajan dimension to the partial concepts setting. Moreover, the set $\{x_i : \star \in \{y_i^{(0)}, y_i^{(1)}\}\}$ is shattered by the set $\mathbb{S} = \{\mathrm{supp}(h) : h \in \mathcal{H}\}$. This implies that $\mathrm{Ndim}_{k+2}(\mathcal{H}) \leq \mathrm{Ndim}(\mathcal{H}) + \mathrm{VCdim}(\mathbb{S}) < \infty$. $\square$

## D.3 Multiclass Disambiguations

We extend the definition of [AHHM22] to multiclass partial concepts classes.

**Definition 15** (e.g., [AHHM22]). *A total concept class $\overline{\mathcal{H}} \subseteq [k]^\mathcal{X}$ is a special type of partial concept class such that every $h \in \overline{\mathcal{H}}$ has range $\{0, 1, ..., k\}$, i.e., is a total concept. A total concept class $\overline{\mathcal{H}}$ is said to **disambiguate** a partial concept class $\mathcal{H} \subseteq \{0, 1, ..., k, \star\}^\mathcal{X}$ if every finite data sequence $S \in (\mathcal{X} \times [k])^*$ realizable with respect to $\mathcal{H}$ is also realizable with respect to $\overline{\mathcal{H}}$. In this case, $\overline{\mathcal{H}}$ is called a disambiguation of $\mathcal{H}$.*

We will make use of the following key result in graph theory and communication complexity. We let $\chi(G)$ be the chromatic number of the simple graph $G$. We also set $\mathrm{bp}(G)$ be the biclique partition number of $G$, i.e., the minimum number of complete bipartite graphs needed to partition the edge set of $G$. The following result lies in the intersection of complexity theory and graph theory and is a result of numerous works [HS12, Ama14, AHHM22, SA15, Göö15, GPW18, BDHT17, BBDG$^+$22]. Motivation was the Alon–Saks–Seymour problem in graph theory, which asks: How large a gap can there be between the chromatic number of a graph and its biclique partition number?

**Proposition 3** (Biclique Partition and Chromatic Number [BBDG$^+$22]). *For any $n \in \mathbb{N}$, there exists a simple graph $G$ with $\mathrm{bp}(G) = n$ such that*

$$\chi(G) \geq n^{(\log(n))^{1-\epsilon(n)}},$$

*where $\epsilon(n)$ is a sequence that tends to $0$ as $n \to \infty$.*

Let us consider the binary classification setting. In the work of [AHHM22], it was shown that the (combinatorial version) of the Sauer-Shelah-Perles (SSP) lemma fails in the partial concepts setting. In the total concepts setting, this lemma controls the size of a concept class $\mathcal{H}$ in terms of its VC dimension. Another variant of this lemma controls the growth function of the class. Given a set $C = \{x_1, ..., x_m\} \subseteq \mathcal{X}$, the growth function of $\mathcal{H} \subseteq \{0, 1\}^{\mathcal{X}}$ with respect to $C$ is the cardinality of the set $\Pi_{\mathcal{H}}(C)$ of binary patterns realized by hypotheses in $\mathcal{H}$ when projected to $C$, i.e.,

$$\Pi_{\mathcal{H}}(C) = \{(h(x_1), ..., h(x_m)) : h \in \mathcal{H}\} \subseteq \{0, 1\}^m.$$

We define the growth function of $\mathcal{H}$ at $m \in \mathbb{N}$ as

$$\Pi_{\mathcal{H}}(m) = \sup_{C \subseteq \mathcal{X}: |C| = m} |\Pi_{\mathcal{H}}(C)|.$$

This definition naturally extends to partial concepts where we still look only for binary patterns. Interestingly, while the combinatorial [Sau72] and the growth function [SSBD14] versions of the SSP lemma are both true in the total concepts setting, this is not true in the partial case. We show that the growth function variant still holds, while the combinatorial one fails [AHHM22].

**Lemma 13** (Growth Function - SSP Lemma for Partial Concepts). *Let $\mathcal{H} \subseteq \{0, 1, \star\}^{\mathcal{X}}$ be a partial concept class with finite VC dimension. For any $m \in \mathbb{N}$, it holds that*

$$\Pi_{\mathcal{H}}(m) \leq \sum_{i=0}^{\mathrm{VCdim}(\mathcal{H})} \binom{m}{i}.$$

*Proof.* Set $d = \mathrm{VCdim}(\mathcal{H})$. Consider a set $C = \{x_1, ..., x_m\} \subseteq \mathcal{X}$ of $m$ points and define $\Pi_{\mathcal{H}}(C) = \{(h(x_1), ..., h(x_m)) : h \in \mathcal{H}\} \subseteq \{0, 1\}^m$ (where we ignore any vector that contains the $\star$ symbol). Note that if $m \leq d$, then it holds $|\Pi_{\mathcal{H}}(C)| = 2^m$, by the definition of the VC dimension in the partial setting. We are going to prove the next claim.

**Claim 4.** *For any $C = \{x_1, ..., x_m\}$ and any binary partial concept class $\mathcal{H}$, we have that*

$$|\Pi_{\mathcal{H}}(C)| \leq |\{B \subseteq C : \mathcal{H} \text{ shatters } B\}|.$$

The above claim suffices since the RHS is at most $\sum_{i=0}^{\mathrm{VCdim}(\mathcal{H})} \binom{m}{i}$. Now we prove the above claim. For $m = 1$, the result holds. Assume that the claim is true for sets of size $\ell < m$ and let us prove it for sets of size $m$. Fix some partial binary concept class $\mathcal{H} \subseteq \{0, 1, \star\}^{\mathcal{X}}$ and set $C = \{x_1, ..., x_m\}$. Let $C' = C \setminus \{x_1\}$ and define

$$Y_0 = \{(y_2, ..., y_m) \in \{0, 1\}^{m-1} : (0, y_2, ..., y_m) \in \Pi_{\mathcal{H}}(C) \vee (1, y_2, ..., y_m) \in \Pi_{\mathcal{H}}(C)\},$$

and

$$Y_1 = \{(y_2, ..., y_m) \in \{0, 1\}^{m-1} : (0, y_2, ..., y_m) \in \Pi_{\mathcal{H}}(C) \wedge (1, y_2, ..., y_m) \in \Pi_{\mathcal{H}}(C)\},$$

Note that $|\Pi_{\mathcal{H}}(C)| = |Y_0| + |Y_1|$ (due to double counting). Our first observation is that $|Y_0| \leq |\Pi_{\mathcal{H}}(C')|$, since there may exist some $h \in \mathcal{H}$ which is undefined at $x_1$ and that generates a pattern that is not contained in $Y_0$. Using the inductive hypothesis on $\mathcal{H}$ and $C'$ for the second inequality and the definition of $C'$ for the third equality, we get

$$|Y_0| \leq |\Pi_{\mathcal{H}}(C')| \leq |\{B \subseteq C' : \mathcal{H} \text{ shatters } B\}| = |\{B \subseteq C : x_1 \notin B \wedge \mathcal{H} \text{ shatters } B\}|.$$

Let us now set

$$\mathcal{H}' = \{h \in \mathcal{H} : \exists h' \in \mathcal{H} \text{ such that } (h(x_1), h(C')) = (1 - h'(x_1), h'(C')) \wedge \star \notin h(C)\},$$

i.e., $\mathcal{H}'$ contains all the pairs of hypotheses that (i) are well defined over $C$, (ii) agree on $C'$ and (iii) differ on $x_1$. Note that if $\mathcal{H}'$ shatters a set $B \subseteq C'$ then it also shatters $B \cup \{x_1\}$ and vice versa. Moreover we have that $Y_1 = \Pi_{\mathcal{H}'}(C')$. By the inductive hypothesis on $\mathcal{H}'$ and $C'$, we have that

$$|Y_1| = |\Pi_{\mathcal{H}'}(C')| \leq |\{B \subseteq C' : \mathcal{H}' \text{ shatters } B\}|.$$

This gives that

$$|Y_1| \leq |\{B \subseteq C' : \mathcal{H}' \text{ shatters } B \cup \{x_1\}\}| = |\{B \subseteq C : x_1 \in B \wedge \mathcal{H}' \text{ shatters } B\}|.$$

Finally since $\mathcal{H}'$ lies inside $\mathcal{H}$, we get

$$|Y_1| \leq |\{B \subseteq C : x_1 \in B \wedge \mathcal{H} \text{ shatters } B\}|$$

Combining our observations for $Y_0$ and $Y_1$, we get that

$$|\Pi_{\mathcal{H}}(C)| \leq |\{B \subseteq C : \mathcal{H} \text{ shatters } B\}|.$$

$\square$

We invoke the above SSP variant to prove a bound for the Natarajan dimension of a partial concept class with multiple labels.

**Lemma 14.** *Let $\mathcal{H}_{\mathrm{bin}} \subseteq \{0, 1, \star\}^{\mathcal{X}}$ be a partial concept class with finite VC dimension and $\mathcal{H} = \{h = r(h_1, ..., h_\ell) : h_i \in \mathcal{H}_{\mathrm{bin}}\}$ for some $r : \{0, 1\}^\ell \to [k]$. It holds that*

$$\mathrm{Ndim}(\mathcal{H}) \leq \widetilde{O}(k \cdot \mathrm{VCdim}(\mathcal{H}_{\mathrm{bin}})).$$

*Proof.* Let the VC dimension of the binary concept class be $d$. Let $S \subseteq \mathcal{X}$ be a shattered set by the partial concept class $\mathcal{H}$. Hence we have that

$$|\Pi_{\mathcal{H}}(S)| \geq 2^{|S|}.$$

Now any hypothesis $h \in \mathcal{H}$ is identified by $k$ binary partial concepts from $\mathcal{H}_{\mathrm{bin}}$. We have that

$$|\Pi_{\mathcal{H}}(S)| \leq |\Pi_{\mathcal{H}_{\mathrm{bin}}}(S)|^k,$$

by the structure of $\mathcal{H}$. This implies that

$$|\Pi_{\mathcal{H}}(S)| \leq O(|S|^d),$$

by the properties of the growth function. This gives that $|S| \leq \widetilde{O}(dk)$ and concludes the proof. $\square$

The next theorem is one of the main results of this section; it essentially states that there exists some simple (with small Natarajan dimension) partial concept class $\mathcal{H}^\star$ in the multiclass classification setting which cannot be disambiguated in the sense that any extension of $\mathcal{H}^\star$ to a total concept class has unbounded Natarajan dimension. Hence, for this class, there is no way to assign labels to the undefined points and preserve the expressivity of the induced collection of total classifiers.

**Theorem 16** (Disambiguation). *Fix $k \in \mathbb{N}$. For any $n \in \mathbb{N}$, there exists a partial concept class $\mathcal{H}_n \subseteq \{0, 1, ..., k, \star\}^{[n]}$ with $\mathrm{Ndim}(\mathcal{H}_n) = O_k(1)$ such that any disambiguation $\overline{\mathcal{H}}$ of $\mathcal{H}_n$ has size at least $n^{\log(n)^{1-o(1)}}$, where the $o(1)$ term tends to 0 as $n \to \infty$. This implies that there exists $\mathcal{H}_\infty \subseteq \{0, 1, ..., k, \star\}^{\mathbb{N}}$ with $\mathrm{Ndim}(\mathcal{H}_\infty) = O_k(1)$ but $\mathrm{Ndim}(\overline{\mathcal{H}}) = \infty$ for any disambiguation $\overline{\mathcal{H}}$ of $\mathcal{H}_\infty$.*

*Proof.* The proof is essentially a tensorization of the construction of [AHHM22]. Fix $k, n \in \mathbb{N}$. Set $L = \log_2(k + 1)$ and assume that $L \in \mathbb{N}$ without loss of generality. Let us consider the partial concept class

$$\mathcal{H}_n = \mathcal{H}_n^{(1)} \times \ldots \times \mathcal{H}_n^{(L)},$$

where $\mathcal{H}_n^{(i)} \subseteq \{0, 1, \star\}^{[n]}$. Let us now explain the partial concepts that lie in $\mathcal{H}_n^{(i)}$ and subsequently in $\mathcal{H}_n$. To this end, we invoke Proposition 3 which lies in the intersection of combinatorics and

complexity theory. Consider $L$ independent and disjoint copies of the graph promised by Proposition 3 and let $G$ be the union of these $L$ graphs. Fix $i \in [L]$. Define the partial class $\mathcal{H}_n^{(i)} \subseteq \{0, 1, \star\}^{[n]}$ using the graph $G^{(i)}$ and its $n$ bipartite complete graphs $B_j^{(i)} = (L_j^{(i)}, R_j^{(i)}, E_j^{(i)})$ with $j \in [n]$. The class contains $|V(G^{(i)})|$ concepts, each one identified by a vertex $v \in V(G^{(i)})$ with

$$c_v^{(i)}(j) = \left\{ \begin{array}{ll} 0 & \text{if } v \in L_j^{(i)}, \\ 1 & \text{if } v \in R_j^{(i)}, \\ \star & \text{otherwise.} \end{array} \right\}.$$

Using Lemma 31 from [AHHM22], we have that $\text{VCdim}(\mathcal{H}_n^{(i)}) = 1$.

We overload the "+" notation by setting $g + \star = \star$ for any $g \in \mathbb{N}$. The partial concept class $\mathcal{H}_n$ contains all the partial concepts $h_{v_1,\dots,v_L}(i) = \sum_{j \in [L]} 2^L \cdot c_{v_j}^{(j)}(i) \in \{0, 1, \dots, k, \star\}$. Hence we have that $\mathcal{H}_n \subseteq \{0, 1, \dots, k, \star\}^{[n]}$. We can use the growth function of the partial concepts setting and get that $\text{Ndim}(\mathcal{H}_n) = O(\log(k+1) \cdot \text{VCdim}(\mathcal{H}_n^{(1)})) = O_k(1)$.

Consider some disambiguation $\overline{\mathcal{H}} \subseteq [k]^{[n]}$ of $\mathcal{H}_n$. The class $\overline{\mathcal{H}}$ induces $L$ disambiguations $\overline{\mathcal{H}^{(i)}}$ for the binary partial classes $\mathcal{H}_n^{(i)}$. Then $\overline{\mathcal{H}}$ defines a coloring of $G$ using $\min_i |\overline{\mathcal{H}^{(i)}}|$ colors. Proposition 3 implies that

$$\min_i |\overline{\mathcal{H}^{(i)}}| \geq n^{\log(n)^{1-o(1)}}.$$

Finally, we can consider the class $\mathcal{H}_\infty$ as the disjoint union of $\mathcal{H}_n$. Each $\mathcal{H}_n$ has domain $\mathcal{X}_n$, where the domains $\mathcal{X}_n$ are mutually disjoint and $\mathcal{H}_\infty$ is the union $\bigcup_n \widetilde{\mathcal{H}}_n$, where $\widetilde{\mathcal{H}}_n$ is obtained from $\mathcal{H}_n$ by adding $\star$ outside of its domain. Then $\text{Ndim}(\mathcal{H}_\infty) = O_k(1)$. Since the size of its disambiguations is unbounded, then the multiclass Sauer-Shelah-Perles Lemma [BCHL95] implies that the Natarajan dimension of any disambiguation of $\mathcal{H}_\infty$ is infinite. $\qquad \square$