# OpenReview forum: "Multiclass Learnability Beyond the PAC Framework: Universal Rates and Partial Concept Classes"
_NeurIPS.cc/2022/Conference — NeurIPS 2022 Accept_

### Official Review · Reviewer_EhpP · 2022-07-02

**Rating:** 7
**Confidence:** 4
**Soundness:** 4 excellent
**Presentation:** 2 fair
**Contribution:** 4 excellent

**Summary:**

The authors relax multiclass PAC learning in two directions:
1) Allow the sample complexity to depend on the distribution (whereas in classical PAC learning the sample complexity shall hold for _all_ distributions over the domain).
2) Allow hypotheses to be partial functions, i.e. be defined not everywhere in the domain (in contrast to the classical PAC, where hypotheses must always be total functions).

In direction 1), which the authors call the theory of _universal_ learning, the work of [Bousquet et al. (STOC 2021)](https://dl.acm.org/doi/abs/10.1145/3406325.3451087) is generalized from binary to multiclass classification, namely:
- The learning rate trichotomy is shown: learning occurs either at an exponential or linear or arbitrarily slow rate (in sample size).
- Characterization of learnability in terms of Littlestone and Natarajan-Littlestone trees is given.
- The optimal algorithms for each of the learning rates are presented, while they are not ERMs.

In direction 2), which the authors refer to as the PAC theory of learnability of _partial_ concept classes, the authors generalized the work of [Alon et al. (FOCS 2021)](https://ieeexplore.ieee.org/abstract/document/9719837) again from the binary case to the multiclass one:
- The definition of the Natarajan dimension in the case of partial classes is introduced, and through it a characterization of the learnability of partial classes and (lower & upper) bounds on sample complexity are given.
- It is shown that the learnability of a partial class $\mathbb{H}$ does not imply the learnability of a total class $\bar{\mathbb{H}}$ that _disambiguates_ $\mathbb{H}$.
- The connection between $k$-class partial learnability and $(k + 1)$-class total learnability (when the value "undefined" is treated as an additional label) is established.

**Questions:**

Have you tried to find a less obfuscated partial class to prove Theorem 16? Your class relies on the class from Alon et al. (2021), which in turn relies on the superpolynomial separation between the chromatic number and the biclique partition number, which in turn is based on a series of 4 reductions in graph theory and complexity theory ...

**Limitations:**

Understanding the paper requires familiarity with the two previous works on which it is based (and beyond).

**Strengths And Weaknesses:**

*Strengths*

- A solid work that generalizes the recent two theories from the binary case to the multiclass one. That said, these two recent theories were built in an attempt to bridge the gap between learning theory and applied machine/deep learning practices.

*Weaknesses*

- I am most concerned about the presentation of the material: the density of the narrative is highly heterogeneous throughout the text of the paper. The introduction is well-written, and the motivation of the two proposed theories is immediately clear to the reader. The part about the universal theory of learning is already denser, and by the end of this part it is already difficult to understand what is at stake without looking into the Appendix and googling. The part about partial classes is the most dense, and I personally had to familiarize myself with the binary case first (Alon et al., FOCS 2021) before I could understand anything in this part. My impression is that this submission is an attempt to compress a journal article to the size of a conference paper, and the compression was done very unevenly.
- Although both theories try to fill the gap between the theory and practice of machine learning, I honestly did not understand the connection of one theory with another. More precisely, I can imagine a setup where both theories are combined, but this will be a separate framework.
- In the theory of partial classes, (yet) there is no such simple and clear principle as ERM (This is more about the original work of Alon et al. (2021), but nonetheless). I understand that (for example) in applied deep learning, SGD is not an arbitrary ERM. But after all, SGD is ubiquitous and extremely successful _approximate_ ERM in deep learning. If this theory claims to be an extension of the classical PAC theory explaining modern learning phenomena, I would like to have a simple and understandable learning principle (rather than resorting to improper learning with a transductive algorithm).

---

> ### Author Response · Authors · 2022-08-02
> **Response to Reviewer EhpP (Part 1)**
>
> We thank the reviewer for finding our work solid and interesting, and for providing positive and constructive feedback.
>
> > *I am most concerned about the presentation of the material: the density of the narrative is highly heterogeneous throughout the text of the paper. The introduction is well-written, and the motivation of the two proposed theories is immediately clear to the reader. The part about the universal theory of learning is already denser, and by the end of this part it is already difficult to understand what is at stake without looking into the Appendix and googling. The part about partial classes is the most dense, and I personally had to familiarize myself with the binary case first (Alon et al., FOCS 2021) before I could understand anything in this part. My impression is that this submission is an attempt to compress a journal article to the size of a conference paper, and the compression was done very unevenly.*
>
> Thank you for your constructive feedback. We would like to underline that we prefered to compress the presentation of the partial concepts section due to space constraints. In the first revision of our work, we will make the presentation more balanced and detailed as the reviewer proposes. Moreover, if the paper gets accepted, we will dedicate the extra space of the camera-ready version to providing more intuition about partial concept classes and moving some of the material that appears in the appendix to the main body.
>
> > *Although both theories try to fill the gap between the theory and practice of machine learning, I honestly did not understand the connection of one theory with another. More precisely, I can imagine a setup where both theories are combined, but this will be a separate framework.*
>
> There is an interesting and intrinsic connection between these two settings that we did not highlight in the paper: in the universal learning setting, the first step of the approach is to use the data to simulate some Gale-Stewart games and show that, with high probability, most of them will have ``converged’’, i.e., the function that corresponds to the learning strategy of the learner will be correct. In turn, this defines some **data-dependent** constraints. For instance, assume that $g$ is a successful NL pattern avoidance function, i.e., a function which takes as input any $\ell$ points $x_1,\ldots,x_\ell$ and any two (everywhere different) mappings $s^{(0)}, s^{(1)}$ from points to labels and returns an *invalid* pattern, i.e., a binary pattern $y$  of length $\ell$ that is not compatible with the definition of the Natarajan dimension (i.e., there is no function $h \in H$ s.t. if $y_i = 1$ then $h(x_i) = s^{(1)}(x_i)$ and if $y_i = 0$ then $h(x_i) = s^{(0)}(x_i)$, for all $i \in [\ell]$). Then, we can define a **partial** concept class $H’$, the set of all functions from $X$ to {$1,\ldots,k,\star$} that satisfy the constraint of this pattern avoidance function, and it has two important properties: its Natarajan dimension is bounded by $\ell$ and a learning algorithm for $H’$ also learns $H$. Hence, understanding the learnability of partial concept classes is an essential step in coming up with more natural learning strategies in the universal learning setting. We will underline this connection in the next version of our manuscript.
>
> Moreover, a unifying message of both of these settings is that going beyond the traditional PAC learning framework is essential to understanding the behavior of learning algorithms in practice. Importantly, in both of these settings ERM is not an optimal learning algorithm and the one-inclusion graph predictor is an essential part in deriving results in both theories.

---

> > ### Author Response · Authors · 2022-08-02
> > **Response to Reviewer EhpP (Part 2)**
> >
> > > *In the theory of partial classes, (yet) there is no such simple and clear principle as ERM (This is more about the original work of Alon et al. (2021), but nonetheless). I understand that (for example) in applied deep learning, SGD is not an arbitrary ERM. But after all, SGD is ubiquitous and extremely successful approximate ERM in deep learning. If this theory claims to be an extension of the classical PAC theory explaining modern learning phenomena, I would like to have a simple and understandable learning principle (rather than resorting to improper learning with a transductive algorithm).*
> >
> > This is an exciting research question. We totally agree that it is required to develop algorithmic tools and principles that work with partial concepts and, in general, bridge theory with learning phenomena observed in practice. We underline that our results are very general and hold for any class, without assuming any particular structure on it. As a result, the algorithms we propose are abstract. We believe that an interesting starting point for obtaining simple algorithms for partial classes would be to understand the effect of disambiguation of such classes to total concept classes. This may shed light towards the design of algorithmic principles for partial concept classes. This in turn can lead to more natural algorithms for the universal setting, given the important connection between these two settings that we described in our answer above (part 1 of our response).
> >
> > > *Have you tried to find a less obfuscated partial class to prove Theorem 16? Your class relies on the class from Alon et al.(2021), which in turn relies on the superpolynomial separation between the chromatic number and the biclique partition number, which in turn is based on a series of 4 reductions in graph theory and complexity theory …*
> >
> > This is a very good question. Indeed we have tried various (geometric) binary hypothesis classes and they do not seem to obtain such a disambiguation result. Our approach was to consider a total hypothesis class with infinite VC dimension and then tried to construct a partial restriction of this class with small VC dimension so that any extension of the partial concepts will make the VC dimension unbounded. However, we have not yet managed to make it work. This is a task for future work. To this end, in order to obtain our disambiguation result for the multiclass setting we had to use a tensorization trick and build on the highly non-trivial construction of Alon et al. (2021). In order to make this work, we have to observe that the growth function version of the SSP lemma holds even for partial concept classes (the combinatorial SSP lemma fails, as indicated by Alon et al. (2021)).

---

> > > ### Comment · Reviewer_EhpP · 2022-08-03
> > > **Thank you**
> > >
> > > I am satisfied with the response of the authors. Regardless of the decision on this submission, I think the community will benefit from an expanded version of the work with no page limit, a smoother presentation, and a discussion of the connection between the two theories. Thank you for the work you have done.

---

### Official Review · Reviewer_FfZM · 2022-07-11

**Rating:** 6
**Confidence:** 4
**Soundness:** 3 good
**Presentation:** 3 good
**Contribution:** 2 fair

**Summary:**

This paper studies the problem of multiclass classification with a bounded number of different labels. It mainly extends the main results of [1] and [2] to the multiclass case.

Firstly, the paper provides a complete characterization of the achievable learning rates that holds for every fixed distribution in the universal learning setting and shows that for any concept class, the optimal learning rate is either exponential, linear or arbitrarily slow.

Secondly, the paper considers the structured data, which is captured by partial concept classes proposed by [2].

[1] Olivier Bousquet, Steve Hanneke, Shay Moran, Ramon Van Handel, and Amir Yehudayoff. A theory of universal learning. In Proceedings of the 53rd Annual ACM SIGACT Symposium on Theory of Computing, pages 532–541, 2021.
[2] Noga Alon, Steve Hanneke, Ron Holzman, and Shay Moran. A theory of pac learnability of partial concept classes. In 2021 IEEE 62nd Annual Symposium on Foundations of Computer Science (FOCS), pages 658–671. IEEE, 2022.

**Questions:**

1. As a theoretical work, some of the extensions in the paper just follow the standard idea that generalizing definitions from the binary case to multiclass case (For example, the extension from the Littlestone dimension to the multiclass Littlestone dimension is similar to the standard extension from the VC dimension to the Natarajan dimension. It is similar for the definition of NL tree) and the main results are very similar to those in [1] and [2], hence I have some concerns regarding the technical novelty of this paper. I would appreciate it If the authors can list out some techniques that do not appear in [1] or [2], except for the old-fashioned generalization trick mentioned above. I may adjust the rating according to the quality of the list.
2. Since some characterizations of hypotheses may be very intractable when we extend the binary setting to the multiclass setting. For example, we can exactly calculate the VC dimension of linear binary classifiers, yet we can only bound the order of Natarajan dimension of linear multiclass clasifiers. As a result, I am wondering the practicality of the proposed definitions. Specifically, is it traceable to calculate the Natarajan/multiclass Littlestone tree of some common multiclass hypothetical classes, e.g. linear multiclass classifier class?


[1] Olivier Bousquet, Steve Hanneke, Shay Moran, Ramon Van Handel, and Amir Yehudayoff. A theory of universal learning. In Proceedings of the 53rd Annual ACM SIGACT Symposium on Theory of Computing, pages 532–541, 2021.
[2] Noga Alon, Steve Hanneke, Ron Holzman, and Shay Moran. A theory of pac learnability of partial concept classes. In 2021 IEEE 62nd Annual Symposium on Foundations of Computer Science (FOCS), pages 658–671. IEEE, 2022.

**Ethics Review Area:**

["I don’t know"]

**Strengths And Weaknesses:**

Strengths:
1. This paper extends the main works of [1] and [2] to multi-class classification case. The total work has a good contribution to introducing new frameworks beyond the PAC framework in the multiclass setting. I do like the definition of Natarajan-littlestone tree, which is used to generalize the main results of universal learning framework to the multiclass setting.
3. This paper describes the technique overview and proof sketches to help understanding the main tools and techniques used in this paper.

Weaknesses:
1. Some proofs of the main results, e.g. results in Section 1.3, seem to be an adaption of that in [2], except for the difference between the binary setting and multi-class setting. Hence I have some concerns regarding the novelty of this paper.


[1] Olivier Bousquet, Steve Hanneke, Shay Moran, Ramon Van Handel, and Amir Yehudayoff. A theory of universal learning. In Proceedings of the 53rd Annual ACM SIGACT Symposium on Theory of Computing, pages 532–541, 2021.

[2] Noga Alon, Steve Hanneke, Ron Holzman, and Shay Moran. A theory of pac learnability of partial concept classes. In 2021 IEEE 62nd Annual Symposium on Foundations of Computer Science (FOCS), pages 658–671. IEEE, 2022.

---

> ### Author Response · Authors · 2022-08-02
> **Response to Reviewer FfZM (Part 1 - Contributions)**
>
> We thank the reviewer for appreciating our contribution and for providing positive feedback and questions.
>
> > *As a theoretical work, some of the extensions in the paper just follow the standard idea that generalizing definitions from the binary case to multiclass case (For example, the extension from the Littlestone dimension to the multiclass Littlestone dimension is similar to the standard extension from the VC dimension to the Natarajan dimension. It is similar for the definition of NL tree) and the main results are very similar to those in [1] and [2], hence I have some concerns regarding the technical novelty of this paper. I would appreciate it If the authors can list out some techniques that do not appear in [1] or [2], except for the old-fashioned generalization trick mentioned above. I may adjust the rating according to the quality of the list.*
>
> We wish to first start with the technical challenges concerning the universal multiclass classification setting:
> * The first natural idea that we tried was to reduce the problem to the binary setting, i.e., use the algorithms from [BHM+21] as a black-box to derive algorithms for the multiclass case. However, this approach introduces various technical challenges when dealing with the induced binary hypothesis classes. We discuss in detail these challenges for the exponential rates case. First, it was not clear to us how to prove that if the original class $H$ does not have an infinite multiclass Littlestone tree, then each of the induced binary classes satisfies this property as well. We kindly refer to Line 290-298 of our work.
>
> * As a result, we developed new algorithms from scratch. There were several technical challenges that we had to overcome. The straightforward extension of the Gale-Stewart games that appear in [BHM+21], i.e., where the adversary presents a point $x$ and the learner picks a label in $[k]$ (imitating the online learning game), does not seem to work. Therefore, we propose a more involved Gale-Stewart game where the learner proposes both a point $x$ and two potential labels for it, and the learner has to commit to one of the two. This game allows us to obtain tight upper and lower bounds that depend on the finiteness of the multiclass Littlestone tree. In the next step of the approach, i.e., in the transition from the online setting to the statistical setting, one needs to show how to simulate this Gale-Stewart game using the data that the learner has access to. Since our Gale-Stewart game is more convoluted than the one in [BHM+21], simulating it using data becomes more challenging.
>
> * In the case of linear rates these challenges are even more technically involved and this is related to the fact that the Natarajan-Littlestone tree has a more complicated structure than the VCL tree (for example, we need to check all the possible mappings from some given points to labels). The Gale-Stewart game that handles the case of linear rates goes as follows: the adversary presents to the learner a tuple of points $x$,  and two (different) colorings for these points. Similarly as before, we could not use a simpler game to obtain the result. Subsequently, as in the exponential rates case, simulating this game using data becomes more complicated than in [BHM+21].  We refer to Section 2 of our manuscript for a concrete presentation of our approach.

---

> > ### Author Response · Authors · 2022-08-02
> > **Response to Reviewer FfZM (Part 2 - Contributions)**
> >
> > (continuing our previous response)
> >
> > * It is not clear to us how to prove the arbitrarily slow rates lower bound if one does not use the specific structure of the Natarajan-Littlestone tree that we have defined. Another natural candidate would be the Graph-Littlestone tree. We kindly refer to line 1451 of our paper for the exact definition. The issue with this combinatorial measure is that, if one follows the approach from [BHM+21] to derive the lower bound, it is not clear how to ensure the realizability of the distribution. To be a bit more precise, after we have picked the target path and we consider a node at depth $d$, then we know that there is some $f$ so that if the target path assigns 1 to $s, x_i$ the $f(x_i)=s(x_i)$ and if it assigns 0 then $f(x_i) \neq s(x_i)$. Similarly, at depth $d+1$ there is some $f’$ that satisfies the same properties. The issue is that if some $x_j$ is assigned the value $0$, we merely know that $f(x_j) \neq s(x_j), f’(x_j) \neq s(x_j)$, so there is no way to guarantee that $f(x_j) = f’(x_j)$. Thus, it is not clear that using the classical tricks from the uniform setting to reduce multiclass classification to binary classification works in the universal setting. We are aware of the the results which show that all these dimensions are essentially the same up to $\log k$ factors. However, it is not clear to us how one can use these techniques to show that the Graph-Littlestone tree is finite if and only if the Natarajan-Littlestone tree is finite. The main issue is that we only know that these trees are finite and we have no control over their depth. Thus, one needs to extend these proofs to handle ordinals that are less than $\Omega$. We believe that this is an important question and we hope that our work will lead to future research on whether these tree-based structures are also equivalent. We kindly refer to Open Question 1 at Line 1489 of our paper.
> > To summarize, we have spent quite some time trying to establish our bounds for other dimensions. We have an extensive discussion in the Appendix where we introduce the Graph-Littlestone tree and we show that if a class has only finite such trees, then it is learnable at a linear rate. The situation is similar with the pseudo-dimension.
> >
> > In the setting of partial concept classes, our main contribution is a proof that an alternative version of the SSP lemma, which bounds the growth function, holds in this setting. In contrast, [AHHM21] showed that the traditional version of the lemma (the “combinatorial” one) does not hold for partial concept classes. We remark that these two versions are equivalent for total concept classes. This lemma allows us to ``tensorize’’ the disambiguation lower bound that was proved in [AHHM21] for binary classes and establish it for the multiclass setting. The proof of the learning algorithm for partial concept classes in the multiclass setting follows similarly as the one in [AHHM21] and we have stated it in the paper for completeness of our characterization.

---

> > > ### Author Response · Authors · 2022-08-02
> > > **Response to Reviewer FfZM (Part 3 - Tractability of Dimensions)**
> > >
> > > > *Since some characterizations of hypotheses may be very intractable when we extend the binary setting to the multiclass setting. For example, we can exactly calculate the VC dimension of linear binary classifiers, yet we can only bound the order of Natarajan dimension of linear multiclass classifiers. As a result, I am wondering about the practicality of the proposed definitions. Specifically, is it traceable to calculate the Natarajan/multiclass Littlestone tree of some common multiclass hypothetical classes, e.g. linear multiclass classifier class?*
> > >
> > > This is an interesting point. Let us define the class of linear classifiers with $k$ labels in $\mathbb{R}^d$ as $L = $ { $h_W(x) = \arg\max W x | W \in \mathbb{R}^{k \times d} $}.
> > > We first remark that the Natarajan dimension of $L$ is $\widetilde{\Theta}(kd)$ as noted for instance in [DSBDSS’11] and so for fixed $k,d$ this is a Natarajan class.
> > > Let us discuss the complexity of learning this class in the universal setting. It is important to note that our characterization of universal multiclass learning depends only on whether the two proposed trees are infinite or not. In particular, we note that class $L$ has an infinite multiclass Littlestone tree and a finite Natarajan tree (since it can only shatter a finite number of points as a Natarajan class).
> > > We also remark that if we consider the class of linear multiclass classifiers over $\mathbb{N}^d$ (a discrete geometric space), then this class does not even have an infinite multiclass Littlestone tree and hence is learnable at an exponentially fast rate. To see this, one can use the one-versus-one reduction and observe that any classifier $h_W$ corresponds to a collection of $\binom{k}{2}$ binary classifiers which are halfspaces in $\mathbb{N}^d$ where $h_W^{i,j}(x) = \text{sgn}((W_i - W_j) x)$ for any $i < j$. Then, due to realizability, one can use an argument from infinite Ramsey theory (see Example 2.10 from [BHM+21]) and prove that after a finite number of mistakes, they can detect the correct halfspace for any pair $i<j$. Aggregating these predictors, we get a multiclass linear classifier that enjoys an exponentially fast convergence rate. We will add this example to the next revision of our manuscript.
> > > In general, we agree with the reviewer that it requires some work for general hypothesis classes in order to understand their exact universal convergence rate. Obtaining tools for easily understanding whether a class has a finite “tree” is an interesting and important research direction.

---

### Official Review · Reviewer_XbtE · 2022-07-11

**Rating:** 7
**Confidence:** 3
**Soundness:** 4 excellent
**Presentation:** 3 good
**Contribution:** 3 good

**Summary:**

This paper considers an extension of the work by Bousquet, Hanneke, Moran, Van Handel, and Yehudayoff on universal learning, and the work by Alon, Hanneke, Holzman, and Moran on the partial concept classes to the multiclass classification problems.
In the first part of the paper, the authors consider the same setting as in the universal learning paper, and aim for finding the distribution-dependent rates for the decay of the risk. Similar to the universal learning paper, they show that a trichotomy occurs. More precisely, they show that by replacing the VC dimension with the Natarjan dimension in the universal learning paper, one can prove similar results.
In the second part of the paper, the authors consider the problem of partial concept classes. In this section, the authors, by recent results on the performance of the one-inclusion graph algorithm for multi-class problems, propose a learning algorithm for partial concept classes. More precisely, they show that the learnability in this setting can be identified by natural extension of Natarjan dimension to partial concept classes.


**Questions:**

1- The authors should discuss why they do not consider other dimensions such as Graph dimensions, the Pseudo-dimension, etc.

2- What are the obstacles for obtaining the high probability versions of the bounds for the universal learning problems?

3- Is the dependence of your results on the number of classes optimal? In light of recent results for infinite classes in characterization of multiclass learnability by Brukhim, Carmon, Dinur, Moran, and Yehudayoff, how can one extend the results to infinite classes?

4- For partial concept classes, in prop 2 the authors provide a sufficient condition for success of ERM. How about the necessary condition?

5- In the partial concept classes paper, the authors also present results for the agnostic case. What are the obstacles for proving such a result in the multi-class setting?



**Limitations:**

The authors provide a list of open problems and future directions.

**Strengths And Weaknesses:**

In general, I find this paper very interesting. To me, it seems that the authors follow very closely the results in “a theory of universal learning “ and “A Theory of PAC Learnability of Partial Concept Classes “ to obtain the results. I think this does not affect the quality of the paper. Having said that, I think the authors should discuss with more details the new ideas and techniques beyond the existing ones that they use in the paper. I have checked the proof of the learning algorithm for the partial concept classes, and it follows very closely from the existing results. Therefore, I think 1) the authors should highlight the technical challenges for extending the existing techniques, 2) the authors should discuss why the other dimensions instead of Natarjan dimensions such as graph dimension do not work for these problems.

---

> ### Author Response · Authors · 2022-08-02
> **Response to Reviewer XbtE (Part 1)**
>
> We would like to thank the reviewer for finding our work interesting and for pointing out various comments and questions.
>
> > *I think the authors should discuss with more details the new ideas and techniques beyond the existing ones that they use in the paper. I have checked the proof of the learning algorithm for the partial concept classes, and it follows very closely from the existing results. Therefore, I think 1) the authors should highlight the technical challenges for extending the existing techniques.*
>
> We wish to first start with the technical challenges concerning the universal multiclass classification setting:
> The first natural idea that we tried was to reduce the problem to the binary setting, i.e., use the algorithms from [BHM+21] as a black-box to derive algorithms for the multiclass case. However, this approach introduces various technical challenges when dealing with the induced binary hypothesis classes. We discuss in detail these challenges for the exponential rates case.
>
> First, it was not clear to us how to prove that if the original class $H$ does not have an infinite multiclass Littlestone tree, then each of the induced binary classes satisfies this property as well. We kindly refer to Line 290-298 of our work. As a result, we developed new algorithms from scratch. There were several technical challenges that we had to overcome. The straightforward extension of the Gale-Stewart games that appear in [BHM+21], i.e., where the adversary presents a point $x$ and the learner picks a label in $[k]$ (imitating the online learning game), does not seem to work. Therefore, we propose a more involved Gale-Stewart game where the learner proposes both a point $x$ and two potential labels for it, and the learner has to commit to one of the two. This game allows us to obtain tight upper and lower bounds that depend on the finiteness of the multiclass Littlestone tree. In the next step of the approach, i.e., in the transition from the online setting to the statistical setting, one needs to show how to simulate this Gale-Stewart game using the data that the learner has access to. Since our Gale-Stewart game is more complex than the one in [BHM+21], simulating it using data becomes more challenging.
>
>
> In the case of linear rates these challenges are even more technically involved and this is related to the fact that the Natarajan-Littlestone tree has a more complicated structure than the VCL tree (for example, we need to check all the possible mappings from some given points to labels). The Gale-Stewart game that handles the case of linear rates goes as follows: the adversary presents to the learner a tuple of points $x$,  and two (different) colorings for these points. Similarly as before, we could not use a simpler game to obtain the result. Subsequently, as in the exponential rates case, simulating this game using data becomes more complicated than in [BHM+21].  We refer to Section 2 of our manuscript for a concrete presentation of our approach.
> Proving the arbitrarily slow rates lower bound required some extra care  than in the VCL tree in order to guarantee that the designed distribution is realizable. We kindly refer to the response to your question regarding other dimensions for further details (part 2 of our response).
>
>
> In the setting of partial concept classes, our main contribution is a proof that an alternative version of the SSP lemma, which bounds the growth function, holds in this setting. In contrast, [AHHM21] showed that the traditional version of the lemma (the "combinatorial" one) does not hold for partial concept classes. We remark that these two versions are equivalent for total concept classes. This lemma allows us to "tensorize" the disambiguation lower bound that was proved in [AHHM21] for binary classes and establish it for the multiclass setting. Indeed, as you correctly point out, the proof of the learning algorithm for partial concept classes in the multiclass setting follows similarly as the one in [AHHM21] and we have stated it in the paper for completeness of our characterization. We will explicitly emphasize that this proof is for completeness of presentation, in a first revision.

---

> > ### Author Response · Authors · 2022-08-02
> > **Response to Reviewer XbtE (Part 2)**
> >
> > > *The authors should discuss why the other dimensions instead of Natarjan dimensions such as graph dimension do not work for these problems.*
> >
> > This is an important question. We indeed tried to establish our bounds for other dimensions too. We have an extensive discussion in the Appendix where we introduce the Graph-Littlestone tree and we show that if a class has only finite such trees, then it is learnable at a linear rate. However, it is not clear to us how we can obtain the lower bound when there is an infinite such tree. The main technical difficulty is that if one follows the construction of the lower bound we currently use, it is not clear how to establish that the distribution that is defined is realizable. The situation is similar with the pseudo-dimension. We are aware of the results which show that, in the uniform setting, all these dimensions are the same up to $\log k$ factors. However, the proof techniques that were used do not seem to help us establish a result of the form “the Natarajan-Littlestone tree is finite if and only if the Graph-Littlestone tree is finite”. The main issue is that we only know that these trees are finite and we have no control over their depth. Thus, one needs to extend these proofs to handle ordinals that are less than $\Omega$. We believe that this is an important question and we hope that our work will lead to future research on whether these tree-based structures are also equivalent. We kindly refer to Open Question 1 at Line 1489 of our paper.

---

> > > ### Author Response · Authors · 2022-08-02
> > > **Response to Reviewer XbtE (Part 3 - Questions)**
> > >
> > > > *The authors should discuss why they do not consider other dimensions such as Graph dimensions, the Pseudo-dimension, etc.*
> > >
> > > We kindly refer to the above part of our response.
> > >
> > > > *What are the obstacles for obtaining the high probability versions of the bounds for the universal learning problems?*
> > >
> > > We highlight that we focus on the behavior of the expected misclassification error as the number of samples increases (learning curve), following [BHM+21], in order to make the result cleaner and easier for the reader to compare it to the uniform setting. In the case of exponential rates, high-probability bounds can be obtained directly from our approach. Essentially, we prove that a specific ``good’’ event happens with high probability and under this event we get exponentially small error. For the linear rate case, one would have to use high probability bounds for the error of the one-inclusion hypergraph predictor and follow our approach to derive a high probability bound in this setting.
> > >
> > > > *Is the dependence of your results on the number of classes optimal? In light of recent results for infinite classes in characterization of multiclass learnability by Brukhim, Carmon, Dinur, Moran, and Yehudayoff, how can one extend the results to infinite classes?*
> > >
> > > This is a very interesting question. In our work we focus on the setting of a bounded number of classes. We present our results with a focus on the dependence on the number of samples. The case of exponential rates depends only on the finiteness of the $k$-multiclass Littlestone tree and the rate does not have an explicit dependence on $k$. The number of different classes implicitly affects the finiteness of this tree. As a result, our bounds hold even when one deals with an infinite number of labels, provided that the induced Littlestone tree is of finite depth. For the linear rates case, we make use of the one-inclusion hypergraph algorithm, hence the rate we obtain is of order $\log(k)/n$, so it explicitly depends on the number of labels. We underline that this result holds only if the number of labels is finite. It is a very interesting direction for future work to obtain the linear-rate characterization in the case where the number of labels is infinite. We believe that one needs to define a tree-like version of the DS dimension and adapt the technique of list learning that appeared in [BCD+22] to the universal setting.
> > >
> > > > *For partial concept classes, in prop 2 the authors provide a sufficient condition for success of ERM. How about the necessary conditions?*
> > >
> > > This is a very interesting question as well. We underline that Proposition 2 provides a necessary and sufficient condition for ERM learnability with a (total) class over $k+1$ different labels. This is because there is an equivalence between  finite Natarajan dimension classes with a bounded number of labels and ERM learnability. To characterize ERM learnability in the partial concept class, one needs to provide necessary and sufficient conditions for disambiguation of a partial concept class with finite Natarajan dimension to a total class with finite Natarajan dimension. This question is very challenging even in the binary setting.
> > >
> > > > *In the partial concept classes paper, the authors also present results for the agnostic case. What are the obstacles for proving such a result in the multi-class setting?*
> > >
> > > In fact there are no obstacles in extending our partial concept results to the agnostic case. We chose not to deal with this case for coherence of presentation (in the whole paper we deal with realizable distributions). We will comment on this point in a first revision of our work.

---

> > > > ### Comment · Reviewer_XbtE · 2022-08-07
> > > > **Response**
> > > >
> > > > I would like to thanks the authors for their detailed response. I suggest that the authors include the aforementioned technical challenges in the final version of their paper.

---

### Meta-Review · Area_Chair_ayDt · 2022-08-26

**Recommendation:** Accept
**Confidence:** Certain

**Metareview:**

This work is an extension of the theories of partial concept classes and the universal learning framework to multi-class classification tasks. The reviewers have found the work well-rounded and correct, and of substantial interest to the learning theory sub-community at NeurIPS. A drawback might be that this submission might appear as a natural follow up work on previously established results.

**Award:**

No

---

### Decision · Program_Chairs · 2022-09-14

Accept